# The Skipped Beat:
# A Study of Sociopragmatic Understanding in LLMs for 64 Languages

**Chiyu Zhang**[ξ] **Khai Duy Doan**[λ,*] **Qisheng Liao**[λ,*] **Muhammad Abdul-Mageed**[ξ,λ]

[ξ]Deep Learning & Natural Language Processing Group, The University of British Columbia

[λ]Department of Natural Language Processing & Department of Machine Learning, MBZUAI

{chiyuzh@mail,muhammad.mageed@}.ubc.ca,
{duy.doan,qisheng.liao}@mbzuai.ac.ae

## Abstract

Instruction tuned large language models (LLMs), such as ChatGPT, demonstrate remarkable performance in a wide range of tasks. Despite numerous recent studies that examine the performance of instruction-tuned LLMs on various NLP benchmarks, there remains a lack of comprehensive investigation into their ability to understand cross-lingual sociopragmatic meaning (SM), i.e., meaning embedded within social and interactive contexts. This deficiency arises partly from SM not being adequately represented in any of the existing benchmarks. To address this gap, we present SPARROW, an extensive multilingual benchmark specifically designed for SM understanding. SPARROW comprises 169 datasets covering 13 task types across six primary categories (e.g., anti-social language detection, emotion recognition). SPARROW datasets encompass 64 different languages originating from 12 language families representing 16 writing scripts. We evaluate the performance of various multilingual pretrained language models (e.g., mT5) and instruction-tuned LLMs (e.g., BLOOMZ, ChatGPT) on SPARROW through fine-tuning, zero-shot, and/or few-shot learning. Our comprehensive analysis reveals that existing open-source instruction tuned LLMs still struggle to understand SM across various languages, performing close to a random baseline in some cases. We also find that although ChatGPT outperforms many LLMs, it still falls behind task-specific finetuned models with a gap of 12.19 SPARROW score. Our benchmark is available at: https://github.com/UBC-NLP/SPARROW

## 1 Introduction

Multilingual LLMs have recently transformed NLP, due to their powerful capabilities on a

---

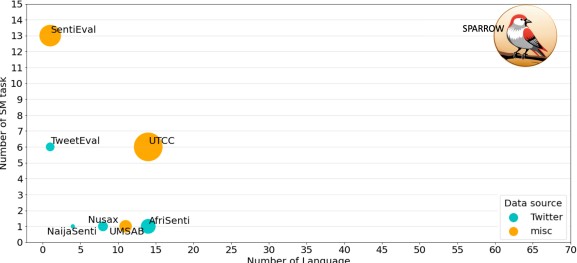

Figure 1: Comparison of SM benchmarks with leaderboards. The bubble size indicates the number of datasets. Previous works: TweetEval (Barbieri et al., 2020), UMSAB (Barbieri et al., 2022), Nusax (Winata et al., 2022), UTCC (Risch et al., 2021), NaijaSenti (Muhammad et al., 2022), AfriSenti (Muhammad et al., 2023a), SentiEval (Zhang et al., 2023b).

wide range of tasks (Xue et al., 2021; Scao et al., 2022). Methods such instruction tuning and reinforcement learning from human feedback (RLHF) (Ouyang et al., 2022) have further enhanced the zero-shot generalizability of these models. Notably, ChatGPT exhibits impressive capabilities in this regard. Human language, however, is intrinsically ambiguous and far from solved. In fact, some forms of meaning are deeply embedded in social interactions. We collectively refer to this type of meaning as *sociopragmatic meaning* (SM). To appreciate SM, consider how the meaning of an utterance in social interaction (e.g., on social media) can be highly subtle and how it incorporates both the social variation related to language users (from a sociolinguistics perspective) (Tagliamonte, 2015) and their communicative intentions (from a pragmatics perspective) (Boxer and Cortés-Conde, 2021). Although SM is quite established within linguistics, NLP systems still struggle with this type of meaning that is intertwined in social and interactive contexts (Zhang and Abdul-Mageed, 2022). The extent to which instruction tuned models such as ChatGPT can capture SM across languages re-

---

* Equal contribution

mains largely unclear as these models are yet to be evaluated on appropriate datasets under standardized conditions easy to replicate.

To facilitate evaluation of LLMs and enhance fairness of model comparisons and reproducibility, early work introduces evaluation benchmarks. Most existing benchmarks, however, focus on the monolingual setting. These include GLUE (Wang et al., 2019), SentEval (Conneau and Kiela, 2018), and TweetEval (Barbieri et al., 2020) for English, ARLUE (Abdul-Mageed et al., 2021) for Arabic, CLUE (Xu et al., 2020a) for Chinese, and IndoNLU (Wilie et al., 2020) for Indonesian. Although XTREME (Hu et al., 2020) and XGLUE (Liang et al., 2020) introduce multilingual benchmarks, they only include a few SM tasks for a limited number of languages. They are also limited to standard language use (e.g., Wikipedia). Barbieri et al. (2022) propose a multilingual sentiment analysis benchmark (UMSAB), but it solely contains tweet sentiment analysis datasets in only eight languages. As such, absence of a unified, diverse, and comprehensive benchmark and a fragmented evaluation landscape hamper NLP work for cross-lingual SM.

Another challenge for SM research is the issue of *data inaccessibility* (Assenmacher et al., 2022). Although many studies release the IDs of posts (e.g., tweets), substantial amounts of these social posts become inaccessible over time due to deletion, etc. (Zhang et al., 2022). In our benchmark, we attempt to re-collect text contents of 25 datasets by using their IDs but can only retrieve 58% samples on average (see Table 8 in Appendix). This data decay also hinders fair comparisons in NLP for SM research. This issue has already become worse as corporations such as Twitter tighten access to their API, making it even harder to collect historical data. To address this bottleneck, we introduce a massively multilingual SM evaluation benchmark, dubbed *SPARROW*, that comprises 169 datasets covering 64 languages from 12 language families, 16 types of scripts, across diverse online platforms (e.g., Twitter, YouTube, and Weibo). We then perform an extensive evaluation of ChatGPT, comparing it to 13 other models ranging in size between 110M-7B parameters. Our evaluations allow us to answer multiple questions related to how it is that LLMs fare across languages on SM. To facilitate future comparisons, we also design a modular, interac-

| Studies | Lang. | Tasks | SM Tasks | Dataset | Models | LeaderBrd |
|---------|-------|-------|----------|---------|--------|-----------|
| Zhong et al. (2023) | en | 5 | 1 | 8 | 5 | ✗ |
| Qin et al. (2023) | en | 7 | 1 | 20 | 29 | ✗ |
| Ahuja et al. (2023) | 70 | 10 | 3 | 16 | 11 | ✗ |
| Laskar et al. (2023) | 12 | 12 | 2 | 140 | 27 | ✗ |
| Bang et al. (2023) | 8 | 8 | 1 | 23 | 3 | ✗ |
| Lai et al. (2023) | 37 | 7 | 0 | 8 | 7 | ✗ |
| Das et al. (2023) | 11 | 2 | 2 | 2 | 1 | ✗ |
| Wang et al. (2023) | en | 5 | 5 | 18 | 3 | ✗ |
| Zhang et al. (2023b) | en | 13 | 13 | 26 | 5 | ✓ |
| Ziems et al. (2023) | en | 24 | 18 | 24 | 13 | ✗ |
| Ours | 64 | 13 | 13 | 169 | 14 | ✓ |

Table 1: Our work in comparison.

tive leaderboard on top of our new benchmark.

To summarize, the contributions of this paper are as follows: **(1)** We collect, uniformize, and responsibly release massively multilingual SM datasets in a *new benchamark*; **(2)** Our SPARROW benchmark is essentially an archive of SM datasets that alleviates the serious issue of *data decay*; **(3)** We evaluate a wide range of models on our SPARROW benchmark via fine-tuning SoTA encoder-only pretrained language models and zero-shot learning of a number of generative models, including instruction tuned models (e.g., BLOOMZ) as well as ChatGPT; and **(4)** We establish standard settings for future research in this area across a large number of languages and tasks. through a *public leaderboard*.

## 2   Related Work

**Evaluation of LLMs.**   There have been many attempts to evaluate ChatGPT and instruction tuned LLMs. Qin et al. (2023); Laskar et al. (2023); Zhong et al. (2023); Wu et al. (2023) utilize existing English evaluation benchmarks, such as GLUE (Wang et al., 2019) and BigBench (Srivastava et al., 2022), to evaluate LLMs' capacities on various NLP tasks. These studies find that although ChatGPT performs less effectively than the models finetuned specifically for each task, it demonstrates superior capabilities compared to other instruction tuned LLMs (e.g., FLAN (Chung et al., 2022)). Ahuja et al. (2023); Bang et al. (2023); Laskar et al. (2023); Lai et al. (2023); Huang et al. (2023) evaluate LLMs on more diverse languages using existing multilingual benchmarks (e.g., XNLI, PAWS-X, XLSum) involving monolingual NLP tasks and crosslingual tasks (e.g., machine translation). Their findings point to a large gap in performance of instruction tuned LLMs and ChatGPT, especially on low-resource languages and those with non-Latin scripts.

SM is still not adequately represented in existing benchmarks, hindering comprehensive evaluations on more languages. As we summarize in Table 1, benchmarks used for listed evaluations only include a few SM tasks focusing on sentiment analysis. Wang et al. (2023); Zhang et al. (2023b) investigate LLMs on a number of SM tasks (e.g., offensive language detection), but only on English. Ziems et al. (2023) investigate ChatGPT performance on a range of computational social science tasks covering subjects such as sociology, psychology, and linguistics, but they again focus only on English. Das et al. (2023) extend evaluation of ChatGPT on hate speech detection to 11 languages. Compared to these works, our objective is to investigate more diverse SM tasks on a massively multilingual setting.

**Sociopragmatic Meaning Benchmarks.** Many previous works introduce unified benchmarks such as GLUE (Wang et al., 2019), SentEval (Conneau and Kiela, 2018), XTREME (Hu et al., 2020), and XGLUE (Liang et al., 2020). These benchmarks include a wide range of NLP tasks, but comprise a sole SM task (i.e., sentiment analysis). Some recent studies started to construct benchmarks focusing on SM: Barbieri et al. (2020) introduce TweetEval benchmark that contains seven English datasets of six SM tasks; Zhang et al. (2023b) develop SentiEval that involves 26 English datasets of 13 sentiment-related tasks. Beyond English, NusaX (Winata et al., 2022), NaijaSenti (Muhammad et al., 2022), and AfriSenti (Muhammad et al., 2023a) propose benchmarks for sentiment analysis with eight Indonesian languages, four African languages, and 14 African languages, respectively. UMSAB introduced by Barbieri et al. (2022) contains 11 sentiment analysis datasets in 11 languages. For detecting antisocial online comments, Risch et al. (2021) introduces a toxic comment collection that contains 43 datasets of six antisocial detection tasks in 14 languages. Compared to these works, our SPARROW benchmark includes significantly more SM tasks and languages, from more diverse sources (refer to Figure 1 for a comparison).

## 3 SPARROW Benchmark

In this section, we describe clusters of tasks in our benchmark as well as our preprocessing and unification. SPARROW consists of 13 types of tasks in six main categories. It contains 169 datasets

|  | Tasks | Dataset | Lang. | LF | Scr |
|---|---|---|---|---|---|
| Antisocial | Aggressive | 1 | 1 | 1 | 1 |
| | Dangerous | 1 | 1 | 1 | 1 |
| | Hate | 16 | 11 | 6 | 5 |
| | Offense | 7 | 6 | 3 | 3 |
| | H/O-Group | 3 | 3 | 2 | 3 |
| | H/O-Target | 8 | 8 | 4 | 7 |
| | Antisocial | 36 | 20 | 7 | 10 |
| **Emotion** | | 26 | 17 | 7 | 5 |
| **Humor** | | 4 | 4 | 1 | 2 |
| Irn&Sarc | Irony | 9 | 7 | 3 | 3 |
| | Sarcasm | 10 | 4 | 3 | 3 |
| | Irony-Type | 1 | 1 | 1 | 1 |
| | Irony&Sarcasm | 20 | 8 | 3 | 3 |
| **Sentiment** | | 77 | 58 | 10 | 15 |
| **Subjectivity** | | 6 | 5 | 2 | 2 |
| **SPARROW** | | 169 | 64 | 12 | 16 |

Table 2: Summary of datasets in SPARROW. **Lang:** number of languages, **LF:** number of language families, **Scr:** number of scripts.

from diverse online platforms and covers a wide period of time (1986-2022). We group different tasks in our benchmark by what we perceive to be an affinity between these tasks. For example, we group tasks of hate speech, offensive language, and dangerous language detection as anti-social language detection. Meanwhile, we keep particular tasks (such as sentiment analysis and emotion recognition) distinct due to the popularity of these tasks and since there are multiple datasets representing each of them. Table 2 summarizes statistics of SPARROW. We now briefly introduce our task clusters. We provide more information about languages in SPARROW in Table 7 of the Appendix. We also provide detailed descriptions with full citations of all our datasets in Tables 9, 10, 11, 12, 13, and 14 in Appendix.

### 3.1 Task Clusters

**Antisocial Language Detection.** The proliferation of antisocial language (e.g., hate speech) toxifies public discourse, incites violence, and undermines civil society (Sap et al., 2019; Vidgen and Derczynski, 2020). Antisocial language detection is thus a useful task. We include under the umbrella of antisocial language the following: **(1)** aggressive language (Kumar et al., 2018), **(2)** dangerous language (Alshehri et al., 2020), **(3)** hate speech (e.g., Waseem and Hovy (2016); Deng et al. (2022)), **(4)** offensive language (e.g., Mubarak et al. (2020); Kralj Novak et al. (2021)), **(5)** offense type identification (e.g., Zampieri et al. (2019)), and **(6)** offense target identification (e.g., Ousidhoum et al. (2019); Jeong et al. (2022)).

**Emotion Recognition.** Emotion affects our decision-making as well as mental and physical health (Abdul-Mageed and Ungar, 2017). SPARROW includes 26 emotion datasets in 17 languages (e.g., Kajava (2018); Bianchi et al. (2021)).

**Humor Detection.** Humor is a type of figurative language which induces amusing effects, such as laughter or well-being sensations. We include four humor detection datasets in four languages (e.g., Blinov et al. (2019); Meaney et al. (2021)).

**Irony & Sarcasm Detection.** Irony and sarcasm also involve figurative language. An ironic/sarcastic expression intentionally uses diametric language to signify implied meaning. We include **(1)** nine irony detection datasets in seven languages (e.g., Xiang et al. (2020)), **(2)** ten sarcasm detection datasets in four languages (e.g., Walker et al. (2012)), and **(3)** an irony type identification dataset (Van Hee et al., 2018).

**Subjectivity and Sentiment Analysis.** Subjectivity analysis aims to understand the opinions, feelings, judgments, and speculations expressed via language (Abdul-Mageed et al., 2014). Our benchmark includes six subjectivity analysis datasets in five different languages (e.g., Pang and Lee (2004); Pribán and Steinberger (2022)). Subjectivity incorporates sentiment. Sentiment analysis (Poria et al., 2020) is one of the most popular tasks in SM understanding where the objective is to identify the underlying sentiment of a given text. Our benchmark contains 77 sentiment analysis datasets in 58 languages (e.g., Pang and Lee (2005); Marreddy et al. (2022)).

## 3.2 Preprocessing, Splits, and Metrics

We apply light normalization on all the samples by converting user mentions and hyperlinks to 'USER' and 'URL', respectively. We standardize label names for consistency across datasets without reassigning nor aggregating the original labels of the datasets. For instance, in certain sentiment analysis datasets, we map '*0*' and '*1*' to '*Negative*' and '*Positive*' respectively. Regarding data splits, if the dataset already has Train, Dev, and Test sets, we maintain the same splits. If the original dataset does not include a Dev set, we randomly select 10% of training data to be a Dev set. In cases without pre-defined splits, we use an 80% Train, 10% Dev, and 10% Test random split. For computing constraints, we also prepare a smaller Test set for

each dataset by randomly sampling 500 samples from Test (if its size exceeds 500). We refer to this smaller test set as Test-S.

We evaluate on each dataset using its original metric as Tables 9, 10, 11, 12, 13, and 14 in Appendix summarize.[1] We report the performance on individual datasets, aggregate datasets into 13 tasks, and report an average score over each task. Moreover, we introduce a metric for each main category, calculated as the average of dataset-specific metrics within that category. Inspired by previous evaluation benchmarks like GLUE (Wang et al., 2019), we define a global metric called *SPARROW score*, which represents the unweighted average of all dataset-specific metrics. The SPARROW score provides an overall indication of performance on SM tasks.

## 4 Evaluation Methods

### 4.1 Finetuning on Encoder-Only Models

We evaluate the following Transformer-encoder-based multilingual models on SPARROW: **(1) Multilingual-BERT** (mBERT) (Devlin et al., 2019) (Base, 110M parameters), **(2) XLM-RoBERTa$_{Base}$** (XLM-R) (Conneau et al., 2020) (270M parameters), **(3) Bernice** (DeLucia et al., 2022), a 270M-parameter model trained with 2.5B tweets in 66 languages, and **(4) InfoDCL** (Zhang et al., 2023a), a SoTA for SM understanding, which further trains XLM-R with 100M tweets in 66 languages with contrastive learning. More details about all models are in Appendix B.

### 4.2 Zero- and Few-Shot on LLMs

We investigate zero-shot performance on a wide range of generative models, including ***pre-trained*** **generative models**: **(1) BLOOM** (Scao et al., 2022), **(2) mT5** (Xue et al., 2021), **(3) LLaMA** (Touvron et al., 2023), ***instruction tuned*** **models**: **(4) BLOOMZ** (Scao et al., 2022), a BLOOM-initialized model tuned with multilingual xP3 corpus, **(5) BLOOMZ-P3** (Muennighoff et al., 2022), a BLOOM-initialized model tuned with English-only P3 corpus, **(6) BLOOM-Bactrian** (Li et al., 2023), a BLOOM-initialized model tuned with 3.4M instruction-following samples in 52 languages, **(7) mT0** (Muennighoff et al., 2022), an mT5 model tuned with xP3 corpus, **(8) Alpaca** (Taori et al., 2023), a

---

[1]We select the macro-average $F_1$ score as the main metric if the original paper utilizes more than one metric.

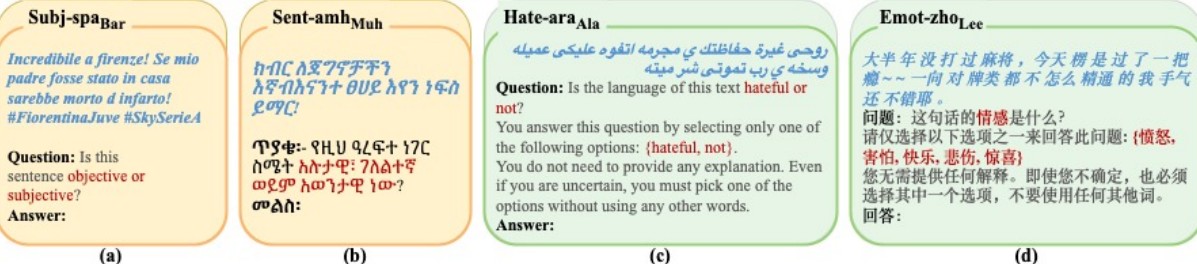

Figure 2: Examples of prompts used for zero-shot evaluation with lm-evaluation-harness ( yellow ) and ChatGPT ( green ). We use an English prompt (Figures a, c) and machine translated the prompt in the corresponding language (Figures b, d), repetively. The prompts construct each task as question-and-answer tasks. The actual input sample is in blue, and the label options are in red.

LLaMA-initialized model tuned with 52K English instruction-following samples, **(9) Vicuna** (Chiang et al., 2023), a LLaMA-initialized model on 70K conversational data, and **(10) ChatGPT**, for which we use the gpt-3.5-turbo-0301 version via OpenAI API.[2] We use 7B-size version of BLOOM- and LLaMA-based models and 4B-size version of mT5-based models. We also evaluate six open-source LLMs (i.e, BLOOM, BLOOMZ-P3, mT5, mT0, LLaMA, and Vicuna) via few-shot in-context learning.

## 5 Experiments

### 5.1 Implementation

**Finetuning.** To keep computation reasonable, we randomly select 45 datasets for hyperparameter tuning and only tune the learning rate of each model.[3] For all experiments, we finetune a pretrained model with an arbitrary batch size of 32 and sequence length of 128 tokens. Each model is finetuned on the full Train set of each dataset for 20 epochs (with patience = 5) based on performance on Dev set. We run each experiment *three* times with different random seeds and identify the best model on Dev in each run. We report the average performance on Test-S over three runs.[4]

**Zero-shot Evaluation.** We perform a zero-shot evaluation on SPARROW for BLOOM-, mT5-, and LLaMA-based models using language model evaluation harness (lm-evaluation-harness Gao et al. (2021)).[5] While we do not tailor prompts

specifically for each model, customized prompts are employed for each set of tasks. These prompts follow the structure of question-and-answer tasks, where we present sample content alongside a task-specific question, as shown in Figure 2. The prompts are summarized in Appendix Table 15. We then instruct the model to generate an answer based on the provided option labels. Each option label represents a potential answer, and we calculate the log-likelihood for each candidate. The prediction with the highest log-likelihood is chosen as the model's final prediction. For the evaluation of ChatGPT, we draw inspiration from previous practices for prompt design (Ziems et al., 2023), and incorporate additional instructions to guide its generation of the desired labels. As shown in Figure 2, we provide an instruction that compels ChatGPT to select a single label for the given input text without providing any additional explanation. We set the temperature to 0 to generate *deterministic and reproducible results* from ChatGPT. For a few instances, we observe that ChatGPT is unable to provide a direct answer. In these cases, we randomly assign a false label to each sample. In addition, we also use machine translation to translate English prompts and label names to the corresponding language of each dataset.[6]

**Few-shot Evaluation.** We utilize lm-evaluation-harness tool with the same prompts employed in zero-shot evaluation to explore the few-shot in-context learning abilities of open-source LLMs. Before the actual test examples, we prepend $m$ examples from the Train set. Each example consists

---

[2]In rest of this paper, ChatGPT refers to gpt-3.5-turbo-0301.

[3]For more information, refer to Section C.1 in Appendix.

[4]We also report the performance of Dev and standard deviations in Appendix Table 16.

[5]https://github.com/EleutherAI/lm-evaluation-harness

[6]We use Google Translate for most languages. NLLB model is used to translate the languages of ace, ban, bjn, bug, and min because Google Translate does not cover these. The prompts of pcm datasets are translated by a native speaker.

of an input text, task-specific instruction, and the corresponding answer. We set $m$ to either 3 or 5.

## 5.2 Results

We present the aggregated performance of Test-S on each task and main category, respectively, in Table 3. We also present test results on all datasets and compare to dataset-specific SoTA performance in Tables 17, 18, 19, 20, 21, and 22 in Appendix.

**(1) How is the overall performance over different models?** *All the fully finetuned models surpass the zero-shot generative models as well as ChatGPT, as shown in Table 3.* The most superior among the finetuned models is InfoDCL, which achieves a SPARROW score of 71.60 and outperforms ChatGPT with 11.56 points SPARROW score. On the other hand, the open-source models (i.e., BLOOM, mT5 and LLaMA) still significantly lag behind on multilingual SM understanding with performance close to a random baseline. Meanwhile, the instruction tuned multilingual LLMs (BLOOMZ and mT0) only slightly perform better than the random baseline.

**(2) Can instruction tuning enhance LLMs' ability on SM understanding?** *Yes, but it depends on the instruction training data.* Following instruction tuning on the English-only P3 dataset, BLOOMZ-P3 demonstrates an improvement of 7.76 SPARROW score compared to BLOOM. Also, BLOOMZ improves 5.85 points over BLOOM (but falls short of BLOOMZ-P3). MT0 also outperforms mT5. However, there remains a substantial gap between all instruction tuned models and finetuned models. BLOOM-Bactrian performs worse than BLOOMZ and BLOOMZ-P3, which are instruction tuned with NLP tasks. This indicates that the general purpose instruction-response dataset is not very useful for SM understanding.

To further probe how instruction tuning improves BLOOM-based models, we compare BLOOM with BLOOMZ-P3 and BLOOMZ in terms of individual tasks, finding sentiment analysis to exhibit the most significant improvement. BLOOMZ-P3 and BLOOMZ achieve a sentiment score improvement of 16.37 and 12.36, respectively, based on average calculation across 77 sentiment analysis datasets. However, BLOOM-Bactrian obtains an improvement of only 1.79 sentiment score, perhaps implying that the Bactrian

instruction-response data is not all that useful for some SM tasks. After tuning mT5 on xP3 dataset, mT0 also experiences a 13.88 improvement in the sentiment score. These may be stemming from inclusion of five English sentiment analysis datasets in both P3 and xP3 during the training phase. For example, we observe that BLOOM, BLOOMZ, BLOOMZ-P3, mT5, and mT0 obtain an accuracy of 56.4, 92.2, 93.00, 49.00, and 76.8 on Sent-eng$_{Soc}$ (not included in either xP3 or P3), respectively and that BLOOM-Bactrian still performs poorly (accuracy= 53.60) after instruction tuning. Again, these results indicate that it is still important to include task-related datasets in the instruction tuning stage.

**(3) How do LLMs perform across different SM tasks?** *They are inferior at humor and antisocial language detection while being relatively better at sentiment and emotion recognition tasks.* BLOOMZ-P3, BLOOMZ, and mT0 exhibit considerable enhancements ($> 5$ points) on sentiment and emotion when compared to their respective initialization models. On the other hand, we find that instruction tuned models perform significantly worse on aggressive language detection and humor detection tasks. BLOOMZ-P3, BLOOMZ, BLOOM-Bactrian, and mT0 all incur a loss of more than 5 points on these two tasks. Upon investigating the predictions of instruction tuned models, we find that they tend to assign negative labels (i.e., non-aggressive or non-humor) which results in many false negative predictions. For a concrete example, we show that BLOOMZ-P3 predict most samples as non-humor in Figure 3a shows.

ChatGPT outperforms the open-source LLMs on all tasks except dangerous language detection. Comparing ChatGPT to InfoDCL, we find gaps favoring InfoDCL in subjectivity analysis (a difference of 9.47), emotion recognition (a difference of 10.68), and irony & sarcasm detection (a difference of 10.70). ChatGPT also largely lags behind InfoDCL in humor detection (a difference of 15.40) and antisocial language detection (a difference of 14.06). As the example shows in Figure 3b, ChatGPT makes more false positive errors (classifies non-hateful as hateful).

**(4) How do LLMs perform across different languages?** We now examine the impact of instruction finetuning on the model's language-wise performance. We categorize the performance of each

| | Rand. | Finetuning | | | | Zero-shot | | | | | | | | | | | | |
|---|---|---|---|---|---|---|---|---|---|---|---|---|---|---|---|---|---|---|
| Tasks | — | mB. | X-R | Ber. | InfoD | BM | BMZ | BMZ (MT) | BMZ P3 | BM Bac. | mT5 | mT0 | mT0 (MT) | LLa. | Alp. | Vic. | CG | CG (MT) |
| | | 110M | 270M | 270M | 270M | 7B | 7B | 7B | 7B | 7B | 4B | 4B | 4B | 7B | 7B | 7B | 175B | 175B |
| **Antisocial** Aggressive | 43.14 | 72.71 | 74.64 | **75.45** | 73.96 | 51.06 | 15.82 | 15.82 | 18.72 | 16.37 | 53.67 | 15.82 | 22.00 | 18.31 | 49.29 | 25.07 | **63.53** | 54.36 |
| Dangerours | 42.06 | 62.36 | 63.57 | **67.13** | 65.23 | 46.87 | 46.87 | **50.84** | 46.87 | 46.87 | 49.31 | 46.87 | 46.87 | 46.87 | 46.87 | 46.87 | 37.93 | 33.68 |
| Hate | 43.62 | 72.97 | 74.37 | 76.76 | **75.85** | 39.83 | 39.44 | 37.76 | 38.52 | 42.23 | 23.29 | 37.33 | 39.05 | 37.80 | 44.31 | 41.59 | **66.06** | 58.74 |
| Offense | 39.48 | 77.53 | 75.88 | 78.45 | **78.88** | 41.06 | 40.42 | 20.28 | 38.59 | 40.43 | 24.99 | 39.90 | 21.11 | 39.85 | 16.82 | 48.70 | **67.31** | 52.70 |
| H/O-Group | 14.82 | 46.18 | 42.39 | **51.15** | 50.24 | 13.63 | 17.26 | 14.23 | 21.23 | 14.81 | 7.02 | 16.25 | 17.01 | 12.35 | 14.13 | 9.26 | **39.66** | 26.74 |
| H/O-Target | 20.39 | 53.16 | 57.67 | **60.96** | 60.79 | 18.73 | 19.03 | 18.74 | 16.89 | 18.77 | 6.69 | 20.58 | 17.99 | 19.32 | 16.83 | 17.01 | **35.89** | 28.67 |
| **AS** | 35.20 | 66.92 | 67.99 | **71.14** | 70.61 | 33.70 | 32.80 | 27.93 | 31.97 | 33.79 | 20.14 | 32.02 | 28.79 | 31.68 | 30.55 | 34.50 | **56.55** | 47.40 |
| **Emotion** | 15.86 | 61.42 | 66.87 | 68.13 | **69.27** | 9.71 | 17.18 | 13.85 | 15.07 | 15.19 | 7.75 | 27.87 | 24.21 | 15.14 | 31.80 | 18.12 | **59.58** | 50.85 |
| **Humor** | 49.65 | 84.35 | 85.19 | 86.75 | **87.05** | 41.78 | 33.12 | 33.82 | 33.17 | 33.04 | 35.91 | 43.60 | 33.12 | 39.78 | 41.72 | 46.19 | 71.65 | **72.70** |
| **I&S** Irony | 42.39 | 64.24 | 65.53 | 66.88 | **68.38** | 36.63 | 35.15 | 38.69 | 44.46 | 36.18 | 36.52 | 34.69 | 33.99 | 40.78 | 27.49 | 47.48 | **58.23** | 56.24 |
| Sarcasm | 45.48 | 72.41 | 73.40 | 74.78 | **74.94** | 43.00 | 41.62 | 32.23 | 32.22 | 41.68 | 46.34 | 36.09 | 41.62 | 41.17 | 32.48 | 47.67 | **65.55** | 65.55 |
| Irony-Type | 22.36 | 47.35 | 46.43 | 56.04 | **57.58** | 18.83 | 18.83 | 18.83 | 18.83 | 18.83 | 18.83 | 18.83 | 18.83 | 18.83 | 18.83 | 18.83 | **30.81** | 30.81 |
| **I&S** | 42.93 | 67.48 | 68.51 | 70.29 | **71.12** | 38.92 | 37.57 | 34.46 | 41.79 | 40.39 | 35.42 | 37.36 | 32.35 | 39.87 | 29.56 | 46.14 | **60.41** | 59.63 |
| **Sentiment** | 34.68 | 66.34 | 69.58 | 70.44 | **71.64** | 26.67 | 39.03 | 28.61 | 43.03 | 28.46 | 20.77 | 34.65 | 32.76 | 27.55 | 25.84 | 25.02 | **60.34** | 54.94 |
| **Subjectivity** | 41.41 | 72.54 | 74.45 | 74.80 | **75.73** | 44.12 | 29.45 | 30.69 | 30.73 | 39.65 | 37.35 | 41.64 | 36.16 | 42.30 | 30.44 | 38.73 | **66.26** | 59.33 |
| **SPARROW** | 33.47 | 66.60 | 69.38 | 70.85 | **71.60** | 27.94 | 33.79 | 27.17 | 35.70 | 29.45 | 21.45 | 33.63 | 30.85 | 28.75 | 28.79 | 29.36 | **60.04** | 53.90 |

Table 3: SPARROW benchmark Test-S results. We report the average of dataset-specific metrics in a task and a category, respectively. **Rand.:** random baseline, **mB.:** mBERT, **X-R:** XLM-R, **Ber.:** Bernice, **InfoD:** InfoDCL, **BM:** BLOOM, **LLa.:** LLaMA, **Alp.:** Aplaca, **Vic.:** Vicuna, **CG:** ChatGPT, **MT:** using machine translated prompts. The best performance in each setting is **Bold**. The red font denotes a performance lower than the random baseline.

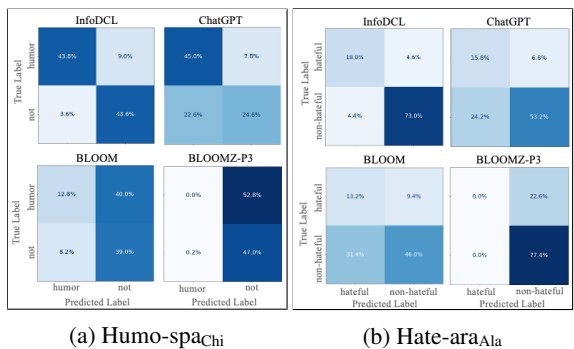

(a) Humo-spa_Chi          (b) Hate-ara_Ala

Figure 3: Confusion matrices of two datasets.

| Lang | Random | InfoDCL | BMZ-P3 | mT0 | Vicuna | CG | CG-MT |
|---|---|---|---|---|---|---|---|
| amh | 37.95 | **65.68** | 16.05 | 22.49 | 2.99 | 20.62 | 46.82 |
| bug | 30.77 | **71.55** | 34.60 | 18.27 | 12.90 | 34.63 | 30.86 |
| ell | 41.24 | **79.13** | 46.71 | 45.47 | 48.21 | 60.94 | 34.98 |
| eng | 37.90 | **75.48** | 43.32 | 39.23 | 39.75 | 66.51 | — |
| fil | 52.37 | **79.01** | 34.47 | 34.47 | 34.47 | 69.13 | 66.67 |
| heb | 47.60 | **95.80** | 71.20 | 76.60 | 40.80 | 84.20 | 57.40 |
| hin | 35.24 | **67.55** | 28.92 | 26.20 | 29.06 | 52.63 | 48.30 |
| mal | 31.68 | **82.70** | 43.84 | 41.65 | 24.85 | 44.03 | 31.44 |

Table 4: Language-wise model performance for sample languages. The complete results are in Table 23 in Appendix. Best performance in each language is **bold**, and the second best is in green highlight . The red font denotes a performance lower than the random baseline.

dataset based on language and calculate the average language scores across all datasets within a language. Since each language contains different tasks and datasets, a direct comparison across languages is not feasible. Therefore, we compare the relative performance between different models for each language. By comparing the instruction tuned models to their initial models, we observe that *most languages experience improvement*. However, we also observe a significant decline in performance for the Amharic (amh) dataset among these models. Specifically, BLOOMZ-P3, BLOOMZ, and mT0 experience a deterioration of 36.07, 24.99, and 26.12 points, respectively, compared to their respective initial models. We hypothesize that this deterioration can be attributed to catastrophic forgetting after instruction tuning, where Amharic was not included in the training set and does not share the writing scripts with the other included languages.

Similarly, the Filipino (fil) tasks exhibit an average decline of approximately 11 points on both BLOOMZ-P3 and BLOOMZ, as Filipino is not included in the xP3 dataset. Although Hindi is included in the xP3 dataset, the three instruction tuned models still show a decline in performance. Upon examining the individual performance of Hindi datasets, we find that the major deteriorations occur in the aggressive language detection and humor detection tasks, while the emotion recognition and sentiment analysis tasks show improvement. The instruction-response data for training Alpaca and Vicuna consist solely of English language. Therefore, we compare the performance of Alpaca and Vicuna to that of LLaMA using both English and non-English datasets. We observe that Alpaca and Vicuna outperform LLaMA when evaluated on English datasets, achieving

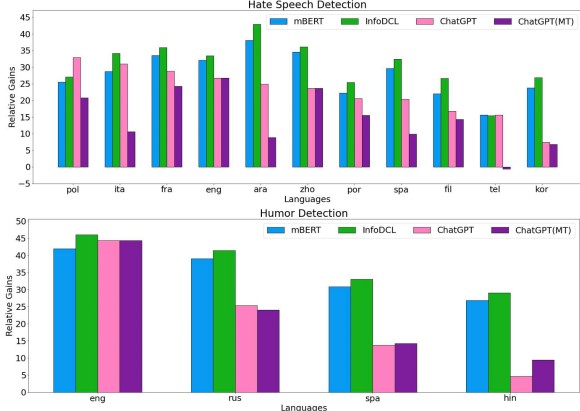

Figure 4: A comparison of different models on multiple tasks across various languages. We show the relative gain of each model compared to the random baseline.

| Task | #Sample | CG (sample MT) | GPT-4 |
|------|---------|----------------|-------|
| Hate-eng$_{Was}$ | 96 | — | 28.82 |
| Hate-eng$_{Dav}$ | 168 | — | 76.30 |
| Hate-ara$_{Ala}$ | 153 | **38.82** | 35.76 |
| Hate-ita$_{Bos}$ | 104 | 29.75 | **38.44** |
| Hate-fil$_{Cab}$ | 145 | 15.76 | **23.96** |
| Hate-ara$_{Mul}$ | 127 | 29.49 | **36.33** |
| Hate-eng$_{Bas}$ | 178 | — | 19.23 |
| Hate-spa$_{Bas}$ | 174 | **21.06** | 17.35 |
| Hate-por$_{For}$ | 142 | 35.37 | **37.34** |
| Hate-pol$_{Pta}$ | 54 | 35.62 | **49.57** |
| Hate-kor$_{Moo}$ | 250 | 14.20 | **15.85** |
| Hate-ara$_{Mub}$ | 89 | **20.54** | 16.74 |
| Hate-zho$_{Den}$ | 125 | 29.24 | **39.24** |
| Hate-kor$_{Jeo}$ | 169 | 27.75 | **44.84** |
| Hate-tel$_{Mar}$ | 2 | **100.00** | **100.00** |
| Sexi-fra$_{Chi}$ | 113 | 34.10 | **43.69** |
| Humo-hin$_{Agg}$ | 220 | 22.73 | **30.91** |
| Humo-rus$_{Bli}$ | 136 | 26.89 | **47.31** |
| Humo-spa$_{Chi}$ | 152 | 30.68 | **36.80** |
| Humo-eng$_{Mea}$ | 48 | — | 50.34 |

Table 5: Case study on using machine translated input and GPT-4 on samples mispredicted by ChatGPT.

average scores of $8.30$ and $5.51$, respectively. However, their performance declines when tested on non-English datasets, resulting in average decreases of $1.53$ and $0.33$, respectively. Compared to task-specific InfoDCL, ChatGPT performs poorly in 63 out of 64 languages, sometimes with a large gap (e.g., $45.06$ lower on Amharic, $38.67$ lower on Malayalam, and $36.91$ lower on Buginese), as Table 4 shows.

We also investigate how different models perform on SM tasks across various languages. Results for two tasks, hate speech detection (top) and humor detection (bottom), are presented in Figure 4. The dataset for each task is grouped according to language, and the average score of each language is obtained. The relative gain of each model against the random baseline is shown, allowing us to compare across these languages.[7] We observe that InfoDCL is the best model across various tasks and languages, with the exception of hate speech in Polish where ChatGPT outperforms it. As Figure 4 shows, ChatGPT performs better for Western languages on hate speech detection. We can also observe wider gaps in hate speech detection between ChatGPT and InfoDCL on Arabic and Korean. Similarly, while ChatGPT demonstrates satisfactory performance in English humor, it remains at significant distance behind InfoDCL in Hindi humor.

**(5) Do machine translated prompts help LLMs?** *Not in general, but they do help in a few cases.* We find, in Table 3, that the SPAR-ROW score of ChatGPT with machine translated

prompts is $6.14$ points lower than ChatGPT with English prompts. Meanwhile, a few tasks such as humor and sarcasm acquire improvements. We also observe a similar pattern for BLOOMZ and mT0, as Table 3 shows. The low-resource languages with non-Latin scripts experience more performance drops in general, which is in line with findings by Lai et al. (2023). Hebrew (heb) and Greek (ell) get the largest performance drops (over 25 points in each case), as shown in Table 4.

**(6) Does GPT-4 outperform ChatGPT?** *Yes, it does.* We provide a study on probing GPT-4's capacities. We exploit 20 datasets from two tasks (i.e., hate speech and humor detection) in 12 languages, only choosing samples whose labels Chat-GPT predicted incorrectly. We refer to this test set as `GPTHard` and provide samples from it to GPT-4 in their original language, employing the same English prompts as those used by ChatGPT. As Table 5 shows, GPT-4 significantly outperforms ChatGPT (McNemar's test with $\alpha < 0.01$) on 19 datasets.[8]

**(7) Can translating input samples into English help improve ChatGPT's predictions?** *Yes, it can.* Here, we use the non-English part of `GPTHard` (16 datasets). We translate these test samples into English using ChatGPT and subsequently employ the translated text and English prompt for classification. As Table 5 shows, we acquire a noteworthy enhancement in ChatGPT's performance (McNemar's test with $\alpha < 0.01$)

---

[7]We note that different annotation artifacts across the different languages still make direct comparisons challenging.

[8]An exception is one dataset where a significance test is not possible due to small sample size ($n = 2$).

| Tasks | Zero-shot | | | | | | Three-shot | | | | | | Five-shot | | | | | |
|---|---|---|---|---|---|---|---|---|---|---|---|---|---|---|---|---|---|---|
| | BM | BMZ P3 | mT5 | mT0 | LLa. | Vic. | BM | BMZ P3 | mT5 | mT0 | LLa. | Vic. | BM | BMZ P3 | mT5 | mT0 | LLa. | Vic. |
| Aggressive | 51.06 | 18.72 | 53.67 | 15.82 | 18.31 | 25.07 | 46.43 | 43.93 | 40.73 | 41.88 | 43.70 | 44.53 | 47.92 | 47.63 | 33.80 | 37.52 | 50.66 | **55.27** |
| Dangerours | 46.87 | 46.87 | **49.31** | 46.87 | 46.87 | 46.87 | 46.87 | 46.87 | 45.68 | 46.87 | 46.87 | 46.87 | 46.87 | 46.87 | 48.91 | 46.87 | 46.87 | 46.87 |
| Hate | 39.83 | 38.52 | 23.29 | 37.33 | 37.80 | 41.59 | 38.83 | 38.30 | 39.43 | 37.82 | 43.51 | **49.17** | 37.95 | 37.14 | 39.53 | 37.70 | 41.87 | 48.37 |
| Offense | 41.06 | 38.59 | 24.99 | 39.90 | 39.85 | 48.70 | 43.94 | 40.25 | 21.99 | 41.53 | 46.36 | **54.49** | 42.42 | 40.59 | 34.10 | 41.67 | 43.83 | 51.72 |
| H/O-Group | 13.63 | **21.23** | 7.02 | 16.25 | 12.35 | 9.26 | 11.43 | 13.04 | 7.92 | 15.98 | 11.81 | 14.19 | 9.68 | 11.76 | 7.23 | 14.50 | 12.58 | 16.27 |
| H/O-Target | 18.73 | 16.89 | 6.69 | 20.58 | 19.32 | 17.01 | 17.09 | 18.41 | 10.48 | 16.55 | 20.56 | **24.84** | 17.44 | 17.45 | 9.32 | 16.56 | 20.09 | 23.60 |
| AS | 33.70 | 31.97 | 20.14 | 32.02 | 31.68 | 34.50 | 33.14 | 32.55 | 27.19 | 32.36 | 36.42 | **41.69** | 32.43 | 31.88 | 29.17 | 32.09 | 35.35 | 40.99 |
| Emotion | 9.71 | 15.07 | 7.75 | 27.87 | 15.14 | 18.12 | 17.08 | 12.17 | 10.66 | 23.12 | 32.12 | 40.28 | 18.48 | 12.35 | 10.07 | 25.57 | 34.20 | **41.79** |
| Humor | 41.78 | 33.04 | 43.60 | 33.12 | 39.78 | 46.19 | 33.67 | 33.12 | 44.70 | 38.19 | 55.20 | 57.15 | 34.06 | 33.12 | 40.20 | 37.08 | 53.86 | **58.75** |
| Irony | 36.63 | 44.46 | 36.52 | 34.69 | 40.78 | **47.48** | 42.21 | 41.58 | 42.61 | 35.18 | 36.76 | 39.78 | 44.34 | 44.14 | 39.67 | 34.82 | 38.40 | 41.61 |
| Sarcasm | 43.00 | 41.68 | 36.09 | 41.62 | 41.17 | 47.67 | 46.14 | 42.91 | 46.72 | 48.42 | 49.75 | **52.55** | 45.43 | 43.05 | 45.75 | 39.88 | 49.03 | 52.51 |
| Irony-Type | 18.83 | 18.83 | 18.83 | 18.83 | 18.83 | 18.83 | 18.83 | 18.83 | 18.83 | 18.83 | 18.83 | 18.83 | 18.83 | 18.83 | 18.83 | 18.83 | 18.83 | **18.83** |
| I&S | 38.92 | 41.79 | 35.42 | 37.36 | 39.87 | 46.14 | 43.01 | 41.11 | 43.48 | 40.98 | 42.36 | 45.12 | 43.61 | 42.33 | 41.67 | 36.55 | 42.74 | 45.92 |
| Sentiment | 26.67 | **43.03** | 20.77 | 34.65 | 27.55 | 25.02 | 34.81 | 35.79 | 24.76 | 31.37 | 37.73 | 34.53 | 33.15 | 37.71 | 23.17 | 29.25 | 40.88 | 39.37 |
| Subjectivity | 44.12 | 30.73 | 37.35 | 41.64 | 42.30 | 38.73 | 36.50 | 30.66 | 37.11 | 34.36 | 44.20 | 54.77 | 33.70 | 30.72 | 39.36 | 31.42 | 46.53 | **56.15** |
| SPARROW | 27.94 | 35.70 | 21.45 | 33.63 | 28.75 | 29.36 | 32.76 | 31.91 | 26.12 | 31.71 | 37.82 | 39.44 | 32.03 | 32.84 | 25.47 | 30.44 | 39.48 | **41.97** |

Table 6: Evaluating open-source LLMs on SPARROW with few-shot in-context learning. The best performance in each setting is **Bold**. The red font denotes a performance lower than the random baseline.

when using the translated input. We also observe that when fed with these English-translated samples, ChatGPT is able to surpass GPT4 with the original inputs in three datasets (i.e., Hate-ara$_{Ala}$, Hate-spa$_{Bas}$, Hate-ara$_{Mub}$). These results suggest that although ChatGPT has inferior ability on several languages in terms of detecting SM, a translate-then-detect approach may be possible.

**(8) How do open-source LLMs perform with few-shot in-context learning?** As Table 6 shows, we compare three-shot and five-shot results with zero-shot results. Based on SPARROW score, we observe that few-shot learning does enhance the performance of BLOOM, mT5, LLaMA, and Vicuna. With the increasing number of shots, the performance of LLaMA and Vicuna increases. Vicuna obtains SPARROW scores of 29.36, 39.44, and 41.97 with zero, three, and five shots, respectively. However, BLOOMZ-P3 and mT0 do not improve with few-shot learning. We suspect this is because the instruction fine-tuning of these two models only uses a zero-shot template that hurts their few-shot learning capacities. BLOOMZ-P3 and mT0 are also different from BLOOM and LLaMA in that they are fine-tuned on several NLP tasks only one of which is an SM task (i.e., sentiment analysis). This probably biases the behavior of these two models.

**(9) Are the open-source LLMs sensitive to prompts used?** We carry out a study to probe the open-source LLMs' sensitivity to prompts. We curate 55 datasets across four tasks from SPARROW and evaluate six models with the prompts we used for evaluating ChatGPT. As Table 24 in Appendix shows, we find that BLOOM, LLaMA, and Vicuna incur sizable performance drops ($>$ 6 points decrease across 55 datasets), while BLOOMZ-P3, mT5, and mT0 demonstrate performance levels akin to those observed in previous experiments ($<$ 2 points different). We leave a more comprehensive evaluation of prompt sensitivity as future work.

## 6 Public Leaderboard

To facilitate future work, we design a public leaderboard for scoring models on SPARROW. Our leaderboard is *interactive* and offers *rich metadata* about the various datasets in our benchmark. It also encourages users to submit information about their models (e.g., number of parameters, time to convergence, pretraining datasets). We also distribute a new *modular toolkit* for fine-tuning or evaluating models on SPARROW.

## 7 Conclusion

In order to understand the abilities of ChatGPT and other instruction tuned LLMs on capturing sociopragmatic meaning, we introduced a massively multilingual evaluation benchmark, dubbed SPARROW. The benchmark involves 169 datasets covering 64 languages from 12 language families and 16 scripts. Evaluating ChatGPT on SPARROW, we find it struggles with different languages. We also reveal that task-specific models finetuned on SM (much smaller than ChatGPT) consistently outperform larger models by a significant margin even on English.

## 8 Limitations

**Benchmark Construction.** Our SPARROW benchmark only includes text classification tasks related to SM. Despite our best efforts, we acknowledge that our benchmark has not covered existing SM datasets exhaustively. We will continue expanding this benchmark and welcome future datasets or metric contributions to it. We also plan to extend SPARROW to more types of tasks related to SM, such as span-based sentiment analysis (Xu et al., 2020b), affective language generation (Goswamy et al., 2020), and conversational sentiment analysis (Ojamaa et al., 2015). We only include text-based SM tasks. Another improvement direction is to extend this benchmark to more tasks that involve more modalities, such affective image captioning (Mohamed et al., 2022) and multi-modal emotion recognition (Firdaus et al., 2020).

**Model Selection.** Due to computation constraints, we cannot evaluate on model sizes > 7B. However, we hope SPARROW will be used in the future to evaluate larger-sized models. Again, due to budget constraints, we only conduct a relatively small case study on GPT-4 and do not evaluate more diverse commercial instruction tuned models that are more expensive (e.g., `text-davinci-003` by OpenAI).

**Experiments.** While we customize prompts employed for each task, we do not tailor prompts specifically for each model. We acknowledge that the performance of models may be influenced by different prompt variants. In future work, we will test diverse prompt variations for more robust results. We only experiment with machine translated prompts in our analyses and acknowledge that the performance drop may stem from the poor quality of machine translation. We will investigate the utility of human translated prompts in a future study. In this paper, we only evaluate LLMs on zero-shot learning. The adoption of few-shot in-context learning may enhance performance, which we also leave to future work.

## Ethics Statement and Broad Impacts

**Data Collection and Releasing.** All the 169 datasets are produced by previous research. Since there are large numbers of datasets and languages in SPARROW, it is hard to manually verify the quality of all the datasets. As a quality assurance measure, we only include in SPARROW datasets that are introduced in peer-reviewed published research. To facilitate access to information about each dataset, we link to each published paper describing each of these datasets inTables 9, 10, 11, 12, 13, and 14.

Following privacy protection policies, we anonymize all SPARROW data as described in Section 3.2. With reference to accessibility of the original individual dataset, SPARROW data can be categorized into three releasing strategies: **(1)** In the case of datasets requiring approval by the original authors, we require future researchers to obtain approval first and will share our splits once approval has been obtained. We indicate these nine datasets in our data description tables. **(2)** For the 25 datasets (see Table 8 in Appendix) that are shared via tweet IDs, we share our obtained data for research use. By doing so, we expect to mitigate the issue of data decay and allow fair comparisons. **(3)** We will share the other 135 publicly accessible datasets upon request. We will also require a justification for responsible use of the datasets. Each dataset will be shared in our Train, Dev, and Test splits along with a dataset card to indicate the original publication of the dataset.

**Intended Use.** The intended use of SPARROW benchmark is to construct a scoring board to facilitate model comparisons as well as enhance fairness and reproducibility across different languages and tasks. We also aim to mitigate data decay issues in social media research. SPARROW could help researchers investigate model's capacity on SM tasks across languages. SPARROW may also be used to investigate model transferability across a wide range of tasks and diverse languages in different settings (such as zero- or few-shot settings and prompting).

**Potential Misuse and Bias.** We notice that some annotations in the datasets of SPARROW (e.g., for hate speech task (Waseem and Hovy, 2016)) can carry annotation and temporal biases. We recommend that any dataset in SPARROW not be used for research or in applications without careful consideration of internal biases of the datasets and potential biases of the resulting systems. We also suggest that users of SPARROW not only focus on the overall SPARROW score but also a model performance on each task and

dataset. The SPARROW score is an unweighted average score over all the dataset-specific metrics, which may lose the fine-grained information and be dominated by the largest task cluster (i.e., sentiment analysis) or languages (e.g., languages from Indo-European language family).

## Acknowledgements

We acknowledge support from Canada Research Chairs (CRC), the Natural Sciences and Engineering Research Council of Canada (NSERC; RGPIN-2018-04267), the Social Sciences and Humanities Research Council of Canada (SSHRC; 435-2018-0576; 895-2020-1004; 895-2021-1008), Canadian Foundation for Innovation (CFI; 37771), Digital Research Alliance of Canada,[9] and UBC ARC-Sockeye.[10]

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

# Appendices

## A Benchmark

Table 7 summarizes language distribution of datasets in SPARROW and taxonomy of these language according to Ethnologue (Gordon Jr, 2005) and Glottolog (Nordhoff and Hammarström, 2011). Tables 9, 10, 11, 12, 13, and 14 describe the datasets in tasks of antisocial language detection, emotion recognition, humor detection, irony and sarcasm detection, sentiment analysis, and subjectivity analysis, respectively.

We empirically characterize the issue of data inaccessibility by re-collecting tweets content via tweet IDs. Table 8 shows the data decay issue of 25 datasets.

## B Models

### B.1 Finetuning on Encoder-only LLMs

We evaluate the following Transformer-encoder-based multilingual PLMs on SPARROW. We fine-tune each PLMs on the full training set and update all the parameters of the model during the training.

**(1) Multilingual-BERT** (mBERT) (Devlin et al., 2019) is trained on a Wikipedia corpus including 104 languages with masked language modelling (MLM) and next sentence prediction objectives. It contains 110M parameters. mBERT tokenizes text by using WordPiece with a vocabulary size of 172K.

**(2) XLM-RoBERTa$_{Base}$** (XLM-R) (Conneau et al., 2020) is trained on CommonCrawl data involving 100 languages with MLM objective. It uses a SentencePiece tokenizer with a vocabulary size of 250K and contains 270M parameters.

**(3) Bernice** (DeLucia et al., 2022) is trained with 2.5B tweets in 66 languages and MLM objective. Bernice consists of 270M parameters and a tweet-specific SentencePiece tokenizer including a vocabulary size of 250K.

**(4) InfoDCL** (Zhang et al., 2023a) further trains XLM-R with 100M tweets in 66 languages with two contrastive learning, MLM, and distant label prediction objectives. InfoDCL shows that it effectively learns language representations for understanding SM.

### B.2 Zero-shot Setting on LLMs

We also investigate the zero-shot performance on a wide range of LLMs:

**(1) BLOOM** (Scao et al., 2022) is a Transformer decoder-only model trained on the ROOTS corpus consisting of 46 natural and 13 programming languages. BLOOM uses a multilingual vocabulary with 250K tokens and is trained with auto-regressive language modelling objectives.

**(2) Multilingual T5 (mT5)** (Xue et al., 2021) is Transformer encoder-decoder model trained on CommonCrawl data involving 101 languages and contains a vocabulary with 250K tokens. It trained with sequence-to-sequence MLM objective.

**(3) LLaMA** (Touvron et al., 2023) is a Transformer decoder-only model pretrained on 1.4T tokens where the majority are English and a small amount of data in 20 other languages. We utilize LLaMA with 7B parameters and a vocabulary with 30K tokens.

**(4) BLOOMZ** (Muennighoff et al., 2022) is also an instruction finetune model. It further finetunes BLOOM on xP3 corpus that contains 13 type of tasks in 46 languages with English prompt. We benchmark SPARROW on the BLOOM-based models with a size of 7.1B parameters.

**(5) BLOOMZ-P3** (Muennighoff et al., 2022) is an instruction finetuned model. It is initialized by BLOOM and further finetunes on English-only P3 corpus (Sanh et al., 2022) containing $2,073$ natural language prompts for eight types of NLP tasks.

**(6) BLOOM-Bactrian** (Li et al., 2023) tune BLOOM on a 3.4M instruction-following dataset in 52 languages with low-rank adaptation modules. Li et al. (2023) translate the English 67K instructions from Alpaca and Dolly datasets into 51 languages and utilize ChatGPT API to generate responses in the corresponding language.

**(7) mT0** (Muennighoff et al., 2022) is instruction fine-tuned mT5 model with xP3 corpus. We evaluate the mT5-based models with XL size (with 3.7B parameters).

**(8) Alpaca** (Taori et al., 2023) further tune LLaMA on a 52K instruction-following dataset that is generated by `gpt-3.5-turbo` of OpenAI API. The dataset includes diverse English instruction-following tasks, e.g., question answering and programming.

**(9) Vicuna** (Chiang et al., 2023) further tune LLaMA on 70K diverse user-shared conversations with ChatGPT in English.

**(10) ChatGPT** is a conversation-based LLM trained GTP-3 (Brown et al., 2020) through reinforcement learning with human feedback (Ouyang et al., 2022; Christiano et al., 2017). We exploit `gpt-3.5-turbo-0301` via OpenAI API.[11]

## C  Experiments

### C.1  Hyperparameters

To be computation friendly, we only tune the peak learning rate of each model in a set of $\{1e-4, 5e-5, 3e-5, 1e-5\}$ and randomly select 45 datasets for hyper-parameter tuning. We fine-tune a PLM with an arbitrary batch size of 32, sequence length of 128 tokens, and 20 epochs with patience of five epochs based on the model performance on Dev set. We fine-tune each dataset three time with different seeds and identify the best model based on Dev set performance. The best learning rate for each model is identified based on the average score of Dev set of the 45 datasets. The best peak learning rate is $3e-5$ for mBERT, XLM-T, and Bernice and $1e-5$ for other models.

### C.2  Prompts

The prompts we use in our experiments are summarized in Table 15.

### C.3  Results

Table 16 shows aggregated performance of fine-tuned models on Dev and Test-S. We report the average of dataset-specific metrics and standard deviation in a task and a category. We also report the Test-S performance of tasks of antisocial language detection, emotion recognition, humor detection, irony and sarcasm detection, sentiment analysis, and subjectivity analysis in Tables 17, 18, 19, 20, 21, and 22, respectively.

We provide a concise study to probe the sensitivity of open-source LLMs to prompts and present the results in Table 24.

---

[11] https://openai.com/

| Lang. Family | Lang. | Code | # dataset | Script |
|---|---|---|---|---|
| Afro-Asiatic | Amharic | amh | 1 | Ethiopic |
| | Arabic | ara | 15 | Arabic |
| | Darija | ary | 1 | Arabic |
| | Dziria | arq | 1 | Arabic |
| | Hausa | hau | 1 | Latin |
| | Hebrew | heb | 1 | Hebrew |
| | Maltese | mlt | 1 | Latin |
| Atlantic-Congo | Bambara | bam | 1 | Latin |
| | Igbo | ibo | 1 | Latin |
| | Kinyarwanda | kin | 1 | Latin |
| | Swahili | swh | 1 | Latin |
| | Twi | twi | 1 | Latin |
| | Tsonga | tso | 1 | Latin |
| | Yoruba | yor | 2 | Latin |
| Austroasiatic | Vietnamese | vie | 1 | Latin |
| Austronesian | Acehnese | ace | 1 | Latin |
| | Balinese | ban | 1 | Latin |
| | Banjarese | bjn | 1 | Latin |
| | Buginese | bug | 1 | Latin |
| | Filipino | fil | 1 | Latin |
| | Indonesian | ind | 3 | Latin |
| | Javanese | jav | 1 | Latin |
| | Madurese | mad | 1 | Latin |
| | Minangkabau | min | 1 | Latin |
| | Ngaju | nij | 1 | Latin |
| | Sundanese | sun | 1 | Latin |
| | Toba batak | bbc | 1 | Latin |
| Dravidian | Kannada | kan | 2 | Kannada, Latin |
| | Malayalam | mal | 2 | Malayalam, Latin |
| | Tamil | tam | 2 | Tamil, Latin |
| | Telugu | tel | 2 | Telugu |
| Indo-European | Albanian | sqi | 1 | Latin |
| | Bosnian | bos | 1 | Latin |
| | Bulgarian | bul | 1 | Cyrillic |
| | Bengali | ben | 4 | Bengali, Latin |
| | Croatian | hrv | 1 | Latin |
| | Czech | ces | 2 | Latin |
| | Danish | dan | 1 | Latin |
| | English | eng | 27 | Latin |
| | French | fra | 6 | Latin |
| | German | deu | 3 | Latin |
| | Greek | ell | 1 | Greek |
| | Hindi | hin | 5 | Devanagari, Latin |
| | Italian | ita | 11 | Latin |
| | Marathi | mar | 1 | Devanagari |
| | Nigerian Pidgin | pcm | 2 | Latin |
| | Norwegian | nor | 1 | Latin |
| | Persian | fas | 3 | Arabic |
| | Portuguese | por | 4 | Latin |
| | Polish | pol | 4 | Latin |
| | Romanian | ron | 2 | Latin |
| | Russian | rus | 3 | Cyrillic |
| | Spanish | spa | 9 | Latin |
| | Serbian | srp | 1 | Cyrillic |
| | Slovak | slk | 1 | Latin |
| | Slovenian | slv | 2 | Latin |
| | Swedish | swe | 1 | Latin |
| Japonic | Japanese | jpn | 1 | Han, Hir., Kat. |
| Koreanic | Korean | kor | 5 | Hangul |
| Sino-Tibetan | Chinese | zho | 6 | Han |
| Tai-Kadai | Thai | tha | 1 | Thai |
| Turkic | Turkish | tur | 2 | Latin |
| Uralic | Finnish | fin | 3 | Latin |
| | Hungarian | hun | 1 | Latin |

Table 7: Summary of languages covered in SPARROW. **Lang.:** Language. Language code is marked by ISO 639-3 code. Language information is retrieved from Ethnologue (Gordon Jr, 2005) and Glottolog (Nordhoff and Hammarström, 2011). The column **# dataset** shows the number of datasets covered by SPARROW per language. **Hir.:** Hiragana, **Kat.:** Katakana

| Dataset | Study | Year | Original | Retrieval | Decay % |
|---|---|---|---|---|---|
| Sarc-eng$_{Ril}$ | Riloff et al. (2013) | 2013 | 3K | 1K | 0.41 |
| Sarc-ces$_{Pta}$ | Ptáček et al. (2014) | 2013 | 7K | 4K | 0.29 |
| Sarc-eng$_{Pta}$ | Ptáček et al. (2014) | 2013 | 100K | 89K | 0.11 |
| Sarc-eng$_{Bam}$ | Bamman and Smith (2015) | 2015 | 19K | 14K | 0.24 |
| Sent-bul$_{Moz}$ | Mozetič et al. (2016) | 2016 | 67K | 27K | 0.59 |
| Sent-bos$_{Moz}$ | Mozetič et al. (2016) | 2016 | 44K | 20K | 0.54 |
| Sent-deu$_{Moz}$ | Mozetič et al. (2016) | 2016 | 109K | 52K | 0.52 |
| Sent-eng$_{Moz}$ | Mozetič et al. (2016) | 2016 | 103K | 43K | 0.58 |
| Sent-spa$_{Moz}$ | Mozetič et al. (2016) | 2016 | 275K | 153K | 0.44 |
| Sent-hrv$_{Moz}$ | Mozetič et al. (2016) | 2016 | 97K | 66K | 0.32 |
| Sent-hun$_{Moz}$ | Mozetič et al. (2016) | 2016 | 109K | 40K | 0.63 |
| Sent-pol$_{Moz}$ | Mozetič et al. (2016) | 2016 | 223K | 109K | 0.51 |
| Sent-por$_{Moz}$ | Mozetič et al. (2016) | 2016 | 157K | 49K | 0.69 |
| Sent-rus$_{Moz}$ | Mozetič et al. (2016) | 2016 | 107K | 41K | 0.62 |
| Sent-slk$_{Moz}$ | Mozetič et al. (2016) | 2016 | 70K | 38K | 0.46 |
| Sent-slv$_{Moz}$ | Mozetič et al. (2016) | 2016 | 133K | 74K | 0.44 |
| Sent-sqi$_{Moz}$ | Mozetič et al. (2016) | 2016 | 53K | 36K | 0.31 |
| Sent-srp$_{Moz}$ | Mozetič et al. (2016) | 2016 | 73K | 27K | 0.63 |
| Sent-swe$_{Moz}$ | Mozetič et al. (2016) | 2016 | 58K | 34K | 0.42 |
| Hate-eng$_{Was}$ | Waseem and Hovy (2016) | 2016 | 16K | 10K | 0.36 |
| Sent-por$_{Bru}$ | Brum (2018) | 2017 | 157K | 56K | 0.64 |
| Sent-eng$_{Ros}$ | Rosenthal et al. (2017) | 2017 | 50K | 42K | 0.15 |
| Iron-hin$_{Vij}$ | Vijay et al. (2018) | 2018 | 3K | 2K | 0.10 |
| Sexi-fre$_{Chi}$ | Chiril et al. (2020) | 2018 | 12K | 9K | 0.22 |
| Humo-hin$_{Agg}$ | Aggarwal et al. (2020) | 2018 | 7K | 5K | 0.30 |

Table 8: Data decay issue in social media data. These 25 datasets are distribute by tweet IDs. We retrieve these tweet on Nov. 2020 - Jan. 2022 and find that 42% samples are inaccessible.

| Dataset | Study | Lang. | Source / Domain | Year | #Lb | Labels | Data Slipt | Metric |
|---|---|---|---|---|---|---|---|---|
| Aggr-hin$_{Kum}$ | Kumar et al. (2018) | hin | Twitter, Facebook | 2018 | 2 | {Aggressive, Not} | 9,306/1,163/1,164 | M-F1 |
| Dang-ara$_{Als}$ | Alshehri et al. (2020) | ara | Twitter | 2020 | 2 | {Dangerous, Not} | 3,474/615/663 | M-F1 |
| Hate-eng$_{Was}$ | Waseem and Hovy (2016) | eng | Twitter | 2016 | 3 | {Not, Racism, Sexism} | 8,683/1,086/1,085 | W-F1 |
| Hate-eng$_{Dav}$ | Davidson et al. (2017) | eng | Twitter | 2017 | 3 | {Hate, Not, Offensive} | 19,826/2,478/2,479 | W-F1 |
| Hate-ara$_{Ala}$ | Alakrot et al. (2018) | ara | YouTube comment | 2017 | 2 | {Hate, Not} | 9,014/1,127/1,127 | M-F1 |
| Hate-ita$_{Bos}$ $^{\star}$ | Bosco et al. (2018) | ita | Twitter | 2018 | 2 | {Hate, Not} | 2,700/300/1,000 | M-F1 |
| Hate-fil$_{Cab}$ | Cabasag et al. (2019) | fil | Twitter | 2016 | 2 | {Hate, Not} | 10,000/4,232/4,232 | M-F1 |
| Hate-ara$_{Mul}$ | Mulki et al. (2019) | ara | Twitter | 2019 | 3 | {Abusive, Hate, Not} | 4,208/468/1,170 | M-F1 |
| Hate-eng$_{Bas}$ | Basile et al. (2019) | eng | Twitter | 2019 | 2 | {Hate, Not} | 9,000/1,000/3,000 | M-F1 |
| Hate-spa$_{Bas}$ | Basile et al. (2019) | spa | Twitter | 2019 | 2 | {Hate, Not} | 4,500/500/1,600 | M-F1 |
| Hate-por$_{For}$ | Fortuna et al. (2019) | por | Twitter | 2019 | 2 | {Hate, Not} | 4,536/567/567 | M-F1 |
| Hate-pol$_{Pta}$ | Ptaszynski et al. (2019) | pol | Twitter | 2019 | 2 | {Hate, Not} | 9,037/1,004/1,000 | M-F1 |
| Hate-kor$_{Moo}$ | Moon et al. (2020) | kor | News comment | 2020 | 3 | {Hate, Not, Offensive} | 7,106/790/471 | M-F1 |
| Hate-ara$_{Mub}$ | Mubarak et al. (2020) | ara | Twitter | 2020 | 2 | {Hate, Not} | 6,839/1,000/2,000 | M-F1 |
| Hate-zho$_{Den}$ | Deng et al. (2022) | zho | Weibo | 2022 | 2 | {Hate, Not} | 25,726/6,431/5,323 | M-F1 |
| Hate-kor$_{Jeo}$ | Jeong et al. (2022) | kor | News and YouTube comment | 2022 | 2 | {Hate, Not} | 32,343/4,043/4,043 | M-F1 |
| Hate-tel$_{Mar}$ | Marreddy et al. (2022) | tel | Misc | 2022 | 2 | {Hate, Not} | 24,599/3,510/7,033 | M-F1 |
| Sexi-fra$_{Chi}$ | Chiril et al. (2020) | fra | Twitter | 2018 | 2 | {Not, Sexism} | 7,670/959/959 | M-F1 |
| Offe-eng$_{Zam}$ | Zampieri et al. (2019) | eng | Twitter | 2019 | 2 | {Not, Offensive} | 11,916/1,324/860 | M-F1 |
| Offe-ara$_{Zam}$ | Zampieri et al. (2020) | ara | Twitter | 2019 | 2 | {Not, Offensive} | 7,055/784/1,827 | M-F1 |
| Offe-dan$_{Zam}$ | Zampieri et al. (2020) | dan | Misc | 2019 | 2 | {Not, Offensive} | 2,664/296/329 | M-F1 |
| Offe-ell$_{Zam}$ | Zampieri et al. (2019) | ell | Twitter | 2019 | 2 | {Not, Offensive} | 7,869/874/1,544 | M-F1 |
| Offe-tur$_{Zam}$ | Zampieri et al. (2020) | tur | Twitter | 2019 | 2 | {Not, Offensive} | 28,149/3,128/3,515 | M-F1 |
| Offe-ara$_{Mub}$ | Mubarak et al. (2020) | ara | Twitter | 2020 | 2 | {Not, Offensive} | 6,839/1,000/2,000 | M-F1 |
| Offe-slv$_{Nov}$ | Kralj Novak et al. (2021) | slv | Twitter | 2020 | 4 | {Appropriate, Inappropriate, Not, Offensive} | 65,021/8,127/8,128 | M-F1 |
| Offe-G-eng$_{Zam}$ | Zampieri et al. (2019) | eng | Twitter | 2019 | 3 | {Group, Individual, Others} | 3,485/391/213 | M-F1 |
| Hate-G-ara$_{Ous}$ | Ousidhoum et al. (2019) | ara | Twitter | 2019 | 13 | {African_descent, Arabs, Asians, Christian, Gay, Immigrants, Indian/hindu, Individual, Jews, Muslims, Others, Refugees, Women} | 2,682/334/335 | M-F1 |
| Hate-G-fra$_{Ous}$ | Ousidhoum et al. (2019) | fra | Twitter | 2019 | 16 | {African_descent, Arabs, Asians, Christian, Gay, Gispanics, Immigrants, Indian/hindu, Individual, Jews, Left_wing_people, Muslims, Others, Refugees, Special_needs, Women} | 3,211/401/402 | M-F1 |
| Offe-T-eng$_{Zam}$ | Zampieri et al. (2019) | eng | Twitter | 2019 | 2 | {Targeted, Untargeted} | 3,963/437/240 | M-F1 |
| Hate-T-ara$_{Ous}$ | Ousidhoum et al. (2019) | ara | Twitter | 2019 | 4 | {Gender, Origin, Others, Religion} | 2,682/334/336 | M-F1 |
| Hate-T-fra$_{Ous}$ | Ousidhoum et al. (2019) | fra | Twitter | 2019 | 6 | {Disability, Gender, Origin, Others, Religion, Sexual_Orientation} | 3,211/401/402 | M-F1 |
| Hate-T-ben$_{Kar}$ | Karim et al. (2021) | ben | Misc | 2020 | 4 | {Geopolitical, Personal, Political, Religion} | 4,558/570/570 | M-F1 |
| Offe-T-kan$_{Cha}$ | Chakravarthi et al. (2022) | kan | YouTube comment | 2019 | 5 | {Group, Individual, Not, Others, Untargeted} | 4,694/586/593 | M-F1 |
| Offe-T-mal$_{Cha}$ | Chakravarthi et al. (2022) | mal | YouTube comment | 2019 | 4 | {Group, Individual, Not, Untargeted} | 14,723/1,836/1,844 | M-F1 |
| Offe-T-tam$_{Cha}$ | Chakravarthi et al. (2022) | tam | YouTube comment | 2019 | 5 | {Group, Individual, Not, Others, Untargeted} | 33,685/4,216/4,232 | M-F1 |
| Hate-T-kor$_{Jeo}$ | Jeong et al. (2022) | kor | News and YouTube comment | 2022 | 4 | {Group, Individual, Other, Untargeted} | 16,239/2,049/2,022 | M-F1 |

Table 9: Description of 36 antisocial language detection datasets. **Lang.:** Language is marked by ISO 639-3, **#Lb:** the label size of a dataset. **M-F1:** Macro-F1, **W-F1:** Weighted-F1. $^{\star}$ indicates that data sharing needs approval from the original authors.

| Dataset | Study | Lang. | Source / Domain | Year | #Lb | Labels | Data Slipt | Metric |
|---|---|---|---|---|---|---|---|---|
| Emot-eng$_{Wal}$ | Wallbott and Scherer (1986) | eng | Questionnaire | 1986 | 7 | {Anger, Disgust, Fear, Guilt, Joy, Sadness, Shame} | 6,132/767/767 | M-F1 |
| Emot-zho$_{Lee}$ | Lee and Wang (2015) | zho | Weibo | 2015 | 5 | {Anger, Fear, Happy, Sadness, Surprise} | 3,122/347/418 | Accuracy |
| Emot-fin$_{Kaj}$ | Kajava (2018) | fin | Subtitle | 2016 | 8 | {Anger, Anticipation, Disgust, Fear, Joy, Sadness, Surprise, Trust} | 5,197/577/653 | M-F1 |
| Emot-fra$_{Kaj}$ | Kajava (2018) | fra | Subtitle | 2016 | 8 | {Anger, Anticipation, Disgust, Fear, Joy, Sadness, Surprise, Trust} | 5,198/577/653 | M-F1 |
| Emot-ita$_{Kaj}$ | Kajava (2018) | ita | Subtitle | 2016 | 8 | {Anger, Anticipation, Disgust, Fear, Joy, Sadness, Surprise, Trust} | 5,197/577/653 | M-F1 |
| Emot-ara$_{Abd}$ | Abdul-Mageed et al. (2020) | ara | Twitter | 2016 | 8 | {Anger, Anticipation, Disgust, Fear, Joy, Sadness, Surprise, Trust} | 50,000/910/941 | M-F1 |
| Emot-eng$_{Moh}$ | Mohammad et al. (2018) | eng | Twitter | 2018 | 4 | {Anger, Joy, Optimism, Sadness} | 3,257/374/1,421 | M-F1 |
| Emot-ara$_{Moh}$ | Mohammad et al. (2018) | ara | Twitter | 2018 | 4 | {Anger, Joy, Fear, Sadness} | 2,284/490/1,188 | M-F1 |
| Emot-spa$_{Moh}$ | Mohammad et al. (2018) | spa | Twitter | 2018 | 4 | {Anger, Joy, Fear, Sadness} | 2,708/479/1,696 | M-F1 |
| Emot-ind$_{Sap}$ | Saputri et al. (2018) | ind | Twitter | 2019 | 5 | {Anger, Fear, Happy, Love, Sadness} | 3,520/440/441 | M-F1 |
| Emot-tur$_{Guv}$ | Güven et al. (2020) | tur | Twitter | 2020 | 5 | {Anger, Fear, Happy, Sadness, Surprise} | 3,200/400/400 | Accuracy |
| Emot-ind$_{Wil}$ | Wilie et al. (2020) | ind | Twitter | 2018 | 5 | {Anger, Fear, Happy, Love, Sadness} | 3,169/352/440 | M-F1 |
| Emot-vie$_{Ho}$ | Ho et al. (2019) | vie | Facebook | 2019 | 7 | {Anger, Disgust, Fear, Joy, Others, Sadness, Surprise} | 5,548/686/693 | W-F1 |
| Emot-eng$_{Pla}$ | Plaza del Arco et al. (2020) | eng | Twitter | 2020 | 7 | {Anger, Disgust, Fear, Joy, Others, Sadness, Surprise} | 5,842/730/731 | M-F1 |
| Emot-spa$_{Pla}$ | Plaza del Arco et al. (2020) | spa | Twitter | 2020 | 7 | {Anger, Disgust, Fear, Joy, Others, Sadness, Surprise} | 6,727/841/841 | M-F1 |
| Emot-fin$_{Ohm}$ | Öhman et al. (2020) | fin | Subtitle | 2020 | 8 | {Anger, Anticipation, Disgust, Fear, Joy, Sadness, Surprise, Trust} | 8,864/1,118/1,086 | M-F1 |
| Emot-eng$_{Dem}$ | Demszky et al. (2020) | eng | Reddit | 2020 | 27 | {Admiration, Amusement, Anger, Annoyance, Approval, Caring, Confusion, Curiosity, Desire, Disappointment, Disapproval, Disgust, Embarrassment, Excitement, Fear, Gratitude, Grief, Joy, Love, Nervousness, Optimism, Pride, Realization, Relief, Remorse, Sadness5, Surprise6} | 23,485/2,956/2,984 | M-F1 |
| Emot-ita$_{Bia}$ | Bianchi et al. (2021) | ita | Twitter | 2021 | 4 | {Anger, Fear, Joy, Sadness} | 1,629/204/204 | M-F1 |
| Emot-ron$_{Cio}$ | Ciobotaru and Dinu (2021) | ron | Twitter | 2020 | 4 | {Anger, Fear, Joy, Sadness} | 2,600/318/324 | M-F1 |
| Emot-hin$_{Deb}$ | Shome (2021) | hin | Machine Translation | 2021 | 27 | {Admiration, Amusement, Anger, Annoyance, Approval, Caring, Confusion, Curiosity, Desire, Disappointment, Disapproval, Disgust, Embarrassment, Excitement, Fear, Gratitude, Grief, Joy, Love, Nervousness, Optimism, Pride, Realization, Relief, Remorse, Sadness, Surprise} | 23,485/2,956/2,984 | M-F1 |
| Emot-por$_{Cor}$ | Cortiz et al. (2021) | por | Twitter | 2021 | 28 | {Admiration, Amusement, Anger, Annoyance, Approval, Compassion, Confusion, Curiosity, Desire, Disappointment, Disapproval, Disgust, Embarrassment, Envy, Excitement, Fear, Gratitude, Grief, Joy, Longing, Love, Nervousness, Optimism, Pride, Relief, Remorse, Sadness, Surprise} | 24,919/2,769/12,966 | M-F1 |
| Emot-fas$_{Sab}$ | Sabri et al. (2021) | fas | Twitter | 2021 | 6 | {Anger, Fear, Happy, Hatred, Sadness, Wonder} | 4,180/523/523 | M-F1 |
| Emot-rus$_{Sbo}$ | Sboev et al. (2020) | rus | Misc | 2021 | 5 | {Anger, Fear, Joy, Sadness, Surprise} | 3,951/427/1,128 | M-F1 |
| Emot-ben$_{Iqb}$ | Iqbal et al. (2022) | ben | Misc | 2022 | 6 | {Anger, Disgust, Fear, Joy, Sadness, Surprise} | 5,600/700/700 | M-F1 |
| Emot-fra$_{Bia}$ ⋆ | Bianchi et al. (2022) | fra | Machine Translation | 2018 | 4 | {Anger, Fear, Joy, Sadness} | 3,798/476/476 | M-F1 |
| Emot-deu$_{Bia}$ ⋆ | Bianchi et al. (2022) | deu | Machine Translation | 2018 | 4 | {Anger, Fear, Joy, Sadness} | 3,798/476/476 | M-F1 |

Table 10: Description of 26 emotion recognition datasets. **Lang.:** Language is marked by ISO 639-3, **#Lb:** the label size of a dataset. **M-F1:** Macro-F1, **W-F1:** Weighted-F1.

| Dataset | Study | Lang | Source / Domain | Year | #Lb | Labels | Data Slipt | Metric |
|---|---|---|---|---|---|---|---|---|
| Humo-hin$_{Agg}$ | Aggarwal et al. (2020) | hin | Twitter | 2018 | 2 | {Humor, Not} | 4,187/524/523 | Accuracy |
| Humo-rus$_{Bli}$ | Blinov et al. (2019) | rus | Misc | 2018 | 2 | {Humor, Not} | 251,416/61,794/1,877 | M-F1 |
| Humo-spa$_{Chi}$ | Chiruzzo et al. (2021) | spa | Twitter | 2019 | 2 | {Humor, Not} | 24,000/6,000/6,000 | M-F1 |
| Humo-eng$_{Mea}$ | Meaney et al. (2021) | eng | Twitter | 2021 | 2 | {Humor, Not} | 8,000/1,000/1,000 | M-F1 |

Table 11: Description of four humor detection datasets. **Lang:** Language is marked by ISO 639-3, **#Lb:** the label size of a dataset. **M-F1:** Macro-F1.

| Dataset | Study | Lang. | Source / Domain | Year | #Lb | Labels | Data Slipt | Metric |
|---|---|---|---|---|---|---|---|---|
| Iron-ita$_{Bas}$ | Basile et al. (2014) | ita | Twitter | 2014 | 2 | {Irony, Not} | 4,062/453/1,936 | M-F1 |
| Iron-spa$_{Bar}$ | Barbieri et al. (2016) | spa | Twitter | 2014 | 2 | {Irony, Not} | 6,669/741/1,997 | M-F1 |
| Iron-eng$_{Hee}$ | Van Hee et al. (2018) | eng | Twitter | 2018 | 2 | {Irony, Not} | 3,450/384/784 | F1-irony |
| Iron-ita$_{Cig}$ | Cignarella et al. (2018) | ita | Twitter | 2018 | 2 | {Irony, Not} | 3,579/398/872 | M-F1 |
| Iron-hin$_{Vij}$ | Vijay et al. (2018) | hin | Twitter | 2018 | 2 | {Irony, Not} | 2,217/277/277 | M-F1 |
| Iron-ara$_{Gha}$ | Ghanem et al. (2019) | ara | Twitter | 2019 | 2 | {Irony, Not} | 3,622/402/1,006 | M-F1 |
| Iron-spa$_{Ort}$ | Ortega-Bueno et al. (2019) | spa | Twitter | 2019 | 2 | {Irony, Not} | 2,160/240/600 | M-F1 |
| Iron-fas$_{Gol}$ * | Golazizian et al. (2020) | fas | Twitter | 2019 | 2 | {Irony, Not} | 2,352/295/294 | Accuracy |
| Iron-zho$_{Xia}$ * | Xiang et al. (2020) | zho | Weibo | 2020 | 5 | {Insufficient_Evidence, Irony, Not, Unlikely_Ironic, Weakly_Irony} | 7,014/876/876 | M-F1 |
| Sarc-eng$_{Wal}$ | Walker et al. (2012) | eng | Debate Forum | 2012 | 2 | {Not, Sarcasm} | 900/100/995 | M-F1 |
| Sarc-eng$_{Ril}$ | Riloff et al. (2013) | eng | Twitter | 2013 | 2 | {Not, Sarcasm} | 1,413/177/177 | F1-sarcasm |
| Sarc-ces$_{Pta}$ | Ptáček et al. (2014) | ces | Twitter | 2013 | 2 | {Not, Sarcasm} | 3,977/497/497 | M-F1 |
| Sarc-eng$_{Pta}$ | Ptáček et al. (2014) | eng | Twitter | 2013 | 2 | {Not, Sarcasm} | 71,433/8,929/8,930 | M-F1 |
| Sarc-eng$_{Bam}$ | Bamman and Smith (2015) | eng | Twitter | 2015 | 2 | {Not, Sarcasm} | 11,864/1,483/1,484 | Accuracy |
| Sarc-eng$_{Raj}$ | Rajadesingan et al. (2015) | eng | Twitter | 2015 | 2 | {Not, Sarcasm} | 41,261/5,158/5,158 | Accuracy |
| Sarc-eng$_{Ora}$ | Oraby et al. (2016) | eng | Debate Forum | 2016 | 2 | {Not, Sarcasm} | 900/100/2,260 | M-F1 |
| Sarc-zho$_{Gon}$ * | Gong et al. (2020) | zho | News comment | 2019 | 2 | {Not, Sarcasm} | 3,978/497/497 | M-F1 |
| Sarc-ara$_{Abu}$ | Abu Farha and Magdy (2020) | ara | Twitter | 2020 | 2 | {Not, Sarcasm} | 7,593/844/2,110 | M-F1 |
| Sarc-ara$_{Far}$ | Farha et al. (2021) | ara | Twitter | 2020 | 2 | {Not, Sarcasm} | 11,293/1,255/3,000 | M-F1 |
| Iron-T-eng$_{Hee}$ | Van Hee et al. (2018) | eng | Twitter | 2018 | 4 | {Ironic_by_clash, Not, Other_irony, Situational_irony} | 3,450/384/784 | M-F1 |

Table 12: Description of 20 irony and sarcasm detection datasets. **Lang.:** Language is marked by ISO 639-3, **#Lb:** the label size of a dataset. **M-F1:** Macro-F1. * indicates that data sharing needs an approval from the original authors.

| Dataset | Study | Lang. | Source / Domain | Year | #Lb | Labels | Data Slipt | Metric |
|---|---|---|---|---|---|---|---|---|
| Sent-eng$_{Pan}$ | Pang and Lee (2005) | eng | Moview review | 2005 | 2 | {Negative, Positive} | 8,529/1,066/1,067 | Accuracy |
| Sent-zho$_{Tan}$ | Tan and Zhang (2008) | zho | Misc | 2008 | 2 | {Negative, Positive} | 9,600/1,200/1,200 | M-F1 |
| Sent-T-eng$_{The}$ | Thelwall et al. (2012) | eng | Twitter | 2012 | 2 | {Negative, Positive} | 900/100/1,113 | Accuracy |
| Sent-Y-eng$_{The}$ | Thelwall et al. (2012) | eng | YouTube comment | 2012 | 2 | {Negative, Positive} | 900/100/1,142 | Accuracy |
| Sent-5-eng$_{Soc}$ | Socher et al. (2013) | eng | Moview review | 2013 | 5 | {Negative, Neutral, Positive, Very_Negative, Very_Positive} | 8,544/1,101/2,210 | Accuracy |
| Sent-kor$_{Jan}$ * | Jang et al. (2013) | kor | News article | 2013 | 4 | {Complex, Negative, Neutral, Positive} | 4,187/523/524 | M-F1 |
| Sent-eng$_{Soc}$ | Socher et al. (2013) | eng | Moview review | 2013 | 2 | {Negative, Positive} | 6,920/872/1,821 | Accuracy |
| Sent-ita$_{Bas}$ | Basile et al. (2014) | ita | Twitter | 2014 | 2 | {Negative, Positive} | 2,376/265/1,207 | M-F1 |
| Sent-ita$_{Bas}$ | Barbieri et al. (2016) | ita | Twitter | 2016 | 2 | {Negative, Positive} | 3,738/416/1,018 | M-F1 |
| Sent-mlt$_{Din}$ | Dingli and Sant (2016) | mlt | Moview review | 2016 | 2 | {Negative, Positive} | 596/85/171 | M-F1 |
| Sent-bul$_{Moz}$ | Mozetič et al. (2016) | bul | Twitter | 2016 | 3 | {Negative, Neutral, Positive} | 22,184/2,773/2,773 | M-F1 |
| Sent-bos$_{Moz}$ | Mozetič et al. (2016) | bos | Twitter | 2016 | 3 | {Negative, Neutral, Positive} | 16,335/2,042/2,042 | M-F1 |
| Sent-deu$_{Moz}$ | Mozetič et al. (2016) | deu | Twitter | 2016 | 3 | {Negative, Neutral, Positive} | 42,010/5,251/5,252 | M-F1 |
| Sent-eng$_{Moz}$ | Mozetič et al. (2016) | eng | Twitter | 2016 | 3 | {Negative, Neutral, Positive} | 34,538/4,317/4,318 | M-F1 |
| Sent-spa$_{Moz}$ | Mozetič et al. (2016) | spa | Twitter | 2016 | 3 | {Negative, Neutral, Positive} | 122,410/15,301/15,302 | M-F1 |
| Sent-hrv$_{Moz}$ | Mozetič et al. (2016) | hrv | Twitter | 2016 | 3 | {Negative, Neutral, Positive} | 52,971/6,621/6,622 | M-F1 |
| Sent-hun$_{Moz}$ | Mozetič et al. (2016) | hun | Twitter | 2016 | 3 | {Negative, Neutral, Positive} | 32,717/4,089/4,090 | M-F1 |
| Sent-pol$_{Moz}$ | Mozetič et al. (2016) | pol | Twitter | 2016 | 3 | {Negative, Neutral, Positive} | 87,941/10,993/10,992 | M-F1 |
| Sent-por$_{Moz}$ | Mozetič et al. (2016) | por | Twitter | 2016 | 3 | {Negative, Neutral, Positive} | 39,525/4,941/4,940 | M-F1 |
| Sent-rus$_{Moz}$ | Mozetič et al. (2016) | rus | Twitter | 2016 | 3 | {Negative, Neutral, Positive} | 32,941/4,117/4,118 | M-F1 |
| Sent-slk$_{Moz}$ | Mozetič et al. (2016) | slk | Twitter | 2016 | 3 | {Negative, Neutral, Positive} | 30,694/3,837/3,837 | M-F1 |
| Sent-slv$_{Moz}$ | Mozetič et al. (2016) | slv | Twitter | 2016 | 3 | {Negative, Neutral, Positive} | 59,924/7,491/7,490 | M-F1 |
| Sent-sqi$_{Moz}$ | Mozetič et al. (2016) | sqi | Twitter | 2016 | 3 | {Negative, Neutral, Positive} | 29,375/3,672/3,672 | M-F1 |
| Sent-srp$_{Moz}$ | Mozetič et al. (2016) | srp | Twitter | 2016 | 3 | {Negative, Neutral, Positive} | 22,124/2,765/2,766 | M-F1 |
| Sent-swe$_{Moz}$ | Mozetič et al. (2016) | swe | Twitter | 2016 | 3 | {Negative, Neutral, Positive} | 27,277/3,409/3,410 | M-F1 |
| Sent-deu$_{Rei}$ | Rei et al. (2016) | deu | Twitter | 2016 | 3 | {Negative, Neutral, Positive} | 2,701/337/338 | M-F1 |
| Sent-spa$_{Rei}$ | Rei et al. (2016) | spa | Twitter | 2016 | 3 | {Negative, Neutral, Positive} | 6,099/763/762 | M-F1 |
| Sent-ita$_{Rei}$ | Rei et al. (2016) | ita | Twitter | 2016 | 3 | {Negative, Neutral, Positive} | 6,818/853/852 | M-F1 |
| Sent-eng$_{Ros}$ | Rosenthal et al. (2017) | eng | Twitter | 2017 | 3 | {Negative, Neutral, Positive} | 42,756/4,751/12,284 | M-Recall |
| Sent-ben$_{Pat}$ * | Patra et al. (2018) | ben | Twitter | 2015 | 3 | {Negative, Neutral, Positive} | 2,250/250/3,038 | M-F1 |
| Sent-hin$_{Pat}$ * | Patra et al. (2018) | hin | Twitter | 2015 | 3 | {Negative, Neutral, Positive} | 11,642/1,293/5,525 | M-F1 |
| Sent-heb$_{Amr}$ | Amram et al. (2018) | heb | Facebook | 2018 | 2 | {Negative, Positive} | 8,951/995/2,488 | Accuracy |
| Sent-por$_{Bru}$ | Brum and das Graças Volpe Nunes (2018) | por | Twitter | 2017 | 3 | {Negative, Neutral, Positive} | 45,127/5,585/5,637 | M-F1 |
| Sent-fin$_{Kaj}$ | Kajava (2018) | fin | Subtitle | 2016 | 2 | {Negative, Positive} | 5,197/577/653 | M-F1 |
| Sent-fra$_{Kaj}$ | Kajava (2018) | fra | Subtitle | 2016 | 2 | {Negative, Positive} | 5,198/577/653 | M-F1 |
| Sent-ita$_{Kaj}$ | Kajava (2018) | ita | Subtitle | 2016 | 2 | {Negative, Positive} | 5,197/577/653 | M-F1 |
| Sent-nor$_{Vel}$ | Velldal et al. (2018) | nor | Online review | 2018 | 6 | {Negative1, Negative2, Negative3, Positive4, Positive5, Positive6} | 34,903/4,360/4,351 | M-F1 |
| Sent-pol$_{Koc}$ | Kocoń et al. (2019) | pol | Customer review | 2019 | 4 | {Complex, Negative, Neutral, Positive} | 5,170/574/1,217 | M-F1 |
| Sent-tha$_{Sur}$ | Suriyawongkul et al. (2019) | tha | Facebook | 2019 | 3 | {Negative, Neutral, Positive} | 21,152/2,362/2,614 | M-F1 |
| Sent-zho$_{Wan}$ | Wan et al. (2020) | zho | Weibo | 2019 | 2 | {Negative, Positive} | 95,990/11,999/11,999 | M-F1 |
| Sent-fas$_{Ash}$ * | Ashrafi Asli et al. (2020) | fas | Customer review | 2020 | 3 | {Negative, Neutral, Positive} | 75,094/9,387/9,387 | M-F1 |
| Sent-ron$_{Dum}$ | Dumitrescu et al. (2020) | ron | Customer review | 2020 | 2 | {Negative, Positive} | 16,146/1,795/11,005 | M-F1 |
| Sent-pcm$_{Oye}$ | Oyewusi et al. (2020) | pcm | Twitter | 2020 | 3 | {Negative, Neutral, Positive} | 11,200/1,400/1,400 | M-F1 |
| Sent-pol$_{Ryb}$ | Rybak et al. (2020) | pol | Customer review | 2020 | 5 | {Negative, Neutral, Positive, Very_Negative, Very_Positive} | 8,619/958/1,002 | M-F1 |
| Sent-ind$_{Wil}$ | Wilie et al. (2020) | ind | Misc | 2019 | 3 | {Negative, Neutral, Positive} | 9,900/1,100/1,260 | M-F1 |
| Sent-ara$_{Abd}$ | Abdul-Mageed et al. (2021) | ara | Twitter | 2021 | 3 | {Negative, Neutral, Positive} | 49,301/4,443/4,933 | M-F1 |
| Sent-bam$_{Dia}$ | Diallo et al. (2021) | bam | Misc | 2021 | 3 | {Negative, Neutral, Positive} | 2,436/305/305 | M-F1 |
| Sent-ben$_{Isl}$ | Islam et al. (2021) | ben | Twitter | 2021 | 3 | {Negative, Neutral, Positive} | 12,575/1,567/1,586 | M-F1 |
| Sent-mar$_{Kul}$ | Kulkarni et al. (2021) | mar | Twitter | 2020 | 3 | {Negative, Neutral, Positive} | 12,114/1,500/2,250 | Accuracy |
| Sent-kan$_{Cha}$ | Chakravarthi et al. (2022) | kan | YouTube comment | 2019 | 2 | {Negative, Positive} | 3,995/505/502 | M-F1 |
| Sent-mal$_{Cha}$ | Chakravarthi et al. (2022) | mal | YouTube comment | 2019 | 2 | {Negative, Positive} | 8,410/1,044/1,039 | M-F1 |
| Sent-tam$_{Cha}$ | Chakravarthi et al. (2022) | tam | YouTube comment | 2019 | 2 | {Negative, Positive} | 24,063/2,966/3,047 | M-F1 |
| Sent-ara$_{Muh}$ | Abdul-Mageed et al. (2022) | ara | Twitter | 2021 | 3 | {Negative, Neutral, Positive} | 1,500/500/3,000 | Accuracy |
| Sent-amh$_{Muh}$ | Yimam et al. (2020) | amh | Twitter | 2020 | 3 | {Negative, Neutral, Positive} | 5,984/1,497/1,999 | W-F1 |
| Sent-ary$_{Muh}$ | Muhammad et al. (2023b) | ary | Twitter | 2021 | 3 | {Negative, Neutral, Positive} | 5,583/494/2,961 | W-F1 |
| Sent-arq$_{Muh}$ | Muhammad et al. (2023b) | arq | Twitter | 2021 | 3 | {Negative, Neutral, Positive} | 1,651/414/958 | W-F1 |
| Sent-hau$_{Muh}$ | Muhammad et al. (2022) | hau | Twitter | 2021 | 3 | {Negative, Neutral, Positive} | 14,172/2,677/5,303 | W-F1 |
| Sent-ibo$_{Muh}$ | Muhammad et al. (2022) | ibo | Twitter | 2021 | 3 | {Negative, Neutral, Positive} | 10,192/1,841/3,682 | W-F1 |
| Sent-pcm$_{Muh}$ | Muhammad et al. (2022) | pcm | Twitter | 2021 | 3 | {Negative, Neutral, Positive} | 5,121/1,281/4,154 | W-F1 |
| Sent-kin$_{Muh}$ | Muhammad et al. (2022) | kin | Twitter | 2021 | 3 | {Negative, Neutral, Positive} | 3,302/827/1,026 | W-F1 |
| Sent-swh$_{Muh}$ | Muhammad et al. (2022) | swh | Twitter | 2021 | 3 | {Negative, Neutral, Positive} | 1,810/453/748 | W-F1 |
| Sent-tso$_{Muh}$ | Muhammad et al. (2022) | tso | Twitter | 2021 | 3 | {Negative, Neutral, Positive} | 804/203/254 | W-F1 |
| Sent-twi$_{Muh}$ | Muhammad et al. (2022) | twi | Twitter | 2021 | 3 | {Negative, Neutral, Positive} | 3,481/388/949 | W-F1 |
| Sent-yor$_{Muh}$ | Muhammad et al. (2022) | yor | Twitter | 2021 | 3 | {Negative, Neutral, Positive} | 8,522/2,090/4,515 | W-F1 |
| Sent-yor$_{Sho}$ | Shode et al. (2022) | yor | Misc | 2021 | 2 | {Negative, Positive} | 800/200/500 | M-F1 |
| Sent-jpn$_{Suz}$ | Suzuki et al. (2022) | jpn | SNS post | 2021 | 5 | {Negative, Neutral, Positive, Very_Negative, Very_Positive} | 30,000/2,500/2,500 | Accuracy |
| Sent-ace$_{Win}$ | Winata et al. (2022) | ace | Trans. of online com. | 2022 | 3 | {Negative, Neutral, Positive} | 500/100/400 | M-F1 |
| Sent-ban$_{Win}$ | Winata et al. (2022) | ban | Trans. of online com. | 2022 | 3 | {Negative, Neutral, Positive} | 500/100/400 | M-F1 |
| Sent-bbc$_{Win}$ | Winata et al. (2022) | bbc | Trans. of online com. | 2022 | 3 | {Negative, Neutral, Positive} | 500/100/400 | M-F1 |
| Sent-bjn$_{Win}$ | Winata et al. (2022) | bjn | Trans. of online com. | 2022 | 3 | {Negative, Neutral, Positive} | 500/100/400 | M-F1 |
| Sent-bug$_{Win}$ | Winata et al. (2022) | bug | Trans. of online com. | 2022 | 3 | {Negative, Neutral, Positive} | 500/100/400 | M-F1 |
| Sent-jav$_{Win}$ | Winata et al. (2022) | jav | Trans. of online com. | 2022 | 3 | {Negative, Neutral, Positive} | 500/100/400 | M-F1 |
| Sent-mad$_{Win}$ | Winata et al. (2022) | mad | Trans. of online com. | 2022 | 3 | {Negative, Neutral, Positive} | 500/100/400 | M-F1 |
| Sent-min$_{Win}$ | Winata et al. (2022) | min | Trans. of online com. | 2022 | 3 | {Negative, Neutral, Positive} | 500/100/400 | M-F1 |
| Sent-nij$_{Win}$ | Winata et al. (2022) | nij | Trans. of online com. | 2022 | 3 | {Negative, Neutral, Positive} | 500/100/400 | M-F1 |
| Sent-sun$_{Win}$ | Winata et al. (2022) | sun | Trans. of online com. | 2022 | 3 | {Negative, Neutral, Positive} | 500/100/400 | M-F1 |
| Sent-tel$_{Mar}$ | Marreddy et al. (2022) | tel | Misc | 2022 | 3 | {Negative, Neutral, Positive} | 24,599/3,510/7,033 | M-F1 |

Table 13: Description of 77 sentiment analysis datasets. **Lang.:** Language is marked by ISO 639-3, **#Lb:** the label size of a dataset. **M-F1:** Macro-F1, **M-Recall:** Macro-Recall, **Trans. of online com.:** Trans. of online com. * indicates that data sharing needs approval from the original authors.

| Dataset | Study | Lang. | Source / Domain | Year | #Lb | Labels | Data Slipt | Metric |
|---|---|---|---|---|---|---|---|---|
| Subj-eng$_{Pan}$ | Pang and Lee (2004) | eng | Moview review | 2004 | 2 | {Objective, Subjective} | 8,100/900/1,000 | Accuracy |
| Subj-kor$_{Jan}$ $^\star$ | Jang et al. (2013) | kor | News article | 2013 | 7 | {Agreement, Argument, Emotion, Intention, Judgment, Others, Speculation} | 4,284/535/536 | M-F1 |
| Subj-ita$_{Bas}$ | Basile et al. (2014) | ita | Twitter | 2014 | 2 | {Objective, Subjective} | 4,061/452/1,935 | M-F1 |
| Subj-ita$_{Bas}$ | Barbieri et al. (2016) | ita | Twitter | 2016 | 2 | {Objective, Subjective} | 6,669/741/1,943 | M-F1 |
| Subj-spa$_{Bar}$ | Barbieri et al. (2016) | spa | Twitter | 2014 | 2 | {Objective, Subjective} | 6,669/741/1,998 | M-F1 |
| Subj-ces$_{Pri}$ | Pribán and Steinberger (2022) | ces | Moview review | 2021 | 2 | {Objective, Subjective} | 7,500/500/2,000 | Accuracy |

Table 14: Description of four subjectivity analysis datasets. **Lang.:** Language is marked by ISO 639-3, **#Lb:** the label size of a dataset. **M-F1:** Macro-F1. $^\star$ indicates that data sharing needs an approval from the original authors.

| | Dataset | Prompt |
|---|---|---|
| **Anti-social** | Dang-ara$_{Als}$ | {Content}
Question: Is the language of this sentence {labels}?
Answer: |
| | Aggr-hin$_{Kum}$, Hate-eng$_{Dav}$, Hate-eng$_{Was}$, Hate-ara$_{Ala}$, Hate-ita$_{Bos}$, Hate-fil$_{Cab}$ Hate-ara$_{Mul}$, Hate-eng$_{Bas}$, Hate-spa$_{Bas}$, Hate-por$_{For}$, Hate-pol$_{Pta}$, Hate-kor$_{Moo}$ Hate-ara$_{Mub}$, Hate-zho$_{Den}$, Hate-kor$_{Jeo}$, Hate-tel$_{Mar}$, Sexi-fra$_{Chi}$, Offe-eng$_{Zam}$ Offe-ara$_{Zam}$, Offe-dan$_{Zam}$, Offe-ell$_{Zam}$, Offe-tur$_{Zam}$, Offe-ara$_{Mub}$, Offe-slv$_{Nov}$ | {Content}
Question: Is the language of this text {labels}?
Answer: |
| | Offe-T-kan$_{Cha}$, Offe-G-eng$_{Zam}$, Hate-T-kor$_{Jeo}$ | {Content}
Question: Does this offensive text target {labels}?
Answers: |
| | Hate-G-ara$_{Ous}$, Hate-G-fra$_{Ous}$ | {Content}
Question: Does this hate speech target {labels}?
Answer: |
| | Offe-T-eng$_{Zam}$ | {Content}
Question: Is this offensive text {labels} insult?
Answer: |
| | Hate-T-ara$_{Ous}$, Hate-T-fra$_{Ous}$ | {Content}
Question: Does this hate speech text insult against people based on their attribute of {labels}?
Answer: |
| | Hate-T-ben$_{Kar}$ | {Content}
Question: Does this text express {labels}?
Answer: |
| | Offe-T-mal$_{Cha}$, Offe-T-tam$_{Cha}$ | {Content}
Question: Is this sentence hate speech or not? If yes, does this sentence target individual, group or not?
Answer: |
| **Emotion** | Emot-eng$_{Wal}$, Emot-zho$_{Lee}$, Emot-fin$_{Kaj}$, Emot-fra$_{Kaj}$, Emot-ita$_{Kaj}$, Emot-msa$_{Hus}$, Emot-ara$_{Abd}$ Emot-eng$_{Moh}$, Emot-ara$_{Moh}$, Emot-spa$_{Moh}$, Emot-ind$_{Sap}$, Emot-tur$_{Guv}$, Emot-spa$_{Moh}$, Emot-ind$_{Wil}$ Emot-vie$_{Ho}$, Emot-eng$_{Pla}$, Emot-spa$_{Pla}$, Emot-fin$_{Ohm}$, Emot-eng$_{Dem}$, Emot-ita$_{Bia}$, Emot-ron$_{Cio}$ Emot-hin$_{Deb}$, Emot-por$_{Cor}$, Emot-fas$_{Sab}$, Emot-rus$_{Sbo}$, Emot-ben$_{Iqb}$, Emot-fra$_{Bia}$, Emot-deu$_{Bia}$ | {Content}
Question: Is the emotion of this sentence {labels}?
Answer: |
| **Humor** | Humo-hin$_{Agg}$, Humo-rus$_{Bli}$, Humo-spa$_{Chi}$, Humo-eng$_{Mea}$ | {Content}
Question: Is this sentence {labels}?
Answer: |
| **Irony** | Iron-ita$_{Bas}$, Iron-spa$_{Bar}$, Iron-eng$_{Hee}$, Iron-ita$_{Cig}$, Iron-hin$_{Vij}$, Iron-ara$_{Gha}$, Iron-spa$_{Ort}$ Iron-fas$_{Gol}$, Iron-zho$_{Xia}$, Sarc-eng$_{Wal}$, Sarc-eng$_{Ril}$, Sarc-ces$_{Pta}$, Sarc-eng$_{Pta}$, Sarc-eng$_{Bam}$ Sarc-eng$_{Raj}$, Sarc-eng$_{Ora}$, Sarc-zho$_{Gon}$, Sarc-ara$_{Abu}$, Sarc-ara$_{Far}$ | {Content}
Question: Is this sentence {labels}?
Answer: |
| | Iron-T-eng$_{Hee}$ | {Content}
Question: Is the type of this text {labels}?
Answer: |
| **Sentiment Analysis** | Sent-eng$_{Pan}$, Sent-zho$_{Tan}$, Sent-kor$_{Jan}$, Sent-eng$_{Soc}$, Sent-ita$_{Bas}$, Sent-ben$_{Pat}$, Sent-hin$_{Pat}$ Sent-ita$_{Bas}$, Sent-mlt$_{Din}$, Sent-bul$_{Moz}$, Sent-bos$_{Moz}$, Sent-deu$_{Moz}$, Sent-eng$_{Moz}$, Sent-spa$_{Moz}$ Sent-hrv$_{Moz}$, Sent-hun$_{Moz}$, Sent-pol$_{Moz}$, Sent-por$_{Moz}$, Sent-rus$_{Moz}$, Sent-slk$_{Moz}$, Sent-slv$_{Moz}$ Sent-sqi$_{Moz}$, Sent-srp$_{Moz}$, Sent-swe$_{Moz}$, Sent-deu$_{Rei}$, Sent-spa$_{Rei}$, Sent-ita$_{Rei}$, Sent-eng$_{Ros}$ Sent-heb$_{Amr}$, Sent-por$_{Bru}$, Sent-fin$_{Kaj}$, Sent-fra$_{Kaj}$, Sent-ita$_{Kaj}$, Sent-pol$_{Koc}$, Sent-tha$_{Sur}$ Sent-zho$_{Wan}$, Sent-fas$_{Ash}$, Sent-ron$_{Dum}$, Sent-pcm$_{Oye}$, Sent-pol$_{Ryb}$, Sent-ind$_{Wil}$, Sent-ara$_{Abd}$ Sent-bam$_{Dia}$, Sent-ben$_{Isl}$, Sent-mar$_{Kul}$, Sent-kan$_{Cha}$, Sent-mal$_{Cha}$, Sent-tam$_{Cha}$, Sent-ara$_{Muh}$ Sent-amh$_{Muh}$, Sent-ary$_{Muh}$, Sent-arq$_{Muh}$, Sent-hau$_{Muh}$, Sent-ibo$_{Muh}$, Sent-pcm$_{Muh}$, Sent-kin$_{Muh}$ Sent-swh$_{Muh}$, Sent-tso$_{Muh}$, Sent-twi$_{Muh}$, Sent-yor$_{Muh}$, Sent-yor$_{Sho}$, Sent-jpn$_{Suz}$, Sent-ace$_{Win}$ Sent-ban$_{Win}$, Sent-bbc$_{Win}$, Sent-bjn$_{Win}$, Sent-bug$_{Win}$, Sent-jav$_{Win}$, Sent-mad$_{Win}$, Sent-min$_{Win}$ Sent-nij$_{Win}$, Sent-sun$_{Win}$, Sent-tel$_{Mar}$, Sent-T-eng$_{The}$, Sent-Y-eng$_{The}$, Sent-5-eng$_{Soc}$ | {Content}
Question: Is the sentiment of this sentence {labels}?
Answer: |
| | Sent-nor$_{Vel}$ | {Content}
Question: Is this text rated as {labels}? Higher is better.
Answer: |
| **Subjective** | Subj-kor$_{Jan}$ | {Content}
Question: Does this sentence express {labele}?
Answer: |
| | Subj-eng$_{Pan}$, Subj-ita$_{Bas}$, Subj-ita$_{Bas}$, Subj-spa$_{Bar}$, Subj-ces$_{Pri}$ | {Content}
Question: Is this sentence {labels}?
Answer: |

Table 15: Prompts use for zero-shot evaluation with lm-evaluation-harness.

| | | Dev Set | | | | Test-S Set | | | |
|---|---|---|---|---|---|---|---|---|---|
| | | mBERT | XLM-R | Bernice | InfoDCL | mBERT | XLM-R | Bernice | InfoDCL |
| Antisocial | Aggressive | 73.39±0.30 | 73.92±0.50 | 76.79±0.52 | 76.09±0.38 | 72.71±1.92 | 74.64±0.14 | 75.45±0.73 | 73.96±0.91 |
| | Dangerours | 69.76±1.56 | 69.53±1.04 | 74.92±1.18 | 73.42±0.80 | 62.36±1.08 | 63.57±1.15 | 67.13±0.56 | 65.23±1.60 |
| | Hate | 77.73±0.76 | 79.40±0.85 | 81.16±1.15 | 80.62±0.60 | 72.97±1.40 | 74.37±1.48 | 76.76±2.43 | 75.85±0.90 |
| | Offense | 78.96±1.04 | 80.21±0.92 | 82.15±0.68 | 81.55±0.46 | 77.53±1.27 | 75.88±2.43 | 78.45±1.89 | 78.88±2.63 |
| | H/O-Group | 51.57±2.35 | 41.24±3.16 | 48.30±1.90 | 46.05±2.25 | 46.18±4.10 | 42.39±3.30 | 51.15±2.01 | 50.24±4.19 |
| | H/O-Target | 54.02±3.60 | 59.23±1.71 | 60.83±1.26 | 60.14±1.21 | 53.16±4.49 | 57.67±1.79 | 60.96±2.26 | 60.79±1.38 |
| | AS | 70.18±1.59 | 71.47±1.24 | 73.80±1.13 | 73.04±0.85 | 66.92±2.29 | 67.99±1.84 | 71.14±2.15 | 70.61±1.64 |
| Emotion | | 62.30±0.90 | 67.43±0.68 | 68.56±0.85 | 69.34±0.53 | 61.42±1.51 | 66.87±0.99 | 68.13±1.28 | 69.27±1.03 |
| Humor | | 85.15±0.32 | 85.83±0.57 | 86.72±0.63 | 86.74±0.42 | 84.35±1.23 | 85.19±1.62 | 86.75±1.13 | 87.05±0.82 |
| I&S | Irony | 69.16±1.29 | 69.94±1.02 | 72.19±1.24 | 71.12±0.72 | 64.24±1.16 | 65.53±1.57 | 66.88±1.23 | 68.38±1.00 |
| | Sarcasm | 74.65±1.14 | 75.64±1.92 | 77.82±1.56 | 77.21±1.05 | 72.41±1.38 | 73.40±2.42 | 74.78±1.69 | 74.94±1.13 |
| | Irony-Type | 53.51±2.11 | 52.18±2.15 | 58.55±3.17 | 57.72±2.92 | 47.35±1.89 | 46.43±0.63 | 56.04±1.87 | 57.58±1.42 |
| | I&S | 71.13±1.26 | 71.90±1.52 | 74.32±1.50 | 70.39±1.49 | 67.48±1.31 | 68.51±1.95 | 70.29±1.49 | 71.12±1.09 |
| Sentiment | | 69.29±1.14 | 71.34±0.78 | 72.95±0.88 | 73.81±0.70 | 66.34±1.92 | 69.58±1.41 | 70.44±1.61 | 71.64±1.31 |
| Subjectivity | | 75.18±0.63 | 77.28±0.74 | 76.97±0.69 | 77.78±0.87 | 72.54±1.46 | 74.45±1.18 | 74.80±1.08 | 75.73±1.33 |
| SM | | 69.29±1.17 | 71.50±0.93 | 73.16±0.98 | 73.46±0.72 | 66.60±1.83 | 69.38±1.51 | 70.85±1.63 | 71.60±1.30 |

Table 16: Performance of finetuned models on Dev and Test-S set. We finetune each model on each dataset for three runs with different random seeds and calculate the mean and standard deviation of dataset-specific metrics over the three runs. We report the average of dataset-specific metrics and standard deviation in a task and a category. **I&S:** irony and sarcasm.

| Dataset | Metric | Random | mBERT | XLM-R | Bernice | InfoDCL | BLOOM | BLOOMZ | BLOOMZ (MT) | BLOOMZ-P3 | BLOOMZ-Bactrian | mT5 | mT0 | mT0 (MT) | LLaMA | Alpaca | Vicuna | ChatGPT | ChatGPT (MT) | SoTA | SoTA study |
|---|---|---|---|---|---|---|---|---|---|---|---|---|---|---|---|---|---|---|---|---|---|
| Aggr-hin_Kum | M-F1 | 43.14 | 72.71 | 74.64 | **75.45** | 73.96 | 51.06 | 15.82 | 15.82 | 18.72 | 16.37 | 53.67 | 15.82 | 22.00 | 18.31 | 49.29 | 25.07 | 63.53 | 54.36 | 70.00 | Kumar et al. (2018) |
| Dang-ara_Als | M-F1 | 42.06 | 62.36 | 63.57 | 67.13 | 65.23 | 46.87 | 46.87 | 50.84 | 46.87 | 46.87 | 49.31 | 46.87 | 46.87 | 46.87 | 46.87 | 46.87 | 79.93 | 79.98 | 59.60 | Alshehri et al. (2020) |
| Hate-eng_Was | W-F1 | 41.57 | 87.74 | **89.33** | 88.79 | 88.93 | 58.26 | 65.96 | 65.96 | 60.07 | 60.17 | 18.66 | 59.75 | 59.75 | 60.42 | 37.33 | 61.66 | 79.98 | 79.98 | 73.62 | Waseem and Hovy (2016) |
| Hate-eng_Dav | M-F1 | 38.97 | 91.28 | **92.96** | 91.99 | 91.12 | 7.99 | 10.78 | 10.78 | 25.43 | 9.32 | 13.36 | 8.33 | 8.33 | 8.33 | 73.10 | 55.89 | 69.58 | 63.96 | 90.00 | Davidson et al. (2017) |
| Hate-ara_Ala | M-F1 | 43.70 | 82.07 | 81.23 | 83.99 | **85.95** | 54.28 | 43.63 | 43.63 | 43.63 | 52.08 | 24.11 | 43.63 | 43.63 | 43.63 | 43.50 | 43.63 | 63.96 | 52.85 | — | — |
| Hate-ita_Bos | M-F1 | 47.00 | 75.67 | 76.96 | 80.19 | **81.17** | 44.63 | 40.26 | 40.26 | 40.26 | 45.69 | 25.55 | 40.26 | 40.26 | 40.26 | 42.34 | 40.26 | 77.98 | 57.62 | 79.93 | Bosco et al. (2018) |
| Hate-fil_Cab | M-F1 | 52.37 | 74.38 | 78.40 | **79.50** | 79.01 | 46.79 | 34.47 | 45.46 | 34.47 | 46.80 | 32.93 | 34.47 | 57.76 | 34.47 | 47.00 | 34.47 | 69.13 | 66.67 | 71.12 | Cabasag et al. (2019) |
| Hate-ara_Mul | M-F1 | 29.56 | 69.38 | 67.12 | **76.53** | 71.74 | 15.62 | 25.46 | 25.46 | 37.90 | 19.22 | 14.50 | 25.46 | 25.46 | 25.46 | 17.16 | 30.99 | 61.13 | 32.01 | 89.60 | Mulki et al. (2019) |
| Hate-eng_Bas | M-F1 | 52.61 | 50.24 | 51.87 | 54.17 | 53.25 | 51.11 | 36.22 | 36.22 | 36.22 | 52.86 | 29.97 | 36.22 | 36.22 | 36.65 | 53.15 | 37.96 | **63.69** | 63.69 | 65.10 | Basile et al. (2019) |
| Hate-spa_Bas | M-F1 | 44.59 | 74.18 | 76.26 | **78.20** | 76.96 | 46.61 | 37.50 | 29.21 | 37.50 | 48.68 | 31.06 | 37.50 | 37.50 | 38.95 | 59.16 | 37.50 | 64.93 | 54.43 | 73.00 | Basile et al. (2019) |
| Hate-por_For | M-F1 | 47.84 | 70.08 | 70.02 | **74.40** | 73.22 | 48.01 | 39.69 | 39.90 | 39.69 | 51.53 | 29.73 | 39.69 | 39.69 | 40.09 | 55.27 | 39.69 | 68.45 | 63.34 | 72.00 | Fortuna et al. (2019) |
| Hate-pol_Ryb | M-F1 | 44.15 | 69.69 | 70.26 | 71.68 | 71.23 | 47.77 | 46.47 | 46.47 | 46.47 | 48.05 | 12.41 | 46.47 | 46.47 | 47.46 | 53.11 | 46.47 | **72.72** | 66.36 | 50.30 | Rybak et al. (2020) |
| Hate-kor_Moo | M-F1 | 36.74 | 57.10 | 63.17 | **64.80** | 63.09 | 18.03 | 16.90 | 21.94 | 20.95 | 16.90 | 15.88 | 16.90 | 20.30 | 16.90 | 27.59 | 16.90 | 39.79 | 46.90 | 63.30 | Moon et al. (2020) |
| Hate-ara_Mub | M-F1 | 37.84 | 73.92 | 79.67 | 81.16 | **82.00** | 35.69 | 48.67 | 48.67 | 48.67 | 34.08 | 8.36 | 48.67 | 48.67 | 48.67 | 48.51 | 46.87 | 84.88 | 69.36 | 84.79 | Abdul-Mageed et al. (2021) |
| Hate-zho_Den | M-F1 | 48.72 | 83.29 | 83.36 | 84.39 | **84.86** | 69.31 | 36.22 | 30.17 | 36.22 | 60.96 | 30.56 | 36.22 | 38.13 | 36.73 | 36.22 | 36.22 | 72.34 | 74.24 | 81.00 | Deng et al. (2022) |
| Hate-kor_Jeo | M-F1 | 51.96 | 79.03 | 78.50 | **80.19** | 79.32 | 35.55 | 33.69 | 33.69 | 33.69 | 34.75 | 35.77 | 33.69 | 34.05 | 33.87 | 40.06 | 33.69 | 63.61 | 57.09 | 77.20 | Jeong et al. (2022) |
| Hate-tel_Mar | M-F1 | 34.32 | 49.90 | 49.78 | **58.22** | 49.75 | 32.67 | 49.90 | 49.90 | 49.90 | 48.19 | 0.60 | 49.90 | 49.90 | 49.90 | 49.90 | 49.90 | 49.90 | 34.23 | 60.00 | Marreddy et al. (2022) |
| Sexi-fra_Chi | M-F1 | 46.05 | 79.60 | 81.01 | 79.96 | **81.99** | 24.92 | 65.25 | 36.45 | 25.26 | 46.40 | 49.17 | 40.05 | 38.65 | 42.95 | 25.60 | 51.53 | 74.81 | 70.29 | 76.20 | Chiril et al. (2020) |
| Offe-eng_Zam | M-F1 | 44.20 | 75.07 | 75.10 | 77.75 | **78.67** | 42.06 | 42.06 | 42.06 | 42.62 | 42.06 | 23.97 | 42.06 | 42.06 | 42.06 | 25.57 | 59.67 | 67.90 | 67.90 | 82.90 | Zampieri et al. (2019) |
| Offe-ara_Zam | M-F1 | 45.64 | 86.27 | 86.88 | **91.55** | 89.53 | 44.07 | 44.07 | 17.49 | 44.07 | 44.07 | 27.25 | 44.07 | 17.49 | 44.47 | 17.49 | 58.47 | 82.01 | 67.52 | 90.17 | Zampieri et al. (2020) |
| Offe-dan_Zam | M-F1 | 41.14 | 77.53 | 76.11 | 78.03 | **82.09** | 46.68 | 46.68 | 11.08 | 46.68 | 46.68 | 20.35 | 46.68 | 11.08 | 46.68 | 11.46 | 51.84 | 66.91 | 66.79 | 81.19 | Zampieri et al. (2020) |
| Offe-ell_Zam | M-F1 | 41.24 | **80.64** | 76.91 | 79.63 | 79.13 | 45.47 | 45.47 | 14.24 | 46.71 | 45.47 | 33.99 | 45.47 | 14.24 | 45.47 | 14.24 | 48.21 | 60.94 | 34.98 | 85.22 | Zampieri et al. (2020) |
| Offe-tur_Zam | M-F1 | 41.11 | 73.11 | 76.90 | **77.07** | 76.08 | 44.38 | 44.38 | 19.92 | 44.38 | 44.38 | 38.31 | 44.38 | 44.38 | 44.38 | 16.81 | 51.12 | 75.03 | 45.68 | 82.58 | Zampieri et al. (2020) |
| Offe-ara_Mub | M-F1 | 46.28 | 86.85 | 86.44 | **93.16** | 91.40 | 43.69 | 43.69 | 18.30 | 43.69 | 43.63 | 22.86 | 43.69 | 18.30 | 43.69 | 18.58 | 58.96 | 84.48 | 69.36 | 90.50 | Mubarak et al. (2020) |
| Offe-slv_Nov | M-F1 | 16.75 | **63.23** | 52.83 | 51.98 | 55.29 | 21.02 | 16.60 | 18.88 | 1.98 | 16.68 | 8.19 | 12.93 | 0.20 | 12.57 | 13.57 | 12.59 | 33.86 | 16.67 | - | - |
| Offe-G-eng_Zam | M-F1 | 31.22 | 54.49 | 55.44 | **61.93** | 61.45 | 28.72 | 36.00 | 36.00 | 44.00 | 25.66 | 19.73 | 26.95 | 26.51 | 31.46 | 30.73 | 20.68 | 51.52 | 51.52 | 75.50 | Zampieri et al. (2019) |
| Hate-G-ara_Ous | M-F1 | 6.89 | 46.71 | 37.11 | **52.38** | 51.08 | 3.87 | 7.21 | 0.07 | 8.41 | 7.55 | 1.32 | 14.83 | 16.67 | 0.00 | 0.27 | 1.43 | 35.05 | 10.90 | 40.00 | Ousidhoum et al. (2019) |
| Hate-G-fra_Ous | M-F1 | 6.36 | 37.35 | 34.61 | **39.13** | 38.19 | 8.29 | 8.57 | 6.61 | 11.28 | 11.23 | 0.00 | 6.98 | 7.42 | 5.61 | 11.39 | 5.66 | 32.41 | 17.79 | 37.00 | Ousidhoum et al. (2019) |
| Offe-T-eng_Zam | M-F1 | 39.77 | 63.61 | 64.04 | **78.16** | 72.80 | 47.02 | 47.02 | 47.02 | 47.02 | 47.63 | 10.62 | 22.75 | 22.75 | 58.67 | 39.89 | 37.48 | 45.97 | 45.97 | 66.00 | Zampieri et al. (2019) |
| Hate-T-ara_Ous | M-F1 | 21.42 | 48.32 | 52.94 | 52.96 | **53.84** | 10.15 | 19.64 | 8.65 | 15.58 | 14.25 | 11.76 | 30.19 | 11.07 | 6.48 | 7.64 | 6.48 | 44.51 | 33.69 | 53.00 | Ousidhoum et al. (2019) |
| Hate-T-fra_Ous | M-F1 | 14.85 | 45.88 | 43.39 | 43.30 | **48.18** | 4.11 | 8.07 | 19.58 | 7.92 | 2.11 | 2.08 | 18.84 | 0.22 | 2.08 | 2.26 | 3.15 | 39.98 | 41.14 | 43.00 | Ousidhoum et al. (2019) |
| Hate-T-ben_Kar | M-F1 | 22.89 | 83.56 | 85.31 | 85.84 | **86.30** | 11.12 | 15.52 | 22.29 | 7.53 | 12.98 | 12.11 | 23.15 | 42.74 | 18.48 | 12.76 | 23.42 | 49.03 | 9.30 | 87.00 | Karim et al. (2021) |
| Offe-T-kan_Cha | M-F1 | 14.73 | 42.13 | 38.78 | **46.47** | 40.65 | 16.69 | 16.69 | 9.50 | 16.61 | 16.69 | 4.27 | 8.09 | 6.51 | 20.01 | 15.44 | 14.16 | 21.19 | 9.63 | 43.00 | Chakravarthi et al. (2022) |
| Offe-T-mal_Cha | M-F1 | 12.71 | 42.12 | 74.22 | 76.51 | **81.53** | 24.04 | 24.04 | 24.04 | 20.81 | 24.04 | 1.17 | 26.80 | 24.04 | 24.04 | 24.04 | 24.04 | 19.32 | 4.00 | 72.00 | Chakravarthi et al. (2022) |
| Offe-T-tam_Cha | M-F1 | 14.28 | 38.70 | 36.25 | **39.98** | 39.76 | 17.32 | 17.35 | 4.39 | 15.76 | 17.35 | 2.26 | 17.90 | 17.82 | 17.22 | 17.34 | 17.35 | 25.72 | 11.61 | 44.00 | Chakravarthi et al. (2022) |
| Hate-T-kor_Jeo | M-F1 | 22.51 | 60.98 | **66.43** | 64.47 | 63.25 | 19.37 | 3.87 | 14.46 | 3.87 | 15.14 | 4.28 | 16.91 | 18.80 | 7.56 | 15.27 | 9.96 | 41.35 | 29.67 | 62.70 | Jeong et al. (2022) |
| Average | — | 35.20 | 66.92 | 67.99 | **71.14** | 70.61 | 33.70 | 32.80 | 27.93 | 31.97 | 33.79 | 20.14 | 32.02 | 28.79 | 31.68 | 30.55 | 34.50 | 56.55 | 47.40 | — | — |

Table 17: Full Test-S results on Antisocial task. **SoTA:** Previous SoTA performance on each respective dataset. **Underscore** indicates that we have different data splits to the SoTA model. Best model of each dataset is in **bold**.

| Dataset | Metric | Random | mBERT | XLM-R | Bernice | InfoDCL | BLOOM | BLOOMZ | BLOOMZ (MT) | BLOOMZ-P3 | BLOOMZ-Bactrian | mT5 | mT0 | mT0 (MT) | LLaMA | Alpaca | Vicuna | ChatGPT | ChatGPT (MT) | SoTA | SoTA study |
|---|---|---|---|---|---|---|---|---|---|---|---|---|---|---|---|---|---|---|---|---|---|
| Emot-eng_Wal | M-F1 | 12.00 | 65.35 | 69.24 | 69.51 | **70.10** | 16.86 | 21.13 | 21.13 | 11.05 | 24.58 | 4.54 | 27.01 | 27.01 | 47.94 | 60.08 | 44.88 | 68.06 | 68.06 | 57.00 | Suresh and Ong (2021) |
| Emot-zho_Lee | Acc. | 16.27 | 65.87 | 70.81 | 67.30 | **72.97** | 14.11 | 52.63 | 56.94 | 51.44 | 18.42 | 23.92 | 55.50 | 57.42 | 10.63 | 49.76 | 19.14 | 68.93 | 68.93 | 53.90 | Lee and Wang (2015) |
| Emot-fin_Kaj | M-F1 | 13.32 | 46.15 | 58.61 | 49.54 | **58.87** | 4.88 | 3.45 | 3.53 | 3.45 | 8.46 | 3.61 | 18.70 | 8.89 | 3.30 | 3.86 | 5.45 | 52.79 | 46.58 | - | — |
| Emot-fra_Kaj | M-F1 | 12.85 | 49.99 | 58.23 | 60.10 | **60.04** | 4.24 | 12.51 | 12.04 | 4.39 | 8.45 | 3.75 | 20.70 | 22.32 | 8.82 | 17.40 | 14.23 | 58.29 | 45.88 | - | — |
| Emot-ita_Kaj | M-F1 | 13.28 | 54.61 | 59.35 | 60.68 | **60.84** | 6.21 | 8.78 | 4.34 | 5.87 | 12.75 | 5.62 | 20.58 | 14.28 | 8.60 | 14.08 | 11.84 | 60.53 | 38.16 | - | — |
| Emot-ara_Abd | M-F1 | 13.16 | 55.77 | 59.75 | **66.24** | 65.69 | 5.08 | 9.42 | 10.93 | 6.65 | 8.92 | 5.17 | 13.98 | 5.32 | 3.19 | 5.33 | 9.49 | 32.09 | 27.15 | 60.32 | Abdul-Mageed et al. (2020) |
| Emot-eng_Moh | M-F1 | 24.41 | 70.90 | 78.30 | 80.86 | **81.37** | 16.48 | 9.63 | 9.63 | 29.70 | 31.00 | 15.51 | 47.57 | 47.57 | 15.77 | 59.71 | 25.15 | 74.44 | 74.44 | 78.50 | Barbieri et al. (2020) |
| Emot-ara_Moh | M-F1 | 22.98 | 72.93 | 83.19 | **84.10** | 83.29 | 19.12 | 23.15 | 34.83 | 21.07 | 22.68 | 12.70 | 35.32 | 8.75 | 26.18 | 41.16 | 25.85 | 63.13 | 32.01 | 91.00 | Bianchi et al. (2022) |
| Emot-spa_Moh | M-F1 | 25.58 | 78.53 | 82.27 | 83.33 | **85.49** | 15.56 | 25.14 | 8.30 | 19.50 | 33.17 | 12.53 | 42.72 | 23.43 | 25.37 | 54.06 | 29.18 | 75.89 | 74.64 | 91.00 | Bianchi et al. (2022) |
| Emot-ind_Sap | M-F1 | 19.69 | 63.58 | 76.93 | 78.37 | **81.27** | 8.85 | 36.78 | 7.69 | 35.53 | 15.55 | 6.73 | 29.03 | 34.63 | 26.94 | 36.37 | 24.04 | 77.51 | 60.20 | 68.00 | Saputri et al. (2018) |
| Emot-tur_Guv | Acc. | 18.34 | 98.25 | 98.50 | 98.92 | **99.33** | 22.50 | 20.25 | 22.00 | 20.25 | 19.50 | 21.75 | 51.75 | 49.00 | 22.75 | 36.50 | 19.75 | 89.50 | 84.00 | 87.00 | Güven et al. (2020) |
| Emot-vie_Ho | W-F1 | 16.12 | 54.63 | 63.50 | 63.12 | **64.58** | 2.19 | 14.90 | 5.52 | 12.22 | 10.20 | 1.76 | 27.81 | 24.03 | 2.04 | 16.34 | 9.52 | 54.69 | 32.96 | 66.34 | Ho et al. (2019) |
| Emot-eng_Pla | M-F1 | 11.23 | 39.09 | 46.49 | 48.51 | 47.45 | 5.30 | 9.74 | 9.74 | 7.80 | 10.12 | 2.93 | 19.29 | 19.29 | 7.96 | 25.59 | 20.60 | 40.73 | 40.73 | 32.00 | Plaza del Arco et al. (2020) |
| Emot-spa_Pla | M-F1 | 11.47 | 50.89 | 52.28 | **57.95** | 54.25 | 5.64 | 12.63 | 10.31 | 7.77 | 3.52 | 3.36 | 13.54 | 13.45 | 9.41 | 27.48 | 22.31 | 40.51 | 44.00 | 44.00 | Plaza del Arco et al. (2020) |
| Emot-fin_Ohm | M-F1 | 13.60 | 41.74 | 49.82 | 45.07 | 50.36 | 5.96 | 3.78 | 2.39 | 2.84 | 7.67 | 4.39 | 14.47 | 7.32 | 3.73 | 3.73 | 6.04 | 52.94 | 42.55 | - | — |
| Emot-eng_Dem | M-F1 | 22.54 | 66.47 | 67.76 | 76.78 | 75.58 | 16.21 | 27.07 | 15.42 | 21.44 | 22.65 | 10.80 | 38.89 | 13.15 | 27.47 | 50.42 | 24.10 | 73.66 | 73.19 | 64.80 | Suresh and Ong (2021) |
| Emot-ita_Bia | M-F1 | 22.54 | 66.47 | 67.76 | 76.78 | 75.58 | 16.21 | 27.07 | 15.42 | 21.44 | 22.65 | 10.80 | 38.89 | 13.15 | 27.47 | 50.42 | 24.10 | 73.66 | 73.19 | 71.00 | Bianchi et al. (2021) |
| Emot-ron_Cio | M-F1 | 23.99 | 87.63 | 91.24 | 89.26 | **91.43** | 11.37 | 19.59 | 9.90 | 14.75 | 30.85 | 10.25 | 41.07 | 33.29 | 26.95 | 52.76 | 30.81 | 84.74 | 60.97 | 78.00 | Ciobotaru and Dinu (2021) |
| Emot-eng_Deb | M-F1 | 3.10 | 45.42 | 50.17 | 48.52 | **52.46** | 1.17 | 4.70 | 3.03 | 4.51 | 3.67 | 0.37 | 9.07 | 12.67 | 2.44 | 6.70 | 2.88 | 31.31 | 19.91 | - | — |
| Emot-por_Cor | M-F1 | 3.44 | 72.09 | 73.93 | **76.70** | 75.16 | 0.68 | 2.62 | 3.58 | 1.12 | 2.99 | 0.18 | 2.55 | 0.65 | 1.05 | 6.70 | 2.54 | 8.72 | 8.98 | 64.00 | Cortiz et al. (2021) |
| Emot-fas_Sab | M-F1 | 20.27 | 21.49 | 21.12 | 21.15 | 26.21 | 5.23 | 11.15 | 3.35 | 12.00 | 5.24 | 1.69 | 14.03 | 7.79 | 5.13 | 12.26 | 4.47 | **28.59** | 27.80 | 27.80 | |
| Emot-rus_Sbo | M-F1 | 15.84 | 76.49 | 83.17 | 81.96 | **83.75** | 8.61 | 15.05 | 12.36 | 11.02 | 11.74 | 3.57 | 30.27 | 21.97 | 8.30 | 51.06 | 8.59 | 79.62 | 76.53 | 78.00 | Sboev et al. (2020) |
| Emot-ben_Hqb | M-F1 | 15.97 | 53.79 | 59.57 | **63.86** | 63.69 | 10.43 | 13.37 | 15.01 | 11.39 | 9.36 | 5.19 | 17.90 | 30.11 | 2.98 | 9.52 | 2.98 | 53.23 | 48.50 | - | — |
| Emot-fra_Bia | M-F1 | 20.09 | 75.07 | 79.98 | **84.15** | 83.56 | 20.36 | 29.15 | 50.61 | 20.00 | 37.94 | 12.18 | 47.99 | 54.75 | 30.42 | 63.48 | 39.38 | 79.02 | 78.19 | 88.00 | Bianchi et al. (2022) |
| Emot-deu_Bia | M-F1 | 21.56 | 71.35 | 78.21 | **81.75** | 79.18 | 15.48 | 21.00 | 14.22 | 20.26 | 15.96 | 12.25 | 43.21 | 51.05 | 26.96 | 69.86 | 32.10 | 76.07 | 24.99 | 87.00 | Bianchi et al. (2022) |
| Average | — | 15.86 | 61.42 | 66.87 | 68.13 | **69.27** | 9.71 | 17.18 | 13.85 | 15.07 | 15.19 | 7.75 | 27.87 | 24.21 | 15.14 | 31.80 | 18.12 | 59.58 | 50.85 | — | — |

Table 18: Full Test-S results on emotion recognition. **SoTA:** Previous SoTA performance on each respective dataset. **Underscore** indicates that we have different data splits to the SoTA model. Best model of each dataset is in **bold**.

| Dataset | Metric | Random | mBERT | XLM-R | Bernice | InfoDCL | BLOOM | BLOOMZ | BLOOMZ (MT) | BLOOMZ-P3 | BLOOMZ-Bactrian | mT5 | mT0 | mT0 (MT) | LLaMA | Alpaca | Vicuna | ChatGPT | ChatGPT (MT) | SoTA | SoTA study |
|---|---|---|---|---|---|---|---|---|---|---|---|---|---|---|---|---|---|---|---|---|---|
| Humo-hin$_{Agg}$ | Acc. | 51.40 | 78.27 | 78.87 | 78.27 | **80.40** | 58.60 | 39.60 | 39.80 | 39.60 | 40.80 | 52.20 | 39.60 | 42.40 | 56.80 | 60.40 | 37.20 | 56.00 | 61.00 | 69.30 | Aggarwal et al. (2020) |
| Humo-rus$_{Bli}$ | M-F1 | 47.38 | 86.47 | 87.60 | 88.13 | **88.80** | 34.12 | 34.12 | 34.12 | 34.12 | 36.38 | 45.09 | 34.12 | 34.12 | 35.36 | 32.52 | 54.60 | 72.75 | 71.44 | 89.00 | Blinov et al. (2019) |
| Humo-spa$_{Chi}$ | M-F1 | 54.58 | 85.46 | 84.45 | 87.20 | **87.66** | 48.25 | 32.07 | 32.07 | 31.97 | 39.39 | 34.28 | 32.07 | 32.07 | 38.56 | 34.55 | 50.10 | 68.28 | 68.80 | 88.50 | Chiruzzo et al. (2021) |
| Humo-eng$_{Mea}$ | M-F1 | 45.25 | 87.19 | 89.86 | **93.41** | 91.34 | 26.16 | 26.69 | 26.69 | 26.47 | 27.06 | 42.83 | 26.69 | 26.69 | 28.39 | 39.39 | 42.86 | 89.56 | 89.56 | 98.54 | Meaney et al. (2021) |
| Average | — | 49.65 | 84.35 | 85.19 | 86.75 | **87.05** | 41.78 | 33.12 | 33.17 | 33.04 | 35.91 | 43.60 | 33.12 | 33.82 | 39.78 | 41.72 | 46.19 | 71.65 | 72.70 | — | — |

Table 19: Full Test-S results on humor detection. **SoTA:** Previous SoTA performance on each respective dataset. **Underscore** indicates that we have different data splits to the SoTA model.

| Dataset | Metric | Random | mBERT | XLM-R | Bernice | InfoDCL | BLOOM | BLOOMZ | BLOOMZ (MT) | BLOOMZ-P3 | BLOOMZ-Bactrian | mT5 | mT0 | mT0 (MT) | LLaMA | Alpaca | Vicuna | ChatGPT | ChatGPT (MT) | SoTA | SoTA study |
|---|---|---|---|---|---|---|---|---|---|---|---|---|---|---|---|---|---|---|---|---|---|
| Iron-ita$_{Bas}$ | M-F1 | 41.87 | 63.38 | 61.54 | 63.16 | **65.67** | 46.92 | 46.92 | 50.38 | 50.97 | 46.92 | 49.59 | 46.92 | 46.92 | 49.02 | 13.27 | 55.49 | 59.91 | 55.16 | 59.59 | Basile et al. (2014) |
| Iron-spa$_{Bar}$ | M-F1 | 40.70 | 57.90 | 58.01 | 61.66 | **67.52** | 46.47 | 46.47 | 48.65 | 56.77 | 46.47 | 48.89 | 46.47 | 46.47 | 49.34 | 13.88 | 52.78 | 66.10 | 63.97 | 54.12 | Barbieri et al. (2016) |
| Iron-eng$_{Hee}$ | F1-iro. | 45.00 | 59.99 | 63.43 | 67.43 | **68.25** | 0.00 | 0.00 | 0.00 | 25.41 | 0.00 | 1.03 | 0.00 | 0.00 | 5.29 | 55.06 | 41.19 | 59.00 | 59.00 | 70.50 | Van Hee et al. (2018) |
| Iron-ita$_{Cig}$ | M-F1 | 51.80 | 70.37 | 72.66 | **77.44** | 75.92 | 34.67 | 34.21 | 45.82 | 48.21 | 34.21 | 38.78 | 34.21 | 34.21 | 48.22 | 33.24 | 56.22 | 73.32 | 74.20 | 73.10 | Cignarella et al. (2018) |
| Iron-hin$_{Vij}$ | M-F1 | 46.88 | 70.90 | **73.25** | 72.90 | 71.16 | 56.66 | 43.99 | 57.75 | 51.30 | 55.80 | 46.28 | 42.53 | 46.63 | 52.02 | 23.61 | 57.73 | 52.89 | 57.92 | 77.00 | Vijay et al. (2018) |
| Iron-ara$_{Gha}$ | M-F1 | 48.39 | 82.19 | **83.95** | 82.45 | 83.08 | 40.10 | 34.24 | 32.61 | 51.43 | 32.61 | 33.80 | 32.61 | 32.61 | 39.17 | 34.40 | 35.94 | 68.78 | 67.40 | 84.40 | Abdul-Mageed et al. (2020) |
| Iron-eng$_{Ort}$ | M-F1 | 46.84 | 67.42 | 71.18 | 72.79 | **73.88** | 39.47 | 39.47 | 53.40 | 54.46 | 39.47 | 40.22 | 39.47 | 39.47 | 60.64 | 29.47 | 56.77 | 62.92 | 61.69 | 71.67 | Ortega-Bueno et al. (2019) |
| Iron-fas$_{Gol}$ | Acc. | 48.30 | 74.04 | 74.60 | 73.58 | **76.53** | 51.70 | 56.46 | 56.46 | 51.70 | 56.46 | 56.46 | 56.46 | 56.46 | 49.66 | 43.88 | 57.82 | 62.59 | 56.80 | 83.10 | Golazizian et al. (2020) |
| Iron-zho$_{Xia}$ | M-F1 | 11.71 | 31.96 | 31.19 | 30.52 | **33.36** | 13.65 | 14.56 | 3.09 | 9.89 | 13.65 | 13.65 | 13.55 | 3.17 | 13.61 | 0.63 | 13.40 | 18.58 | 10.06 | 57.20 | Xiang et al. (2020) |
| Sarc-eng$_{Wal}$ | M-F1 | 53.33 | 63.32 | 65.36 | 67.45 | 63.65 | 45.21 | 32.80 | 32.80 | 33.23 | 41.54 | 45.01 | 32.80 | 32.80 | 34.55 | 50.47 | 33.38 | **70.79** | 70.79 | 69.00 | Felbo et al. (2017) |
| Sarc-eng$_{Ril}$ | F1-sar. | 27.00 | 46.76 | 52.50 | 54.89 | **57.46** | 10.53 | 0.00 | 0.00 | 0.00 | 0.00 | 36.36 | 0.00 | 0.00 | 0.00 | 38.97 | 14.81 | 50.00 | 50.00 | 51.00 | Riloff et al. (2013) |
| Sarc-ces$_{Pta}$ | M-F1 | 35.92 | 66.12 | 60.17 | 65.97 | **67.89** | 46.90 | 49.03 | 3.68 | 49.03 | 49.75 | 33.94 | 49.03 | 3.68 | 34.46 | 6.22 | 55.00 | 51.20 | 52.48 | 58.20 | Ptáček et al. (2014) |
| Sarc-eng$_{Pta}$ | M-F1 | 49.85 | 94.28 | **95.76** | 94.99 | 95.56 | 41.77 | 38.42 | 38.42 | 39.49 | 37.05 | 45.25 | 38.42 | 38.42 | 40.45 | 33.01 | 48.17 | 74.30 | 74.30 | 92.37 | Ptáček et al. (2014) |
| Sarc-eng$_{Bam}$ | Acc. | 52.20 | 79.73 | 80.40 | **82.40** | 82.27 | 48.60 | 52.00 | 52.00 | 51.80 | 50.60 | 49.00 | 52.00 | 52.00 | 49.20 | 52.00 | 52.20 | 64.60 | 64.60 | 85.10 | Bamman and Smith (2015) |
| Sarc-eng$_{Raj}$ | Acc. | 48.80 | 94.20 | 95.33 | **96.27** | 95.67 | 77.60 | 91.20 | 91.20 | 90.80 | 87.80 | 19.60 | 91.20 | 91.20 | 87.00 | 15.00 | 83.60 | 74.60 | 74.60 | 92.94 | Rajadesingan et al. (2015) |
| Sarc-eng$_{Ora}$ | M-F1 | 49.00 | 72.73 | **75.87** | 74.69 | 75.64 | 41.48 | 32.89 | 32.89 | 32.80 | 42.11 | 41.83 | 32.89 | 32.89 | 33.71 | 47.08 | 36.08 | 73.78 | 73.78 | 75.00 | Felbo et al. (2017) |
| Sarc-zho$_{Gon}$ | M-F1 | 48.25 | 71.01 | 70.63 | **72.75** | 70.53 | 49.04 | 33.29 | 33.20 | 33.20 | 56.10 | 38.89 | 33.29 | 33.29 | 36.25 | 41.41 | 49.36 | 53.50 | 51.05 | 73.68 | Gong et al. (2020) |
| Sarc-ara$_{Abu}$ | M-F1 | 44.26 | 69.06 | 69.57 | 69.57 | 71.87 | 27.16 | 44.57 | 16.39 | 44.57 | 45.15 | 20.66 | 44.57 | 16.39 | 53.38 | 17.21 | 53.74 | 74.38 | **75.47** | 76.30 | Abdul-Mageed et al. (2021) |
| Sarc-ara$_{Far}$ | M-F1 | 46.22 | 66.87 | 68.45 | 68.80 | **68.81** | 41.67 | 42.00 | 21.63 | 41.93 | 53.31 | 30.34 | 42.00 | 21.63 | 42.65 | 23.47 | 50.37 | 66.24 | 68.43 | 73.10 | Farha et al. (2021) |
| Iron-T-eng$_{Hee}$ | M-F1 | 22.36 | 47.35 | 46.43 | 56.04 | **57.58** | 18.83 | 18.83 | 18.83 | 18.83 | 18.83 | 18.83 | 18.83 | 18.83 | 18.83 | 18.83 | 18.83 | 30.81 | 30.81 | 50.70 | Van Hee et al. (2018) |
| Average | — | 42.93 | 67.48 | 68.51 | 70.29 | **71.12** | 38.92 | 37.57 | 34.46 | 41.79 | 40.39 | 35.42 | 37.36 | 32.35 | 39.87 | 29.56 | 46.14 | 60.41 | 59.63 | — | — |

Table 20: Full Test-S results on irony & sarcasm detection. **SoTA:** Previous SoTA performance on each respective dataset. **Underscore** indicates that we have different data splits to the SoTA model. Best model of each dataset is in **bold**.

Table 21 (Full Test-S results on sentiment analysis):

| Dataset | Metric | Random | mBERT | XLM-R | Bernice | InfoDCL | BLOOM | BLOOMZ | BLOOMZ (MT) | BLOOMZ-P3 | BLOOMZ-Bactrian | mT5 | mT0 | mT0 (MT) | LLaMA | Alpaca | Vicuna | ChatGPT | ChatGPT (MT) | SoTA | SoTA study |
|---|---|---|---|---|---|---|---|---|---|---|---|---|---|---|---|---|---|---|---|---|---|
| Sent-eng$_{Pan}$ | Acc. | 51.80 | 81.60 | 86.07 | 85.20 | 86.93 | 51.60 | **97.20** | 97.20 | 96.80 | 56.40 | 51.40 | 76.80 | 76.80 | 61.80 | 60.20 | 55.40 | 88.40 | 88.40 | 90.82 | Ke et al. (2020) |
| Sent-zho$_{Tan}$ | M-F1 | 49.04 | 95.72 | 95.85 | **96.85** | 95.92 | 36.40 | 90.52 | 87.19 | 87.73 | 58.32 | 31.88 | 86.57 | 90.14 | 56.83 | 49.59 | 45.31 | 90.16 | 88.39 | 95.80 | Sun et al. (2020) |
| Sent-T-eng$_{The}$ | Acc. | 47.00 | 79.20 | 88.73 | **91.40** | 89.60 | 47.20 | 75.60 | 75.60 | 78.40 | 45.20 | 141.20 | 73.80 | 73.80 | 43.20 | 54.00 | 55.60 | 90.00 | 90.00 | 88.00 | Felbo et al. (2017) |
| Sent-Y-eng$_{The}$ | Acc. | 50.20 | 86.00 | 90.80 | 92.87 | **93.33** | 42.40 | 84.80 | 84.80 | 85.20 | 36.80 | 130.60 | 81.60 | 81.60 | 32.60 | 47.40 | 33.20 | 90.00 | 90.00 | 93.00 | Felbo et al. (2017) |
| Sent-5-eng$_{Soc}$ | Acc. | 19.80 | 49.73 | **53.93** | 51.73 | 53.67 | 24.80 | 48.60 | 48.60 | 48.60 | 19.80 | 27.60 | 45.60 | 45.60 | 20.00 | 40.40 | 19.60 | 49.60 | 49.60 | 58.59 | Ke et al. (2020) |
| Sent-kor$_{Jan}$ | M-F1 | 20.79 | 38.64 | 42.56 | 41.05 | **44.70** | 2.96 | 29.03 | 3.70 | 28.08 | 1.11 | 13.02 | 25.91 | 19.91 | 12.78 | 25.37 | 8.01 | 42.32 | 36.49 | - | — |
| Sent-eng$_{Soc}$ | Acc. | 46.80 | 84.47 | 88.40 | 86.87 | 88.73 | 56.40 | 92.20 | 92.20 | **93.00** | 53.60 | 49.00 | 76.80 | 76.80 | 53.40 | 58.20 | 49.80 | 91.80 | 91.80 | 96.70 | Tian et al. (2020) |
| Sent-ita$_{Bas}$ | M-F1 | 46.79 | 78.06 | 85.22 | 85.33 | **88.77** | 40.60 | 54.18 | 39.08 | 74.32 | 41.25 | 38.50 | 53.07 | 38.50 | 54.20 | 51.58 | 60.85 | 87.26 | 87.01 | 67.71 | Basile et al. (2014) |
| Sent-ita$_{Bas}$ | M-F1 | 49.61 | 66.68 | 78.51 | 79.93 | 84.73 | 46.61 | 45.37 | 42.04 | 63.71 | 42.63 | 40.76 | 44.99 | 40.76 | 57.97 | 52.71 | 67.62 | 83.09 | **85.21** | 66.38 | Barbieri et al. (2016) |
| Sent-mlt$_{Pin}$ | M-F1 | 47.51 | 63.14 | 39.79 | 65.96 | 68.01 | 39.36 | 36.75 | 39.36 | 47.70 | 41.14 | 39.36 | 39.25 | 25.97 | 53.99 | 39.36 | 48.98 | 78.47 | 77.45 | 54.70 | Dingli and Sant (2016) |
| Sent-bul$_{Moz}$ | M-F1 | 32.48 | 62.01 | 64.22 | 63.09 | 65.09 | 22.05 | 22.61 | 15.15 | 27.35 | 29.54 | 111.75 | 19.37 | 29.02 | 13.24 | 16.88 | 23.69 | 59.32 | 54.39 | 52.00 | Mozetič et al. (2016) |
| Sent-bos$_{Moz}$ | M-F1 | 34.37 | 64.75 | 68.12 | 65.83 | 68.31 | 24.58 | 26.39 | 16.31 | 31.80 | 24.05 | 116.31 | 21.16 | 50.69 | 27.92 | 19.32 | 16.72 | 63.22 | 48.15 | 60.60 | Mozetič et al. (2016) |
| Sent-deu$_{Moz}$ | M-F1 | 32.60 | 61.63 | 61.54 | 62.80 | 62.53 | 17.36 | 26.96 | 9.39 | 26.13 | 20.10 | 9.79 | 20.54 | 21.93 | 22.50 | 15.88 | 25.51 | 49.70 | 47.64 | 53.60 | Mozetič et al. (2016) |
| Sent-eng$_{Moz}$ | M-F1 | 30.33 | 62.81 | 68.53 | 68.59 | 68.78 | 26.73 | 36.94 | 36.94 | 36.52 | 27.25 | 12.90 | 22.92 | 22.92 | 15.77 | 28.86 | 23.87 | 60.61 | 60.61 | 63.00 | Mozetič et al. (2016) |
| Sent-spa$_{Moz}$ | M-F1 | 29.19 | 51.39 | 55.80 | 55.79 | 55.87 | 25.84 | 31.85 | 10.32 | 33.31 | 24.18 | 7.25 | 26.36 | 7.25 | 19.94 | 14.69 | 20.63 | 39.39 | 42.48 | 38.60 | Mozetič et al. (2016) |
| Sent-hrv$_{Moz}$ | M-F1 | 30.23 | 64.79 | 69.44 | 66.68 | 67.83 | 19.35 | 27.96 | 11.02 | 35.18 | 20.68 | 11.02 | 27.47 | 11.02 | 24.32 | 14.45 | 13.67 | 62.02 | 55.66 | 60.60 | Mozetič et al. (2016) |
| Sent-hun$_{Moz}$ | M-F1 | 27.80 | 67.27 | 71.62 | 70.16 | 68.37 | 15.80 | 28.87 | 8.61 | 30.31 | 19.62 | 6.93 | 27.19 | 39.94 | 12.18 | 8.66 | 16.39 | 53.22 | 43.89 | 64.10 | Mozetič et al. (2016) |
| Sent-pol$_{Moz}$ | M-F1 | 30.95 | 67.09 | 66.96 | 67.91 | 68.22 | 18.69 | 27.58 | 14.36 | 34.78 | 26.65 | 14.34 | 25.57 | 14.91 | 24.98 | 20.83 | 14.91 | 61.84 | 60.58 | 67.70 | Mozetič et al. (2016) |
| Sent-por$_{Moz}$ | M-F1 | 29.81 | 57.14 | 56.21 | 56.37 | 56.71 | 32.20 | 26.22 | 18.15 | 27.65 | 34.78 | 18.22 | 18.03 | 18.22 | 31.48 | 26.59 | 18.94 | 44.60 | 42.76 | 55.30 | Mozetič et al. (2016) |
| Sent-rus$_{Moz}$ | M-F1 | 32.33 | 75.24 | 78.35 | 78.84 | 80.37 | 26.77 | 30.42 | 28.64 | 31.33 | 30.70 | 14.99 | 20.60 | 30.55 | 18.19 | 21.73 | 22.35 | 61.67 | 59.27 | 61.50 | Mozetič et al. (2016) |
| Sent-slk$_{Moz}$ | M-F1 | 28.57 | 71.78 | 74.78 | 71.82 | 75.71 | 20.49 | 27.36 | 20.32 | 31.89 | 18.14 | 14.09 | 27.85 | 23.54 | 25.01 | 17.93 | 10.46 | 57.46 | 56.18 | 68.20 | Mozetič et al. (2016) |
| Sent-slv$_{Moz}$ | M-F1 | 35.90 | 59.53 | 61.57 | 61.27 | 61.96 | 26.34 | 19.79 | 14.66 | 26.76 | 22.95 | 14.66 | 20.75 | 14.66 | 27.09 | 20.27 | 20.69 | 58.64 | 57.26 | 55.30 | Mozetič et al. (2016) |
| Sent-sqi$_{Moz}$ | M-F1 | 29.39 | 43.07 | 46.42 | 45.69 | 47.04 | 18.25 | 26.88 | 10.76 | 31.78 | 23.15 | 9.98 | 26.59 | 41.09 | 27.51 | 9.98 | 16.11 | 46.82 | 33.84 | 39.10 | Mozetič et al. (2016) |
| Sent-srp$_{Moz}$ | M-F1 | 34.09 | 53.16 | 56.62 | 52.51 | 56.85 | 28.06 | 20.66 | 19.82 | 27.45 | 24.20 | 16.99 | 20.55 | 44.27 | 31.95 | 22.73 | 19.05 | 55.84 | 52.54 | 60.60 | Mozetič et al. (2016) |
| Sent-swe$_{Moz}$ | M-F1 | 33.35 | 64.61 | 68.84 | 69.93 | 70.98 | 24.85 | 25.84 | 19.32 | 29.31 | 28.14 | 19.66 | 20.77 | 26.36 | 24.57 | 22.56 | 16.05 | 60.10 | 62.58 | 65.70 | Mozetič et al. (2016) |
| Sent-deu$_{Rei}$ | M-F1 | 19.05 | 47.05 | 52.99 | 55.93 | 60.94 | 11.67 | 14.46 | 3.01 | 13.61 | 16.78 | 4.26 | 15.78 | 18.18 | 13.82 | 8.76 | 30.32 | 30.95 | 33.58 | - | — |
| Sent-spa$_{Rei}$ | M-F1 | 19.78 | 39.59 | 51.26 | 51.53 | 52.83 | 17.86 | 6.70 | 6.60 | 8.12 | 12.91 | 1.69 | 8.49 | 1.69 | 15.72 | 14.71 | 31.50 | 25.01 | 28.54 | - | — |
| Sent-ita$_{Rei}$ | M-F1 | 23.76 | 47.29 | 49.57 | 47.14 | 51.20 | 27.92 | 10.75 | 3.30 | 12.45 | 25.39 | 3.45 | 11.30 | 3.30 | 14.80 | 10.59 | 31.17 | 25.39 | 30.61 | - | — |
| Sent-eng$_{Ros}$ | M-Rec | 36.77 | 64.52 | 67.50 | 70.90 | 70.39 | 33.60 | 52.75 | 52.75 | 57.60 | 32.29 | 33.33 | 38.89 | 38.89 | 33.97 | 38.72 | 39.31 | 69.94 | 69.94 | 72.60 | Barbieri et al. (2020) |
| Sent-ben$_{Pat}$ | M-F1 | 28.06 | 54.20 | 58.28 | 57.56 | 59.45 | 23.88 | 34.29 | 30.34 | 34.73 | 22.12 | 13.67 | 28.91 | 26.36 | 19.53 | 18.53 | 21.01 | 55.00 | 29.14 | 52.60 | Patra et al. (2018) |
| Sent-hin$_{Pat}$ | M-F1 | 31.67 | 55.46 | 58.31 | 62.30 | 59.76 | 28.73 | 28.74 | 11.59 | 30.45 | 28.23 | 11.57 | 23.96 | 26.66 | 13.80 | 20.58 | 22.45 | 59.40 | 48.30 | 56.90 | Patra et al. (2018) |
| Sent-heb$_{Amr}$ | Acc. | 47.60 | 93.27 | 95.27 | 95.40 | 95.80 | 64.80 | 71.00 | 32.20 | 71.20 | 60.80 | 32.20 | 76.60 | 32.20 | 67.00 | 34.80 | 40.80 | 84.20 | 57.40 | 89.06 | Amram et al. (2018) |
| Sent-por$_{Bru}$ | M-F1 | 35.59 | 59.05 | 60.00 | 57.83 | 59.29 | 24.38 | 26.73 | 20.55 | 29.33 | 34.23 | 19.68 | 20.31 | 19.65 | 28.19 | 25.63 | 15.49 | 42.64 | 42.39 | 62.14 | Brum and das Graças Volpe Nunes (2018) |
| Sent-fin$_{Kaj}$ | M-F1 | 50.11 | 78.88 | 83.48 | 79.86 | 82.32 | 35.48 | 36.55 | 35.65 | 41.24 | 41.33 | 135.65 | 58.17 | 74.27 | 38.38 | 40.41 | 57.76 | 83.77 | 83.32 | - | — |
| Sent-fra$_{Kaj}$ | M-F1 | 47.70 | 78.44 | 84.21 | 84.12 | 87.50 | 35.72 | 71.71 | 31.32 | 70.88 | 42.54 | 135.23 | 58.67 | 63.50 | 49.58 | 60.49 | 68.81 | 87.96 | 87.78 | - | — |
| Sent-ita$_{Kaj}$ | M-F1 | 49.65 | 79.21 | 85.70 | 84.59 | 86.17 | 59.43 | 56.28 | 35.98 | 65.24 | 36.48 | 35.98 | 64.76 | 35.98 | 57.65 | 56.67 | 64.59 | 86.03 | 84.69 | - | — |
| Sent-nor$_{Vel}$ | M-F1 | 16.24 | 41.05 | 51.15 | 39.73 | 42.36 | 0.00 | 18.43 | 4.09 | 19.84 | 0.00 | 1.93 | 35.73 | 39.75 | 0.00 | 11.76 | 9.80 | 42.11 | 41.41 | - | — |
| Sent-pol$_{Koc}$ | M-F1 | 25.03 | 70.99 | 76.03 | 77.05 | 77.63 | 10.17 | 32.45 | 13.02 | 32.20 | 9.23 | 13.02 | 39.51 | 40.34 | 21.75 | 30.42 | 9.63 | 50.86 | 40.71 | - | — |
| Sent-tha$_{Sur}$ | M-F1 | 31.26 | 65.07 | 72.18 | 71.99 | 75.17 | 20.65 | 20.51 | 24.37 | 20.16 | 11.04 | 13.33 | 23.94 | 29.83 | 13.29 | 14.21 | 24.40 | 52.81 | 36.32 | - | — |
| Sent-zho$_{Wan}$ | M-F1 | 46.66 | 98.80 | 99.00 | 98.67 | 98.93 | 46.49 | 69.80 | 49.89 | 70.65 | 61.31 | 33.51 | 56.51 | 81.16 | 53.89 | 45.32 | 65.70 | 77.83 | 79.08 | 91.20 | Wan et al. (2020) |
| Sent-fas$_{Ash}$ | M-F1 | 30.51 | 82.70 | 84.80 | 84.56 | 84.54 | 10.25 | 33.77 | 26.77 | 41.34 | 10.25 | 9.98 | 45.69 | 49.56 | 12.76 | 10.25 | 17.63 | 73.87 | 68.91 | 80.00 | Ashrafi Asli et al. (2020) |
| Sent-ron$_{Dum}$ | M-F1 | 52.34 | 84.07 | 88.60 | 84.12 | 88.23 | 37.82 | 65.40 | 30.26 | 83.18 | 44.66 | 130.26 | 93.21 | 30.26 | 62.09 | 32.21 | 30.26 | 93.49 | 93.72 | - | — |
| Sent-pcm$_{Oye}$ | M-F1 | 31.09 | 65.11 | 66.86 | 68.87 | 68.84 | 24.96 | 33.63 | 30.52 | 38.99 | 23.01 | 18.19 | 25.29 | 25.29 | 46.16 | 15.17 | 15.93 | 52.19 | 31.79 | - | — |
| Sent-pol$_{Ryb}$ | M-F1 | 21.56 | 45.45 | 53.98 | 51.12 | 55.11 | 8.72 | 11.80 | 3.83 | 13.82 | 5.92 | 3.83 | 16.28 | 21.80 | 10.58 | 15.91 | 5.58 | 43.30 | 31.48 | - | — |
| Sent-ind$_{Wil}$ | M-F1 | 30.52 | 88.28 | 92.06 | 92.52 | 92.70 | 18.36 | 58.25 | 16.42 | 59.11 | 45.24 | 16.19 | 50.79 | 36.23 | 16.16 | 23.89 | 12.78 | 73.49 | 73.98 | 92.72 | Wilie et al. (2020) |
| Sent-ara$_{Abd}$ | M-F1 | 29.17 | 71.96 | 75.75 | 77.36 | 77.30 | 28.05 | 42.96 | 24.15 | 44.48 | 32.36 | 24.15 | 26.60 | 49.01 | 28.43 | 27.56 | 8.21 | 61.90 | 60.56 | 80.86 | Elmadany et al. (2022) |
| Sent-bam$_{Dia}$ | M-F1 | 27.61 | 64.28 | 58.46 | 65.46 | 65.57 | 19.89 | 31.54 | 8.74 | 36.70 | 19.66 | 8.74 | 24.31 | 8.74 | 15.47 | 9.94 | 14.40 | 40.27 | 37.11 | 72.00 | Diallo et al. (2021) |
| Sent-ben$_{Isl}$ | M-F1 | 31.18 | 62.95 | 67.99 | 66.74 | 68.54 | 18.50 | 42.84 | 29.71 | 45.18 | 29.11 | 18.50 | 30.75 | 31.68 | 18.50 | 18.50 | 15.08 | 55.76 | 29.58 | 64.61 | Islam et al. (2021) |
| Sent-mar$_{Kul}$ | Acc. | 29.60 | 84.00 | 86.40 | 84.40 | 86.47 | 35.40 | 46.20 | 35.60 | 43.80 | 35.80 | 35.00 | 35.40 | 39.20 | 36.80 | 38.20 | 34.40 | 68.60 | 50.20 | 84.13 | Kulkarni et al. (2021) |
| Sent-kan$_{Cha}$ | M-F1 | 49.55 | 78.92 | 80.55 | 82.35 | 82.92 | 27.44 | 55.22 | 22.77 | 61.86 | 23.82 | 122.77 | 47.51 | 51.22 | 29.28 | 23.75 | 24.07 | 68.44 | 51.56 | 68.50 | Chakravarthi et al. (2022) |
| Sent-mal$_{Cha}$ | M-F1 | 50.66 | 84.00 | 83.50 | 84.03 | 83.88 | 25.81 | 62.16 | 22.24 | 66.87 | 24.76 | 122.24 | 56.51 | 64.97 | 35.54 | 23.18 | 25.65 | 68.75 | 58.89 | 60.50 | Chakravarthi et al. (2022) |
| Sent-tam$_{Cha}$ | M-F1 | 44.61 | 70.91 | 75.98 | 75.79 | 75.87 | 26.73 | 58.06 | 44.93 | 61.29 | 19.37 | 115.40 | 50.34 | 54.33 | 22.52 | 16.59 | 17.85 | 65.98 | 54.79 | 59.00 | Chakravarthi et al. (2022) |
| Sent-ara$_{Muh}$ | Acc. | 33.20 | 48.73 | 56.40 | 61.20 | 62.20 | 34.80 | 51.60 | 34.40 | 52.80 | 37.00 | 34.40 | 41.60 | 53.40 | 36.60 | 37.80 | 25.00 | 58.40 | 58.20 | 75.16 | Abdul-Mageed et al. (2022) |
| Sent-amh$_{Muh}$ | W-F1 | 37.95 | 24.93 | 60.32 | 57.21 | 65.88 | 54.53 | 27.76 | 22.74 | 16.05 | 49.04 | 54.53 | 22.49 | 17.13 | 54.53 | 54.53 | 2.99 | 20.62 | 46.82 | 78.42 | Muhammad et al. (2023b) |
| Sent-ary$_{Muh}$ | W-F1 | 35.94 | 46.56 | 50.21 | 52.61 | 53.44 | 23.29 | 34.49 | 14.17 | 37.40 | 23.98 | 14.17 | 23.41 | 32.35 | 20.11 | 18.02 | 16.64 | 52.19 | 51.66 | 64.83 | Muhammad et al. (2023b) |
| Sent-arq$_{Muh}$ | W-F1 | 34.23 | 57.16 | 65.09 | 70.21 | 71.25 | 35.26 | 49.79 | 34.23 | 52.02 | 37.83 | 34.23 | 18.05 | 33.91 | 34.91 | 39.80 | 5.33 | 63.89 | 67.58 | 74.20 | Muhammad et al. (2023b) |
| Sent-hau$_{Muh}$ | W-F1 | 35.22 | 69.94 | 73.44 | 73.11 | 72.18 | 24.86 | 20.66 | 21.67 | 30.14 | 33.26 | 16.55 | 20.93 | 16.55 | 19.83 | 16.97 | 17.39 | 55.52 | 34.13 | 82.62 | Muhammad et al. (2023b) |
| Sent-ibo$_{Muh}$ | W-F1 | 32.90 | 76.37 | 76.52 | 78.36 | 76.75 | 25.58 | 16.24 | 27.92 | 23.21 | 37.48 | 12.25 | 11.52 | 22.73 | 14.25 | 14.96 | 28.37 | 57.55 | 33.46 | 82.96 | Muhammad et al. (2023b) |
| Sent-pcm$_{Muh}$ | W-F1 | 35.62 | 60.82 | 62.44 | 63.61 | 64.45 | 28.07 | 53.47 | 45.54 | 55.33 | 48.18 | 139.73 | 23.21 | 22.73 | 40.18 | 42.36 | 11.91 | 70.08 | 53.71 | 55.96 | Muhammad et al. (2023b) |
| Sent-kin$_{Muh}$ | W-F1 | 35.83 | 55.32 | 57.44 | 58.78 | 56.69 | 19.99 | 23.02 | 18.33 | 28.06 | 26.01 | 18.33 | 14.84 | 28.08 | 24.32 | 19.17 | 22.23 | 53.78 | 29.03 | 72.63 | Muhammad et al. (2023b) |
| Sent-swh$_{Muh}$ | W-F1 | 33.72 | 53.04 | 61.15 | 55.19 | 61.29 | 5.09 | 17.80 | 13.20 | 17.60 | 8.05 | 1.96 | 14.02 | 13.20 | 9.45 | 2.36 | 48.55 | 54.39 | 53.84 | 65.68 | Muhammad et al. (2023b) |
| Sent-tso$_{Muh}$ | W-F1 | 34.03 | 42.64 | 48.56 | 52.47 | 52.55 | 21.65 | 38.43 | 30.48 | 45.54 | 25.42 | 18.54 | 30.74 | 30.82 | 24.58 | 21.05 | 6.50 | 42.58 | 35.70 | 60.67 | Muhammad et al. (2023b) |
| Sent-twi$_{Muh}$ | W-F1 | 34.04 | 63.32 | 64.29 | 65.43 | 64.51 | 22.32 | 37.26 | 29.31 | 46.56 | 16.91 | 20.74 | 29.34 | 28.99 | 23.18 | 24.48 | 5.28 | 51.12 | 32.06 | 68.28 | Muhammad et al. (2023b) |
| Sent-yor$_{Muh}$ | W-F1 | 36.56 | 67.87 | 64.38 | 66.71 | 64.84 | 14.14 | 33.14 | 14.12 | 31.70 | 18.87 | 7.29 | 28.58 | 7.29 | 9.27 | 8.11 | 19.01 | 57.57 | 37.21 | 80.16 | Muhammad et al. (2022) |
| Sent-yor$_{Sho}$ | M-F1 | 46.80 | 86.86 | 83.69 | 85.39 | 84.93 | 33.24 | 56.04 | 34.22 | 54.50 | 34.03 | 133.33 | 43.07 | 35.32 | 40.83 | 33.33 | 33.75 | 71.97 | 47.47 | 87.20 | Shode et al. (2022) |
| Sent-jpn$_{Suz}$ | Acc. | 18.00 | 54.33 | 59.67 | 61.13 | 61.47 | 32.00 | 45.20 | 26.00 | 44.20 | 33.40 | 125.80 | 47.60 | 14.20 | 32.00 | 19.60 | 26.80 | 55.80 | 44.40 | 61.50 | Suzuki et al. (2022) |
| Sent-ace$_{Win}$ | M-F1 | 34.78 | 76.59 | 74.19 | 75.74 | 77.38 | 23.31 | 28.22 | 22.89 | 37.89 | 20.40 | 18.44 | 24.73 | 20.99 | 19.39 | 19.38 | 12.79 | 52.63 | 58.05 | 77.40 | Winata et al. (2022) |
| Sent-ban$_{Win}$ | M-F1 | 30.14 | 75.96 | 74.79 | 76.86 | 79.49 | 21.96 | 35.00 | 31.84 | 41.90 | 23.12 | 18.27 | 29.91 | 24.44 | 19.80 | 18.44 | 13.82 | 60.91 | 42.28 | 79.50 | Winata et al. (2022) |
| Sent-bbc$_{Win}$ | M-F1 | 30.60 | 72.66 | 65.57 | 75.17 | 73.58 | 19.23 | 24.61 | 36.37 | 36.42 | 23.68 | 18.48 | 19.20 | 18.27 | 18.77 | 18.44 | 13.86 | 38.43 | 40.65 | 76.70 | Winata et al. (2022) |
| Sent-bjn$_{Win}$ | M-F1 | 30.77 | 75.81 | 79.82 | 82.58 | 84.50 | 20.52 | 41.73 | 43.86 | 50.51 | 34.89 | 18.44 | 27.78 | 18.74 | 18.44 | 22.07 | 14.89 | 69.34 | 75.43 | 86.30 | Winata et al. (2022) |
| Sent-bug$_{Win}$ | M-F1 | 30.77 | 74.57 | 67.72 | 73.85 | 71.55 | 19.04 | 20.69 | 31.51 | 34.60 | 20.92 | 18.39 | 18.27 | 18.27 | 19.59 | 18.44 | 12.90 | 34.63 | 30.86 | 77.20 | Winata et al. (2022) |
| Sent-jav$_{Win}$ | M-F1 | 31.62 | 76.08 | 83.44 | 84.38 | 84.79 | 20.11 | 35.48 | 18.44 | 48.06 | 23.68 | 18.44 | 37.65 | 33.97 | 18.95 | 18.44 | 15.21 | 73.03 | 78.56 | 85.60 | Winata et al. (2022) |
| Sent-mad$_{Win}$ | M-F1 | 28.64 | 70.56 | 73.36 | 77.36 | 78.36 | 22.42 | 31.23 | 37.89 | 45.44 | 23.12 | 18.44 | 21.85 | 18.27 | 18.44 | 18.92 | 13.14 | 60.14 | 61.14 | 77.80 | Winata et al. (2022) |
| Sent-min$_{Win}$ | M-F1 | 34.41 | 77.25 | 80.89 | 80.19 | 84.07 | 18.39 | 41.20 | 35.18 | 49.93 | 28.76 | 18.44 | 32.41 | 21.71 | 18.44 | 18.44 | 14.95 | 69.80 | 62.91 | 83.10 | Winata et al. (2022) |
| Sent-nij$_{Win}$ | M-F1 | 34.86 | 73.82 | 73.47 | 78.19 | 77.22 | 19.01 | 35.18 | 39.43 | 42.86 | 27.70 | 18.44 | 22.89 | 19.65 | 18.44 | 18.44 | 15.21 | 57.64 | 57.07 | 75.80 | Winata et al. (2022) |
| Sent-sun$_{Win}$ | M-F1 | 32.18 | 75.32 | 76.61 | 77.70 | 81.71 | 18.87 | 31.98 | 18.44 | 44.83 | 27.14 | 18.44 | 37.65 | 12.90 | 18.86 | 18.44 | 12.93 | 64.97 | 68.76 | 86.00 | Winata et al. (2022) |
| Sent-tel$_{Mar}$ | M-F1 | 30.27 | 65.08 | 69.44 | 67.14 | 69.47 | 9.98 | 38.81 | 23.32 | 32.08 | 13.52 | 9.99 | 20.40 | 36.96 | 9.98 | 9.98 | 23.32 | 56.77 | 43.14 | 62.00 | Marreddy et al. (2022) |
| Average | — | 34.68 | 66.34 | 69.58 | 70.44 | 71.64 | 26.67 | 39.03 | 28.61 | 43.03 | 28.46 | 20.77 | 34.65 | 32.76 | 27.55 | 25.84 | 25.02 | 60.34 | 54.9 | — | — |

Table 21: Full Test-S results on sentiment analysis. **SoTA:** Previous SoTA performance on each respective dataset. **Underscore** indicates that we have different data splits to the SoTA model. Best model of each dataset is in **bold**.

Table 22 (Full test results on subjectivity analysis):

| Dataset | Metric | Random | mBERT | XLM-R | Bernice | InfoDCL | BLOOM | BLOOMZ | BLOOMZ (MT) | BLOOMZ-P3 | BLOOMZ-Bactrian | mT5 | mT0 | mT0 (MT) | LLaMA | Alpaca | Vicuna | ChatGPT | ChatGPT (MT) | SoTA | SoTA study |
|---|---|---|---|---|---|---|---|---|---|---|---|---|---|---|---|---|---|---|---|---|---|
| Subj-eng$_{Pan}$ | Acc. | 48.60 | 95.00 | 95.13 | 94.53 | **95.20** | 56.80 | 49.00 | 49.00 | 49.00 | 51.00 | 49.60 | 48.80 | 48.80 | 52.80 | 49.60 | 50.60 | 73.40 | 73.40 | 97.10 | Chen et al. (2022) |
| Subj-kor$_{Jan}$ | M-F1 | 10.74 | 36.53 | 36.99 | **37.78** | 36.95 | 2.33 | 3.15 | 3.63 | 10.80 | 3.94 | 9.85 | 12.80 | 1.93 | 6.55 | 7.17 | 12.76 | 27.41 | 15.14 | - | — |
| Subj-ita$_{Bas}$ | M-F1 | 43.30 | 74.02 | 77.18 | 77.71 | **80.16** | 50.00 | 17.76 | 19.57 | 17.76 | 47.24 | 20.68 | 54.09 | 43.82 | 46.67 | 18.89 | 43.88 | 72.48 | 61.84 | 71.40 | Basile et al. (2014) |
| Subj-ita$_{Bas}$ | M-F1 | 48.51 | 68.10 | 71.34 | 71.65 | 72.36 | 50.14 | 26.14 | 27.63 | 26.14 | 48.18 | 43.43 | 47.04 | 41.79 | 47.24 | 26.14 | 39.55 | **73.51** | 60.37 | 74.44 | Barbieri et al. (2016) |
| Subj-spa$_{Bar}$ | M-F1 | 51.13 | 70.95 | 73.92 | 75.54 | **77.33** | 53.67 | 27.85 | 31.51 | 27.85 | 40.33 | 47.72 | 42.94 | 27.85 | 53.33 | 27.85 | 38.60 | 71.38 | 71.21 | 74.44 | Barbieri et al. (2016) |
| Subj-ces$_{Pri}$ | Acc. | 46.20 | 90.67 | 92.13 | 91.60 | **92.40** | 51.80 | 52.80 | 52.80 | 52.80 | 47.20 | 52.80 | 44.20 | 52.80 | 47.20 | 53.00 | 47.00 | 79.40 | 74.00 | 93.56 | Pribán and Steinberger (2022) |
| Average | — | 41.41 | 72.54 | 74.45 | 74.80 | 75.73 | 44.12 | 29.45 | 30.69 | 30.73 | 39.65 | 37.35 | 41.64 | 36.16 | 42.30 | 30.44 | 38.73 | 66.26 | 59.33 | — | — |

Table 22: Full test results on subjectivity analysis. **SoTA:** Previous SoTA performance on each respective dataset. **Underscore** indicates that we have different data splits to the SoTA model. Best model of each dataset is in **bold**.

| Lang Fam. | Lang | Random | InfoDCL | BMZ-P3 | mT0 | Vicuna | CG | CG-MT |
|---|---|---|---|---|---|---|---|---|
| Afro-Asiatic | ara | 34.05 | **73.53** | 36.78 | 35.61 | 33.61 | 60.81 | 52.53 |
| | amh | 37.95 | **65.68** | 16.05 | 22.49 | 2.99 | 20.62 | 46.82 |
| | arq | 34.23 | **71.25** | 52.02 | 18.05 | 5.33 | 63.89 | 67.58 |
| | ary | 35.94 | **53.44** | 37.40 | 23.41 | 16.64 | 52.19 | 51.66 |
| | hau | 35.22 | **72.18** | 30.14 | 20.93 | 17.39 | 55.52 | 34.13 |
| | heb | 47.60 | **95.80** | 71.20 | 76.60 | 40.80 | 84.20 | 57.40 |
| | mlt | 47.51 | 68.01 | 47.70 | 39.25 | 48.98 | **78.47** | 77.45 |
| Atlantic-C. | bam | 27.61 | **65.57** | 36.70 | 24.31 | 14.40 | 40.27 | 37.11 |
| | ibo | 32.90 | **76.75** | 23.21 | 11.52 | 28.37 | 57.55 | 33.46 |
| | kin | 35.83 | **56.69** | 28.06 | 14.84 | 22.23 | 53.78 | 29.03 |
| | swh | 33.72 | **61.29** | 17.60 | 14.02 | 45.55 | 54.39 | 53.84 |
| | twi | 34.04 | **64.51** | 46.56 | 29.34 | 5.28 | 51.12 | 32.06 |
| | tso | 34.03 | **52.55** | 45.54 | 30.74 | 6.50 | 42.58 | 35.70 |
| | yor | 41.68 | **74.88** | 43.10 | 35.83 | 26.38 | 64.77 | 42.31 |
| Austroasi. | vie | 16.12 | **64.58** | 12.22 | 27.81 | 9.52 | 54.69 | 32.96 |
| Austrones. | ace | 34.78 | **77.36** | 37.89 | 24.73 | 12.79 | 52.63 | 58.05 |
| | ban | 30.14 | **79.49** | 41.90 | 29.91 | 13.82 | 60.91 | 42.28 |
| | bjn | 30.77 | **84.50** | 50.51 | 27.78 | 14.89 | 69.34 | 75.43 |
| | bug | 30.77 | **71.55** | 34.60 | 18.27 | 12.90 | 34.63 | 30.86 |
| | fil | 52.37 | **79.01** | 34.47 | 34.47 | 34.47 | 69.13 | 66.67 |
| | ind | 22.85 | **83.05** | 42.86 | 36.77 | 20.69 | 75.29 | 64.14 |
| | jav | 31.62 | **84.79** | 48.06 | 37.65 | 15.21 | 73.03 | 78.56 |
| | mad | 28.64 | **78.36** | 45.44 | 21.85 | 13.14 | 61.07 | 61.14 |
| | min | 34.41 | **84.07** | 49.93 | 32.41 | 14.95 | 69.80 | 62.91 |
| | nij | 34.86 | **77.22** | 42.86 | 22.89 | 15.21 | 57.64 | 57.07 |
| | sun | 32.18 | **81.71** | 44.83 | 37.65 | 12.93 | 64.97 | 68.76 |
| | bbc | 30.60 | **73.58** | 36.42 | 19.20 | 13.86 | 38.43 | 40.65 |
| Dravidian | kan | 32.14 | **61.79** | 39.23 | 27.80 | 19.12 | 44.81 | 30.39 |
| | mal | 31.68 | **82.70** | 43.84 | 41.65 | 24.85 | 44.03 | 31.44 |
| | tam | 29.45 | **57.81** | 38.53 | 34.27 | 17.60 | 45.85 | 33.20 |
| | tel | 32.29 | **59.61** | 40.99 | 35.15 | 36.61 | 53.33 | 38.68 |
| Indo-Euro. | sqi | 29.39 | **47.04** | 31.78 | 26.59 | 16.11 | 46.82 | 33.84 |
| | bos | 34.37 | **68.31** | 31.80 | 21.16 | 16.72 | 63.22 | 48.15 |
| | bul | 32.48 | **65.09** | 27.35 | 19.37 | 23.69 | 59.32 | 54.39 |
| | ben | 24.53 | **69.49** | 24.71 | 25.18 | 15.62 | 53.25 | 37.33 |
| | hrv | 30.23 | **67.83** | 35.18 | 27.47 | 13.67 | 62.02 | 55.66 |
| | ces | 41.06 | **80.15** | 50.91 | 46.61 | 51.00 | 65.30 | 63.24 |
| | dan | 41.14 | **82.09** | 46.68 | 46.68 | 51.84 | 66.91 | 66.79 |
| | eng | 37.90 | **75.48** | 43.32 | 39.23 | 39.75 | 66.51 | — |
| | fra | 24.65 | **66.58** | 23.29 | 32.20 | 30.46 | 62.08 | 56.84 |
| | deu | 24.40 | **67.55** | 20.00 | 26.51 | 29.31 | 52.24 | 35.41 |
| | ell | 41.24 | **79.13** | 46.71 | 45.47 | 48.21 | 60.94 | 34.98 |
| | hin | 35.24 | **67.55** | 28.92 | 26.20 | 29.06 | 52.63 | 48.30 |
| | ita | 39.83 | **74.78** | 38.76 | 41.45 | 45.05 | 70.29 | 64.37 |
| | mar | 29.60 | **86.47** | 43.80 | 35.40 | 34.40 | 68.60 | 50.20 |
| | pcm | 33.36 | **66.65** | 47.16 | 24.25 | 13.92 | 61.13 | 42.75 |
| | nor | 16.24 | **42.36** | 19.84 | 35.73 | 9.80 | 42.11 | 41.41 |
| | fas | 33.03 | **62.43** | 35.01 | 38.73 | 26.64 | 55.02 | 51.17 |
| | por | 29.17 | **66.10** | 24.45 | 20.14 | 19.16 | 41.10 | 39.37 |
| | pol | 30.43 | **68.04** | 31.82 | 31.96 | 19.15 | 58.25 | 49.43 |
| | ron | 38.16 | **89.83** | 48.97 | 67.14 | 30.54 | 89.11 | 77.34 |
| | rus | 31.85 | **84.31** | 25.49 | 28.33 | 28.51 | 71.34 | 69.08 |
| | spa | 35.98 | **70.20** | 30.80 | 32.17 | 37.71 | 57.16 | 56.64 |
| | srp | 34.09 | **56.85** | 27.45 | 20.55 | 19.05 | 55.84 | 52.54 |
| | slk | 28.57 | **75.71** | 31.89 | 27.85 | 10.46 | 57.46 | 56.18 |
| | slv | 26.32 | **58.62** | 14.37 | 16.84 | 16.64 | 46.25 | 36.97 |
| | swe | 33.35 | **70.98** | 29.31 | 20.77 | 16.05 | 60.10 | 62.58 |
| Japonic | jpn | 18.00 | **61.47** | 44.20 | 47.60 | 26.80 | 55.80 | 44.40 |
| Koreanic | kor | 28.55 | **57.46** | 19.48 | 21.24 | 16.26 | 42.90 | 37.06 |
| Sino-Tib. | zho | 36.77 | **76.10** | 48.19 | 46.94 | 38.19 | 63.43 | 61.39 |
| Tai-Kadai | tha | 31.26 | **75.17** | 20.16 | 23.94 | 24.40 | 52.81 | 36.32 |
| Turkic | tur | 29.56 | **87.71** | 32.32 | 48.07 | 35.44 | 82.27 | 64.84 |
| Uralic | fin | 25.68 | **63.85** | 15.84 | 30.45 | 23.08 | 63.17 | 57.48 |
| | hun | 27.80 | **68.37** | 30.31 | 27.19 | 16.39 | 53.22 | 43.89 |

Table 23: Language-wise model performance. The best performance in each language is **bold**, and the second best is in green highlight . The red font denotes a performance lower than the random baseline. **BMZ:** BLOOMZ, **CG:** ChatGPT, **MT:** using machine translated prompts.

| Task | lm-evaluation-harness Prompts | | | | | | ChatGPT Prompts | | | | | |
|---|---|---|---|---|---|---|---|---|---|---|---|---|
| | BLOOM | BLOOMZ | mT5 | mT0 | LLaMA | Vicuna | BLOOM | BLOOMZ | mT5 | mT0 | LLaMA | Vicuna |
| Hate | 39.83 | 38.52 | 23.29 | 37.33 | 37.80 | **41.59** | 18.39 | 31.34 | 28.96 | 38.19 | 18.37 | 37.33 |
| Emotion | 9.71 | 15.07 | 7.75 | 27.87 | 15.14 | 18.12 | 8.61 | 20.07 | 7.57 | **29.63** | 17.61 | 8.65 |
| Humor | 41.78 | 33.04 | 43.60 | 33.12 | 39.78 | **46.19** | 41.59 | 44.99 | 41.34 | 34.13 | 41.59 | 33.12 |
| Irony | 36.63 | 44.46 | 36.52 | 34.69 | 40.78 | **47.48** | 27.33 | 26.02 | 36.08 | 34.31 | 26.02 | 34.70 |
| **Aveage** | 25.21 | 28.01 | 19.58 | 32.12 | 27.72 | 31.80 | 16.92 | 26.14 | 20.91 | **33.21** | 20.95 | 23.03 |

Table 24: Study on model sensitivity to prompts used for zero-shot evaluation.