# OpenReview forum: "The Skipped Beat: A Study of Sociopragmatic Understanding in LLMs for 64 Languages"
_EMNLP/2023/Conference — EMNLP 2023 Main_

### Official Review · Reviewer_GKU9 · 2023-08-04

**Soundness:** 4

**Excitement:**

4: Strong: This paper deepens the understanding of some phenomenon or lowers the barriers to an existing research direction.

**Paper Topic And Main Contributions:**

The paper presents a large multilingual benchmark for NLP tasks related to sociopragmatic understanding, covering topics like antisocial behavior, irony and sarcasm, emotion detection, humor detection, sentiment analysis, and subjectivity prediction. The benchmark should be useful for assessing and comparing the performance of LLMs and maybe other models at some point for tasks that require subjectivity, commonsense, and pragmatics.

At the same time, the authors offer an in-depth analysis and discussion of various finetuned smaller transformer-based models and zero-shot LLMs (with over 1B parameters) on the benchmark, and also have a good discussion related to each task, to each language, and more.

While all these datasets are public, and the paper does not present any original models, there is an important contribution in the amount of work to gather the benchmark and to make it available to the public, together with the initial set of experiments.

**Questions For The Authors:**

A. Can there be any data leakage between different datasets? E.g. test set for some dataset and task contains very similar examples to the train set of another dataset. This is especially important for the finetuned models

B. InfoDCL uses contrastive learning, it was somehow expected to have better performance compared to a transformer used as an encoder without contrastive learning or additional losses. I feel this should be highlighted when presenting the models.

C. The discussions and results would have been more relevant if some of the larger models have been finetuned (not all of them, just pick the top 1-3) on some tasks or at least tested with few shots in context learning. Any comments for not including at least some few-shot results?

D. For a benchmark containing so many different datasets for different languages and tasks, how do you feel about taking dataset quality into consideration? This seems especially important when computing aggregate scores in a benchmark. Maybe the benchmark should be divided into datasets that are somehow validated to have a good quality and other datasets where data quality is not clear yet.

E. "While we customize prompts employed for each task, we do not tailor prompts specifically for each model." -> Is the ChatGPT prompt providing better results for the other models? Have the authors tested this?

**Reasons To Accept:**

- The SPARROW benchmark, in 64 languages, covering for NLP tasks related to sociopragmatic understanding, that will be useful for assessing and comparing the performance of LLMs and maybe other models.
- In depth discussion and experiments using various finetuned smaller transformer-based models and zero-shot LLMs (with over 1B parameters) on the benchmark
- There seems lots of work in building the benchmark and performing all the experiments - the results and discussions are useful for the NLP community

**Reasons To Reject:**

- The paper does not introduce any original datasets or methods
- The discussions and results would have been more relevant if some of the larger models have been finetuned (not all of them, just pick the top 1-3) on some tasks or at least tested with few shots in context learning

**Reproducibility:**

4: Could mostly reproduce the results, but there may be some variation because of sample variance or minor variations in their interpretation of the protocol or method.

**Reviewer Confidence:**

4: Quite sure. I tried to check the important points carefully. It's unlikely, though conceivable, that I missed something that should affect my ratings.

---

> ### Author Rebuttal · Authors · 2023-08-29
>
> Thanks for your valuable review and questions about our paper.
>
> Reasons To Reject:
> * While it is true that our paper does not introduce new original datasets, the power of our new benchmark comes from the scale and the massively multilingual nature. SPARROW is based on 169 datasets covering 64 languages. We believe that, perceived as one massive resource whose whole is much larger than its individual components, SPARROW is novel. We also believe that it has the potential to accelerate research on multilingual socio-pragmatics and enable meaningful comparisons in the community.
>
> Questions For The Authors:
>
> **A.** Thanks for the question. While we did not verify if the test samples of one dataset overlapped with the training samples of other datasets, we do not think that there is any data leakage in our experiments. For finetuned encoder-only baselines, we finetune and test each model on each individual dataset separately. We do not finetune and test across different datasets. The training and testing sets of each dataset are exclusive. We will emphasize these details in the camera-ready version.
>
> **B.** We use InfoDCL as the strongest encoder-only LM from previous studies. It serves as a SoTA encoder-only LM to compare with other LLMs. Due to the page limit, we provided detailed information on all models in Appendix B. We will clarify the model detail in the main paper in the camera-ready version.
>
> **C.** Due to the expensive cost of finetuning these large models, we provide a case study on few-shot in-context learning (but not finetuning):
> We report a new experiment here to investigate the few-shot in-context learning with 55 datasets from 4 tasks. As this table shows, we compare the results of 3-shot and 5-shot in-context learning with 0-shot results included in the paper. We observe that few-shot in-context learning does enhance the performance of most models (i.e., BLOOM, mT0, LLaMA, and Vicuna). With the increasing number of shots, we can also find that the performances increase. For example, we can see that the Vicuna model obtains average scores over 55 datasets of 31.80, 44.01, and 44.91 with 0, 3, and 5 shots, respectively. However, BLOOMZ-P3 and mT0 do not improve with few-shot in-context learning. We suspect this is because BLOOMZ-P3 and mT0 were finetuned only on NLP datasets, which makes them different from Vicuna that is finetuned with open-ended instructions. These two models (BLOOMZ-P3 and mT0) are also different from BLOOM and LLaMA in that they are finetuned on several NLP tasks only one of which is the sociopragmtic task (i.e., sentiment analysis). We believe this biases the behavior of these two models. We note that we do not perform few-shot in-context learning on sentiment analysis as part of the current experiment because it involves a large number of datasets.
>
>
> In the camera-ready, we will complete a few-shot in-context learning on all 169 datasets in SPARROW.
>
> |         | 0-Shot |        |       |       |       |           | 3-Shot |        |       |       |       |           | 5-Shot |        |       |       |       |           |
> |---------|--------|--------|-------|-------|-------|-----------|--------|--------|-------|-------|-------|-----------|--------|--------|-------|-------|-------|-----------|
> |         | BLOOM  | BLOOMZ-P3 | mT5   | mT0   | LLaMA | Vicuna    | BLOOM  | BLOOMZ-P3 | mT5   | mT0   | LLaMA | Vicuna    | BLOOM  | BLOOMZ-P3 | mT5   | mT0   | LLaMA | Vicuna    |
> | Hate    | 39.83  | 38.52  | 23.29 | 37.33 | 37.80 | 41.59     |\| 38.39  | 37.16  | 39.67 | 37.38 | 43.51 | **49.17** |\| 37.95  | 37.14  | 39.53 | 37.70 | 41.87 | 48.37     |
> | Emotion | 9.71   | 15.07  | 7.75  | 27.87 | 15.14 | 18.12     |\| 17.08  | 12.17  | 10.66 | 23.12 | 32.12 | 40.28     |\| 18.48  | 12.35  | 10.07 | 25.57 | 34.20 | **41.79** |
> | Humor   | 41.78  | 33.04  | 43.60 | 33.12 | 39.78 | 46.19     |\| 33.67  | 33.12  | 44.70 | 38.19 | 55.20 | 57.15     |\| 34.06  | 33.12  | 40.20 | 37.08 | 53.86 | **58.75** |
> | Irony   | 36.63  | 44.46  | 36.52 | 34.69 | 40.78 | **47.48** |\| 42.21  | 41.58  | 42.61 | 35.18 | 36.76 | 39.78     |\| 44.34  | 44.14  | 39.67 | 34.82 | 38.40 | 41.61     |
> | Average | 25.21  | 28.01  | 19.58 | 32.12 | 27.72 | 31.80     |\| 28.60  | 25.78  | 26.80 | 30.34 | 37.87 | 44.01     |\| 29.51  | 26.28  | 25.67 | 31.45 | 38.55 | **44.91** |
>
>
> **D.** Since there are large numbers of datasets (169 datasets) and languages (64 languages) in SPARROW, it is really hard to manually verify the quality of all the datasets. To ensure the quality of the dataset, we only select the datasets that are introduced in published papers (all the original papers are cited in Table 8-13), when we construct our benchmark. In our references, we also make sure that we link to each published paper describing each dataset to facilitate access to the annotation information of each dataset.
>
>
> **E.** We provide a concise study to evaluate open-source LLMs with prompts used for evaluating ChatGPT. We curate 55 datasets across 4 tasks from SPARROW and evaluate 6 models with ChatGPT prompts. As results shown in this table, we can see that BLOOM, LLaMA and Vicuna get sizable performance drops (>6 points decrease across 55 datasets), while BLOOMZ-P3, mT5, and mT0 demonstrated performance levels akin to those observed in previous experiments (<2 points different). We believe that the model performance is not only affected by the choice of prompts but also by which model is used and which dataset is tested on. We hope SPARROW will be a useful benchmark for investigating multilingual socio-pragmatic understanding in future work including opportunities to evaluate different types of prompts. We will add this analysis to the camera-ready.
>
> | Task    | lm-eval-harness | Prompts |       |       |       |        | ChatGPT | Prompts |       |       |       |        |
> |---------|-----------------|---------|-------|-------|-------|--------|---------|---------|-------|-------|-------|--------|
> |         | BLOOM           | BLOOMZ-P3  | mT5   | mT0   | LLaMA | Vicuna | BLOOM   | BLOOMZ-P3  | mT5   | mT0   | LLaMA | Vicuna |
> | Hate    | 39.83           | 38.52   | 23.29 | 37.33 | 37.80 | 41.59  | 18.39   | 31.34   | 28.96 | 38.19 | 18.37 | 37.33  |
> | Emotion | 9.71            | 15.07   | 7.75  | 27.87 | 15.14 | 18.12  | 8.61    | 20.07   | 7.57  | 29.63 | 17.61 | 8.65   |
> | Humor   | 41.78           | 33.04   | 43.60 | 33.12 | 39.78 | 46.19  | 41.59   | 44.99   | 41.34 | 34.13 | 41.59 | 33.12  |
> | Irony   | 36.63           | 44.46   | 36.52 | 34.69 | 40.78 | 47.48  | 27.33   | 26.02   | 36.08 | 34.31 | 26.02 | 34.70  |
> | Average  | 25.21           | 28.01   | 19.58 | 32.12 | 27.72 | 31.80  | 16.92   | 26.14   | 20.91 | 33.21 | 20.95 | 23.03  |

---

### Official Review · Reviewer_B91m · 2023-08-05

**Typos Grammar Style And Presentation Improvements:** Footnote 1 is unnecessary
**Soundness:** 5

**Excitement:**

4: Strong: This paper deepens the understanding of some phenomenon or lowers the barriers to an existing research direction.

**Paper Topic And Main Contributions:**

The paper introduces a new benchmark called Sparrow, for multilingual sociopragmatic meaning understanding. The dataset covers 64 languages and 169 datasets across 13 types of tasks. The paper also introduces a SPARROW score based on the benchmark and maintains and public leaderboard.  Finetuned and zero-shot LLMs are evaluated on this benchmark, and it is shown that zero-shot chatgpt lags behind task specific finetuned models by 12 points. Further, the paper analyzes multiple models across dimensions like overall performance, language, task-type etc.

**Questions For The Authors:**

A)How does few-shot in context learning perform?
B) How sensitive is the sparrow score to the choice of prompt?
C) How does the performance vary with source/domain? For example twitter vs youtube.

POST REBUTTAL: My concerns are addressed by the authors in detail. I recommend 'accept'.

**Reasons To Accept:**

1) The proposed benchmark is huge and one of the most comprehensive ones.
2) The benchmark consists of 64 languages which inculudes many low-resource languages.
3) The paper evaluates and compares the perfromance of 14 LLMs on the benchmark.
4) Detailed analysis of multiple models on the datasets across mutiple dimensions. For example overall perfromance of models, effect of instruction tuining, perfromance across tasks, languages etc.
5) The paper shows that for some low resource languages, translating the prompt to english and then making the prediction can give better results.

**Reasons To Reject:**

1) Finetuned LLMs are compared with zero-shot LLMs but evaluation of few-shot In-context learning is missing.
2) All the datasets included are text classification tasks. Datasets on language generation are not included.

POST REBUTTAL: My concerns are addressed by the authors in detail. I recommend 'accept'.

**Reproducibility:**

4: Could mostly reproduce the results, but there may be some variation because of sample variance or minor variations in their interpretation of the protocol or method.

**Reviewer Confidence:**

4: Quite sure. I tried to check the important points carefully. It's unlikely, though conceivable, that I missed something that should affect my ratings.

---

> ### Author Rebuttal · Authors · 2023-08-29
>
> Thank you for your insightful review of our paper.
>
> Reasons To Reject:
> * The focus of our paper is natural language understanding, as our title indicates. For this reason, datasets for natural language generation are outside of the scope of our work. We do not make any claims about them.
>
> Questions For The Authors:
>
> **A.**  Thanks for your questions. We report a new experiment here to investigate the few-shot in-context learning with 55 datasets from 4 tasks. As this table shows, we compare the results of 3-shot and 5-shot in-context learning with 0-shot results included in the paper. We observe that few-shot in-context learning does enhance the performance of most models (i.e., BLOOM, mT0, LLaMA, and Vicuna). With the increasing number of shots, we can also find that the performances increase. For example, we can see that the Vicuna model obtains average scores over 55 datasets of 31.80, 44.01, and 44.91 with 0, 3, and 5 shots, respectively. However, BLOOMZ-P3 and mT0 do not improve with few-shot in-context learning. We suspect this is because BLOOMZ-P3 and mT0 were finetuned only on NLP datasets, which makes them different from Vicuna that is finetuned with open-ended instructions. These two models (BLOOMZ-P3 and mT0) are also different from BLOOM and LLaMA in that they are finetuned on several NLP tasks only one of which is the sociopragmtic task (i.e., sentiment analysis). We believe this biases the behavior of these two models. We note that we do not perform few-shot in-context learning on sentiment analysis as part of the current experiment because it involves a large number of datasets.
>
> In the camera-ready, we will complete a few-shot in-context learning on all 169 datasets in SPARROW.
>
> |         | 0-Shot |        |       |       |       |           | 3-Shot |        |       |       |       |           | 5-Shot |        |       |       |       |           |
> |---------|--------|--------|-------|-------|-------|-----------|--------|--------|-------|-------|-------|-----------|--------|--------|-------|-------|-------|-----------|
> |         | BLOOM  | BLOOMZ-P3 | mT5   | mT0   | LLaMA | Vicuna    | BLOOM  | BLOOMZ-P3 | mT5   | mT0   | LLaMA | Vicuna    | BLOOM  | BLOOMZ-P3 | mT5   | mT0   | LLaMA | Vicuna    |
> | Hate    | 39.83  | 38.52  | 23.29 | 37.33 | 37.80 | 41.59     |\| 38.39  | 37.16  | 39.67 | 37.38 | 43.51 | **49.17** |\| 37.95  | 37.14  | 39.53 | 37.70 | 41.87 | 48.37     |
> | Emotion | 9.71   | 15.07  | 7.75  | 27.87 | 15.14 | 18.12     |\| 17.08  | 12.17  | 10.66 | 23.12 | 32.12 | 40.28     |\| 18.48  | 12.35  | 10.07 | 25.57 | 34.20 | **41.79** |
> | Humor   | 41.78  | 33.04  | 43.60 | 33.12 | 39.78 | 46.19     |\| 33.67  | 33.12  | 44.70 | 38.19 | 55.20 | 57.15     |\| 34.06  | 33.12  | 40.20 | 37.08 | 53.86 | **58.75** |
> | Irony   | 36.63  | 44.46  | 36.52 | 34.69 | 40.78 | **47.48** |\| 42.21  | 41.58  | 42.61 | 35.18 | 36.76 | 39.78     |\| 44.34  | 44.14  | 39.67 | 34.82 | 38.40 | 41.61     |
> | Average | 25.21  | 28.01  | 19.58 | 32.12 | 27.72 | 31.80     |\| 28.60  | 25.78  | 26.80 | 30.34 | 37.87 | 44.01     |\| 29.51  | 26.28  | 25.67 | 31.45 | 38.55 | **44.91** |
>
> **B.**  In our paper, we do not tailor prompts specifically for each model due to the large size of the datasets and models we investigated. Here, we provide a concise study to probe the model’s sensitivity to prompts. We curate 55 datasets across 4 tasks from SPARROW and evaluate 6 models with prompts used for evaluating ChatGPT. As results shown in this table, we can see that BLOOM, LLaMA and Vicuna get sizable performance drops (>6 points decrease across 55 datasets), while BLOOMZ-P3-P3, mT5, and mT0 demonstrated performance levels akin to those observed in previous experiments (<2 points different). We believe that the model performance is not only affected by the choice of prompts but also by which model is used and which dataset is tested on. We hope SPARROW will be a useful benchmark for investigating multilingual socio-pragmatic understanding in future work including opportunities to evaluate different types of prompts. We will add this analysis to the camera-ready.
>
> | Task    | lm-eval-harness | Prompts |       |       |       |        | ChatGPT | Prompts |       |       |       |        |
> |---------|-----------------|---------|-------|-------|-------|--------|---------|---------|-------|-------|-------|--------|
> |         | BLOOM           | BLOOMZ-P3  | mT5   | mT0   | LLaMA | Vicuna | BLOOM   | BLOOMZ-P3  | mT5   | mT0   | LLaMA | Vicuna |
> | Hate    | 39.83           | 38.52   | 23.29 | 37.33 | 37.80 | 41.59  | 18.39   | 31.34   | 28.96 | 38.19 | 18.37 | 37.33  |
> | Emotion | 9.71            | 15.07   | 7.75  | 27.87 | 15.14 | 18.12  | 8.61    | 20.07   | 7.57  | 29.63 | 17.61 | 8.65   |
> | Humor   | 41.78           | 33.04   | 43.60 | 33.12 | 39.78 | 46.19  | 41.59   | 44.99   | 41.34 | 34.13 | 41.59 | 33.12  |
> | Irony   | 36.63           | 44.46   | 36.52 | 34.69 | 40.78 | 47.48  | 27.33   | 26.02   | 36.08 | 34.31 | 26.02 | 34.70  |
> | Average  | 25.21           | 28.01   | 19.58 | 32.12 | 27.72 | 31.80  | 16.92   | 26.14   | 20.91 | 33.21 | 20.95 | 23.03  |
>
>
> **C.**  To investigate the cross-domain performance, we provide a study using binary sentiment analysis datasets that involve 8 different domains. We average the performance of each domain and present the relative gain compared to the random baseline. We can observe that the models’ performance varies across different domains. For example, ChatGPT performs better on Twitter data than YouTube comments. We also would like to mention that a direct comparison across domains is not feasible since each domain contains different tasks, datasets, and languages. While we try to have a fair comparison by using relative gain, we acknowledge the limitation of this experiment.
>
> | Source / Domain           | # Dataset | mBERT | InfoDCL | BLOOM  | BLOOMZ-P3 | mT5    | mT0   | LLaMA  | Vicuna | ChatGPT |
> |---------------------------|-----------|-------|---------|--------|-----------|--------|-------|--------|--------|---------|
> | Customer review           | 1         | 31.73 | 35.90   | -14.52 | 30.84     | -22.07 | 40.87 | 9.75   | -22.07 | 41.15   |
> | Facebook                  | 1         | 45.67 | 48.20   | 17.20  | 23.60     | -15.40 | 29.00 | 19.40  | -6.80  | 36.60   |
> | Moview review             | 3         | 27.70 | 32.52   | 0.42   | 30.46     | -2.12  | 15.58 | 7.69   | 2.69   | 37.52   |
> | Multiple online platforms | 2         | 43.37 | 42.50   | -13.10 | 23.20     | -15.31 | 16.90 | 0.91   | -8.39  | 33.15   |
> | Subtitle                  | 3         | 29.69 | 36.17   | -5.61  | 9.96      | -13.53 | 11.38 | -0.62  | 14.57  | 36.77   |
> | Twitter                   | 3         | 26.85 | 39.90   | -3.00  | 24.34     | -7.65  | 9.49  | 3.99   | 13.56  | 38.98   |
> | Weibo                     | 1         | 52.14 | 52.28   | -0.17  | 23.99     | -13.15 | 9.85  | 7.23   | 19.04  | 31.17   |
> | YouTube comment           | 4         | 31.20 | 35.25   | -18.16 | 20.05     | -26.00 | 10.31 | -18.77 | -23.56 | 24.54   |

---

### Official Review · Reviewer_fC9A · 2023-08-06

**Soundness:** 4

**Excitement:**

4: Strong: This paper deepens the understanding of some phenomenon or lowers the barriers to an existing research direction.

**Paper Topic And Main Contributions:**

This paper proposes a unified multilingual evaluation benchmark which could be very helpful for evaluating the multilingual abilities of LLMs. The authors also tried to answer a series of important questions through thorough analysis. The authors found that ChatGPT outperforms open-sourcing LLMs on nearly all tasks and instruction tuning also helps the model performance in many cases.

Overall I think this paper represents an important contribution to the evaluation of LLMs and I can see it being very helpful once the interactive leaderboard is released.

Other than all the merits, one major weakness of this paper is lacking theoretical grounding of "Sociopragmatic Understanding". It is unclear to me how are these categories selected and why, instead of simply combining all these available datasets. A relevant paper is "Do LLMs Understand Social Knowledge? Evaluating the Sociability of Large Language Models with SocKET Benchmark" (https://arxiv.org/abs/2305.14938).

**Reasons To Accept:**

1. A comprehensive and well-curated benchmark for evaluation LLM's ability of sociopragmatic understanding
2. The benchmark is beyond English, which is novel given most of the existing benchmarks are in English
3. The to be release interactive benchmark would be very helpful to the community

**Reasons To Reject:**

1. Lacking theoretical grounding of sociopragmatic understanding

**Reproducibility:**

4: Could mostly reproduce the results, but there may be some variation because of sample variance or minor variations in their interpretation of the protocol or method.

**Reviewer Confidence:**

4: Quite sure. I tried to check the important points carefully. It's unlikely, though conceivable, that I missed something that should affect my ratings.

---

> ### Author Rebuttal · Authors · 2023-08-29
>
> Thanks for your insightful and valuable review.
>
> Reasons To Reject:
>
> * We agree that we only define sociopragmatics briefly in the paper. To improve, we will provide a more appropriate grounding of sociopragmatic understanding to the camera-ready. We provide some background here as well. The term “sociopragmatic” can be understood as contextualized meaning in social interactions. An example of such a meaning is one that arises through human interaction, such as on social media platforms. This meaning is highly dependent on its particular context and transcends the literal interpretation of statements, encompassing insights about the individual's use of the language (investigated within the realm of "sociolinguistics" [1]) and their distinctive identities, as well as the underlying purposes guiding their communication (explored in "pragmatics" [2]) [3].
> We group different tasks in our benchmark by what we perceive to be an affinity between these tasks. For example, we group tasks of hate speech, offensive language, and dangerous language detection as anti-social language detection.
> Meanwhile, we keep particular tasks (such as sentiment analysis and emotion recognition) distinct due to the popularity of these tasks and since there are multiple datasets representing each of them. We acknowledge that this grouping is not necessarily theoretically motivated, but we feel that it does facilitate modular evaluations through the benchmark.
>
>
> [1] Sali A Tagliamonte. 2015. Making waves: The story of variationist sociolinguistics. John Wiley & Sons.
>
> [2] Jenny A Thomas. 2014. Meaning in interaction: An introduction to pragmatics. Routledge.
>
> [3] Chiyu Zhang, Muhammad Abdul-Mageed, and Ganesh Jawahar. 2023. Contrastive Learning of Sociopragmatic Meaning in Social Media. In Findings of the Association for Computational Linguistics: ACL 2023, pages 2405–2439, Toronto, Canada. Association for Computational Linguistics.

---

### Meta-Review · Area_Chair_tNG6 · 2023-09-08

**Recommendation:** 5

**Metareview:**

The paper presents SPARROW, a multilingual benchmark dataset for evaluating cross-lingual sociopragmatic understanding.  SPARROW consists of 169 datasets for 64 different languages across 13 task types divided into six umbrella categories of sociopragmatic meaning.  Using a proposed SPARROW score, an unweighted average across dataset-specific metrics, the paper evaluates a number of LLMs in different settings (zero-shot for generative models, finetuning for encoder-only models) on their ability to perform sociopragmatic understanding tasks.  They found that overall, smaller, fully finetuned models surpassed larger generative models in the zero-shot setting.

Overall, the paper was well-received by reviewers, who all appreciated the authors’ work in curating a comprehensive benchmark for sociopragmatic tasks across multiple languages and releasing it to the public.  In particular, reviewer fC9A highlights how useful the proposed benchmark would be to the broader NLP community in evaluating the capabilities of LLMs.  Reviewers B91m and GKU9 also appreciated the detailed set of experiments and analyses across various dimensions, such as the impact of finetuning and instruction tuning, performance across tasks and languages, and the impact of using translated prompts and inputs.

The major concern highlighted by reviewers B91m and GKU9, however, is that for the larger models, the submitted paper only presents results in a zero-shot setting without exploring few-shot in-context learning.  Additionally, both reviewers raised questions about how robust the results of their analyses were to the choice of prompt, as in the original paper, prompts were tailored to each task rather than to each model tested.  However, both of these concerns were addressed during the rebuttal/discussion period, with the authors presenting new results with few-shot ICL and prompt variation.  If accepted, the paper would be strengthened with the inclusion of those new results in the camera-ready version.  One potential concern, however, is how substantially the conclusions about overall performance over different models may change with the full set of results, as so far, the authors have not presented ICL results for their best generative model (ChatGPT) or one of the strongest tasks for generative LLMs.

The other concern raised by reviewer fC9A about the paper is the lack of theoretical grounding in how tasks were chosen and categorized for the benchmark.  The authors clarify their use of the term “sociopragmatic” and the motivation behind their grouping of selected tasks in their rebuttal.  While the rebuttal does not fully address the issue of the lack of theoretical motivation in how tasks were selected and grouped, the motivation behind the paper would nevertheless be strengthened by the additional discussion about the definition of sociopragmatics and the interaction between language, identity, and interaction in the rebuttal.

---

### Decision · Program_Chairs · 2023-10-07

**Decision:**

Accept-Main

**Comment:**

The paper presents SPARROW, a multilingual benchmark dataset for evaluating cross-lingual sociopragmatic understanding.  SPARROW consists of 169 datasets for 64 different languages across 13 task types divided into six umbrella categories of sociopragmatic meaning.  Using a proposed SPARROW score, an unweighted average across dataset-specific metrics, the paper evaluates a number of LLMs in different settings (zero-shot for generative models, finetuning for encoder-only models) on their ability to perform sociopragmatic understanding tasks.  They found that overall, smaller, fully finetuned models surpassed larger generative models in the zero-shot setting.

Overall, the paper was well-received by reviewers, who all appreciated the authors’ work in curating a comprehensive benchmark for sociopragmatic tasks across multiple languages and releasing it to the public.  In particular, reviewer fC9A highlights how useful the proposed benchmark would be to the broader NLP community in evaluating the capabilities of LLMs.  Reviewers B91m and GKU9 also appreciated the detailed set of experiments and analyses across various dimensions, such as the impact of finetuning and instruction tuning, performance across tasks and languages, and the impact of using translated prompts and inputs.

The major concern highlighted by reviewers B91m and GKU9, however, is that for the larger models, the submitted paper only presents results in a zero-shot setting without exploring few-shot in-context learning.  Additionally, both reviewers raised questions about how robust the results of their analyses were to the choice of prompt, as in the original paper, prompts were tailored to each task rather than to each model tested.  However, both of these concerns were addressed during the rebuttal/discussion period, with the authors presenting new results with few-shot ICL and prompt variation.  If accepted, the paper would be strengthened with the inclusion of those new results in the camera-ready version.  One potential concern, however, is how substantially the conclusions about overall performance over different models may change with the full set of results, as so far, the authors have not presented ICL results for their best generative model (ChatGPT) or one of the strongest tasks for generative LLMs.

The other concern raised by reviewer fC9A about the paper is the lack of theoretical grounding in how tasks were chosen and categorized for the benchmark.  The authors clarify their use of the term “sociopragmatic” and the motivation behind their grouping of selected tasks in their rebuttal.  While the rebuttal does not fully address the issue of the lack of theoretical motivation in how tasks were selected and grouped, the motivation behind the paper would nevertheless be strengthened by the additional discussion about the definition of sociopragmatics and the interaction between language, identity, and interaction in the rebuttal.